# Exploiting fluctuations in gene expression to detect causal interactions between genes

Euan Joly-Smith[1,2], Mir Mikdad Talpur[1], Paige Allard[3], Fotini Papazotos[3], Laurent Potvin-Trottier[3,4,5], Andreas Hilfinger[1,2,6,7]*

[1]Department of Physics, University of Toronto, Toronto, Canada; [2]Department of Chemical & Physical Sciences, University of Toronto Mississauga, Mississauga, Canada; [3]Centre for Applied Synthetic Biology, Concordia University, Montreal, Canada; [4]Department of Biology, Concordia University, Montreal, Canada; [5]Department of Physics, Concordia University, Montreal, Canada; [6]Department of Mathematics, University of Toronto, Toronto, Canada; [7]Department of Cell & Systems Biology, University of Toronto, Toronto, Canada

*For correspondence: andreas.hilfinger@utoronto.ca

Competing interest: The authors declare that no competing interests exist.

## eLife Assessment

By taking advantage of noise in gene expression, this **important** study introduces a new approach for detecting directed causal interactions between two genes without perturbing either. The main theoretical result is supported by a proof. Preliminary simulations and experiments on small circuits are **solid**, but further investigations are needed to demonstrate the broad applicability and scalability of the method.

**Abstract** Characterizing and manipulating cellular behavior requires a mechanistic understanding of the causal interactions between cellular components. We present an approach to detect causal interactions between genes without the need to perturb the physiological state of cells. This approach exploits naturally occurring cell-to-cell variability which is experimentally accessible from static population snapshots of genetically identical cells without the need to follow cells over time. Our main contribution is a simple mathematical relation that constrains the propagation of gene expression noise through biochemical reaction networks. This relation allows us to rigorously interpret fluctuation data even when only a small part of a complex gene regulatory process can be observed. We show how this relation can, in theory, be exploited to detect causal interactions by synthetically engineering a passive reporter of gene expression, akin to the established 'dual reporter assay'. While the focus of our contribution is theoretical, we also present an experimental proof-of-principle to demonstrate the real-world applicability of our approach in certain circumstances. Our experimental data suggest that the method can detect causal interactions in specific synthetic gene regulatory circuits in *Escherichia coli*, confirming our theoretical result in a narrow set of controlled experimental settings. Further work is needed to show that the approach is practical on a large scale, with naturally occurring gene regulatory networks, or in organisms other than *E. coli*.

## Introduction

Translating molecular abundance data into mechanistic models of cellular processes remains a major challenge of systems biology. High-throughput experimental techniques routinely produce statistical associations between genes through correlation measurements of gene expression variability (*Munsky et al., 2012*; *Stewart-Ornstein et al., 2012*; *Gandhi et al., 2011*; *Saint et al., 2019*; *Bageritz et al., 2019*). While such statistical associations can provide insightful clues, they do not directly identify causal interactions: even if two components *X* and Z are non-spuriously correlated, we cannot conclude that X affects Z because the causal connection could be reversed or a confounding factor could be regulating both components, even with statistical associations that go beyond correlations such as Granger causality (*Wasserman, 2004*; *Yuan and Shou, 2022*).

Perturbation experiments are a conceptually simple solution to avoid this problem and directly infer causal relationships in gene regulatory networks (*Ji et al., 2013*; *Tegnér et al., 2003*; *Kholodenko et al., 2002*). However, they come with practical challenges: drug perturbations can affect multiple targets at once (*Kaufman et al., 2022*) and genetic perturbations, for example, through changing gene copy numbers, are not guaranteed to keep cells in their physiologically relevant regime (*Welf and Danuser, 2014*; *Vilela and Danuser, 2011*). For large enough perturbations, everything is expected to affect everything else in the cell.

Here, we present a novel mathematical relation which can, in theory, be exploited to infer directional causal interactions between cellular components by utilizing stochastic fluctuations of cellular abundances in unperturbed cells. Existing work on analyzing non-genetic variability has focused on testing completely specified mechanistic models against such fluctuation data (*Munsky et al., 2012*; *Schmiedel et al., 2015*; *Baudrimont et al., 2019*; *Pedraza and Paulsson, 2008*; *Eling et al., 2019*). However, completely specifying mechanistic models of cellular processes often requires making a large number of assumptions about unknown details, which can make inferences based on such an approach unreliable (*Kaern et al., 2005*; *Hilfinger et al., 2016a*).

We thus propose a novel inference method to detect whether a gene *X* causally affects a gene *Z* by exploiting a mathematical identity that constrains an entire class of models. Under certain assumptions, we show how covariability measurements of molecular abundances can, in theory, be used to detect causal interactions by specifying only 'local' aspects of the underlying gene expression dynamics (*Hilfinger et al., 2016a*; *Joly-Smith et al., 2021*; *Hilfinger et al., 2016b*).

Experimentally, our proposed inference method is based on an approach similar to the 'dual reporter assay' (*Elowitz et al., 2002*; *Raser and O'Shea, 2004*; *Maamar et al., 2007*; *Raj et al., 2006*) previously established to quantify stochastic fluctuations in expression differences between copies of a gene of interest. However, instead of analyzing the covariance between the dual reporters, we focus on the covariance of each reporter with a third component of biological interest. We prove that under broad conditions, these covariances must be identical in the absence of the causal interaction we wish to establish. Violations of this relation can thus be used to detect causal interactions.

To illustrate the generality of our analytically proven covariance identity, we numerically verify the result in a wide variety of example systems. Additionally, we present an experimental proof-of-principle for a specific set of synthetically constructed gene regulatory circuits in *Escherichia coli* with known interactions. The data establish a baseline estimate for the accuracy of our approach with current experimental methods. Despite the arguable occurrence of a false positive result, the experimental data support our hypothesis that single-cell variability measurements might prove useful in detecting causal interactions in gene regulatory networks.

However, we have not shown that the approach is practical on a large scale we have not tested our approach using naturally occurring gene regulatory networks, and we have not tested our approach in organisms other than *E. coli*. Additional experimental tests beyond our proof-of-principle demonstration are thus needed to establish the scalability of our method and to show that the method reliably detects causal interactions between endogenous genes in more general experimental settings.

## Results

### Invariant covariance relation in the absence of causal interactions

Our main mathematical result is a covariance invariant that constrains the fluctuations of two molecular species in a partially specified reaction network. We consider a molecular species of interest *X*

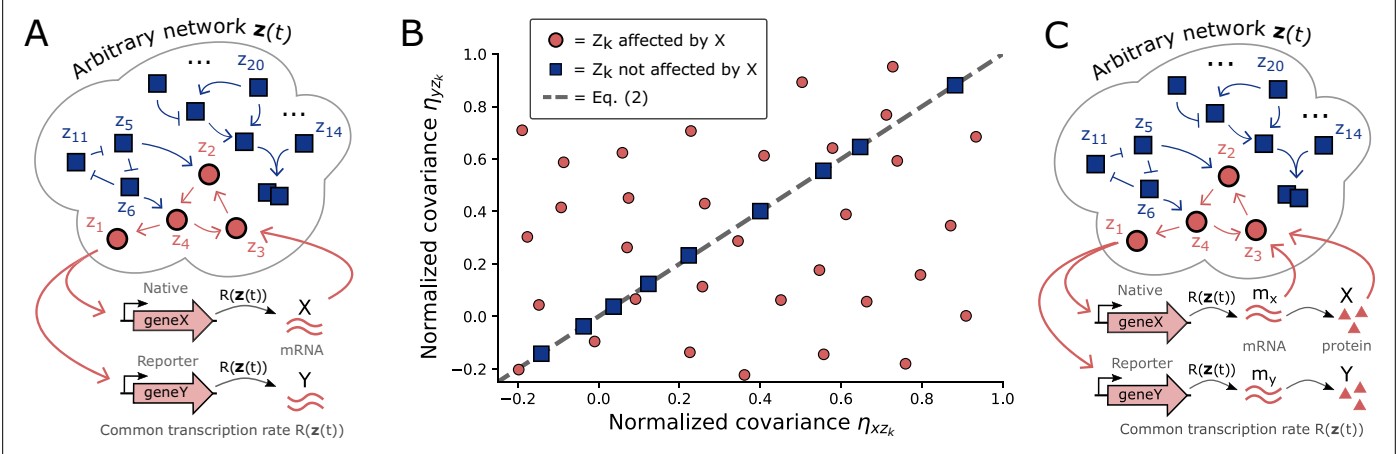

**Figure 1.** Causal interactions between genes can be inferred when non-genetic cell-to-cell variability violates the covariance invariant of *Equation 2*. (**A**) Consider an arbitrary network of interacting cellular components in which an engineered reporter $Y$ is introduced to act as a passive read-out of the transcriptional signal that regulates a gene of interest *geneX*, but itself does not regulate other cellular components. Any cellular component belongs to one of two groups: components affected by $X$ (red circles), and components not affected by $X$ (blue squares), where the arrows indicate directed causal biochemical effects. Specifically, an arrow from $Z_i$ to $Z_j$ denotes the existence of a reaction that changes $Z_j$ with a rate that depends on $Z_i$ (see Appendix 1 for details). (**B**) The dashed line of *Equation 2* constrains the normalized covariance between $X$ and any cellular component $Z_k$ not affected by $X$ (blue squares). In contrast, components $Z_k$ affected by $X$ (red circles) are not constrained by *Equation 2*. A violation of *Equation 2* thus implies the existence of a causal interaction from $X$ to $Z_k$. Data points are numerical simulations of specific example networks (see Appendix 1 for details) to illustrate the analytically proven theorem of *Equation 2*. (**C**) The invariant of *Equation 2* applies not only to transcriptional reporters but also when $X$ and $Y$ correspond to co-regulated fluorescent proteins with translation rates that scale with transcript abundances.

that is subject to first-order degradation and is made with an unspecified and time-varying production rate. A reporter species $Y$ acts as an identical but separate copy of $X$ that is subject to the same control input. This specifies the following stochastic reactions within a larger reaction network

$$x \xrightarrow{R(\mathbf{z}(t))} x + 1 \quad y \xrightarrow{\alpha R(\mathbf{z}(t))} y + 1$$
$$x \xrightarrow{\beta x} x - 1 \quad y \xrightarrow{\beta y} y - 1. \tag{1}$$

Here, $x$ and $y$ denote the number of copies of molecular species $X$ and $Y$, respectively, $\alpha$ is an arbitrary proportionality constant, and the shared production rate $R$ can depend in arbitrary ways on the abundances $z_k$ of any cellular component (including $X$ and $Y$ themselves), collectively denoted as $\mathbf{Z} := \{Z_1, Z_2, \ldots\}$. The birth-death events denoted in *Equation 1* produce or remove one molecule of $X$ or $Y$ at a time and occur stochastically with the state-dependent rates indicated above the arrows.

If the level of $X$ affects the rate of production or degradation of another component, say $Z_3$, we say $X$ causally affects $Z_3$ and we define causal effects transitively, that is if $X$ affects $Z_3$, and $Z_3$ affects $Z_5$, then $X$ affects $Z_5$. In other words, $X$ causally affects a cellular component if a directed path can be drawn from $X$ to said component in the biochemical reaction network as illustrated in *Figure 1A*.

*Theorem*: if neither $X$ nor $Y$ causally affect a third cellular component of interest $Z_k \in \mathbf{Z}$, then

$$\underbrace{\frac{\mathrm{Cov}(x, z_k)}{\langle x \rangle \langle z_k \rangle}}_{\eta_{xz_k}} = \underbrace{\frac{\mathrm{Cov}(y, z_k)}{\langle y \rangle \langle z_k \rangle}}_{\eta_{yz_k}}, \tag{2}$$

where angular brackets denote ensemble averages, and $\eta_{z_k x}$ denotes the normalized covariance between $Z_k$ and $X$ in a population of genetically identical cells that have reached a time-independent distribution of cell-to-cell variability and do not exhibit permanently zero degradation rates for $X$ and $Y$.

*Equation 2* makes no assumptions on the interactions of all the unspecified cellular components and allows arbitrary effects from the component of interest $Z_k$ onto $X$ and $Y$. The invariant constrains any cellular component $Z_k$ that is not affected by $X$ and $Y$ as long as the 'local' dynamics of $X$ and $Y$ is given by *Equation 1*. The detailed proof of *Equation 2* is presented in Appendix 1. The key condition

under which it holds is that the non-genetic population variability has reached a stationary state. In systems in which there is uncertainty about this condition, it can be directly verified experimentally.

Note, a violation of *Equation 2* implies the presence of a causal connection but not vice versa. This is a logical necessity because the dynamics of causally connected genes can be arbitrarily close to that of non-interacting genes.

In a later section, we discuss further generalizations of the dual reporter class defined in *Equation 1*, allowing for growing and dividing cells, measurement noise, and fluctuations in degradation rates as would be relevant for its application to experimental single-cell data. Note that due to the

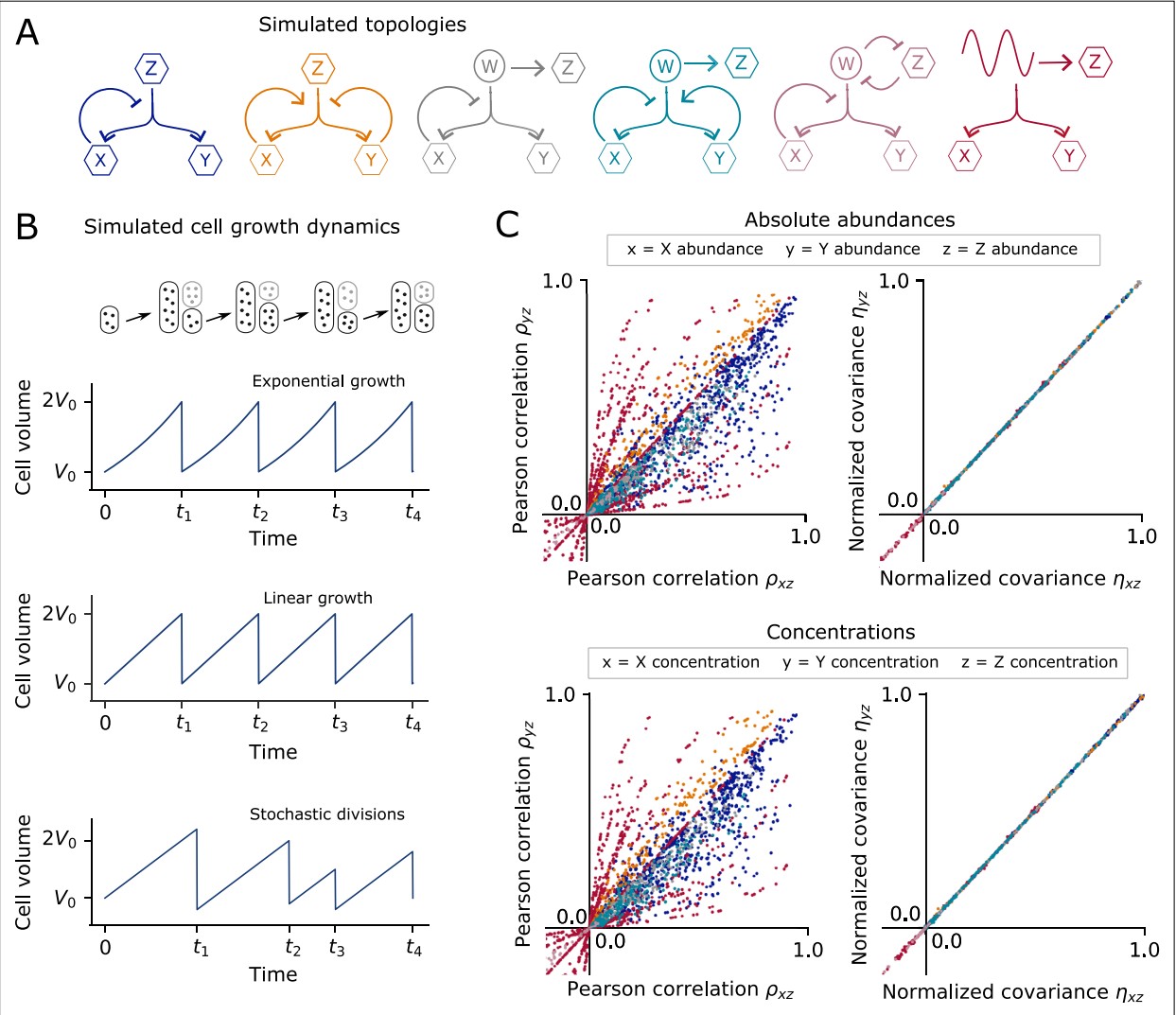

**Figure 2.** Numerical simulations of example systems confirm computationally that the analytically proven *Equation 2* constrains cellular abundances and concentrations in growing and dividing cells under fairly general assumptions as long as $X$ does not affect $Z$. (**A**) To numerically demonstrate the validity of *Equation 2* we consider ten example birth-death processes covering six different network topologies with non-linear rates, closed-loop feedback, time-varying upstream signals, and fluctuating degradation rates. In all cases, chemical species $X$ and $Y$ are co-regulated but do not affect a third chemical species $Z$ of interest. See *Appendix 1—table 1* for details of the simulated systems. (**B**) We consider three different cellular growth dynamics that affect molecular abundances through random partitioning of molecules at cell division and molecular concentrations through dilution. Additionally, system reaction rates depend in varying ways on the cell volume. At cell division, molecular abundances are assumed to be partitioned on average proportional to cell size. (**C**) Simulation results for each of the example topologies from panel A, subject to the different growth dynamics of panel B. Model parameters were varied over several orders of magnitude. The numerically obtained Pearson correlations do not satisfy a general relation, whereas normalized covariances satisfy *Equation 2* for absolute abundances as well as concentrations. Colors correspond to the topologies indicated in panel A (results for individual topologies are compared in *Appendix 1—figures 3 and 4*). For each dot, single-cell trajectories of 20,000 cell divisions were simulated at least 40 times, with the center of the dot corresponding to the average of the simulation ensemble (see Materials and methods).

possibility of feedback, the dynamics of *X* and *Y* is not generally symmetric even though *X* and *Y* are co-regulated (*Joly-Smith et al., 2021*), see *Appendix 1—figure 1A*. *Equation 2* is a statement about stochastic covariability and not a trivial relation based on the (incorrect) assumption that $x(t) = y(t)$. For example, the Pearson correlation coefficients $\rho_{xz_k}$, $\rho_{yz_k}$ are not necessarily equal even when *X* and *Y* do not affect $Z_k$, see *Figure 2* and *Appendix 1—figure 1B*.

## Detecting causal interactions between genes through experimentally engineered 'dual reporters'

Violation of *Equation 2* can be used to experimentally deduce the existence of a causal effect from a gene of interest *X* onto another gene of interest $Z_k$ as follows. If we engineer a co-regulated 'dual reporter' system with the properties defined in *Equation 1*, in which *Y* is a passive reporter (i.e. it does not significantly interact with other cellular components) that responds to the same input as *X*, then a violation of *Equation 2* implies that changes in *X* must causally affect the abundance of component $Z_k$. This is because intrinsic fluctuations from the expression of *X* propagate through the network and affect $Z_k$, but those from the passive reporter Y do not. As detailed next, this approach can be applied to dual reporters for gene expression both on the transcriptional as well as the translational level, see *Figure 1*.

### Transcriptional dual reporters

Consider two genes of interest, *geneX* with transcript *X* and *geneZ* with transcript $Z_k$. Our approach relies on engineering a passive reporter gene *geneY* with identical transcriptional control to *geneX* and mRNA lifetime, but with transcript that does not affect other cellular components, see *Figure 2A*. This could be achieved with a transcriptional reporter that is put under control of the same promoter as *geneX* and placed at a similar gene locus. One way to ensure that *geneY* interacts minimally with other cellular components would be to remove its start codon so that it will not be translated into a protein.

Co-regulated mRNA reporters have been engineered to satisfy the assumptions underlying *Equation 1*, with transcripts counted with smFISH (*Baudrimont et al., 2019*; *Raj et al., 2006*; *Skinner et al., 2013*). Alternatively, future improvements in RNAseq (*Saliba et al., 2014*; *Lubeck et al., 2014*; *Svensson et al., 2018*) accuracy might allow for reliable fluctuation measurements of transcript levels in single-cells through sequencing approaches.

### Translational dual reporters

The dual reporter assay has been frequently implemented as co-regulated fluorescent proteins (*Schmiedel et al., 2015*; *Pedraza and Paulsson, 2008*; *Elowitz et al., 2002*; *Bar-Even et al., 2006*). The invariant of *Equation 2* directly applies when *X* and *Y* are co-regulated fluorescent proteins with first-order translation rates and maturation times (see Appendix 1 Section 4B for details of the proof). Fluorescent proteins can then be used to detect causal interactions in gene regulation as follows. The gene of interest *geneX* is fused to a fluorescent protein to make a functional fusion *geneX-FP*. A passive reporter protein *Y* is made by introducing a spectrally distinguishable fluorescent protein *Y* under the control of the same (but distinct) promoter as *geneX-FP*. The expression level $z_k$ of *geneZ* can be measured either through a transcriptional reporter or a functional fusion protein with a third fluorescence reporter. Under the assumption that the fluorescent protein *Y* does not directly affect other cellular components, *Equation 2* directly applies to the covariances of fluorescence levels as long as *X* does not causally affect $Z_k$. As the normalized covariances in *Equation 2* are independent of scaling factors, standard fluorescence microscopy methods can be used without the need to determine absolute numbers. Two stable fluorescent proteins with similar maturation times (*Balleza et al., 2018*) should be chosen to ensure that the assumptions underlying the class of systems are satisfied. Additionally, the translation rates of *X* and *Y* need not be identical but can be proportional with a fluctuating proportionality factor as defined in the transitions of *Equation 1*. As a result, different fluorescent proteins with different ribosome binding sites, mRNA secondary structures, and gene lengths can be used as long as the translation rates remain proportional.

## Typical experimental single-cell data can be analyzed using the invariant relation of *Equation 2*

In the above sections, we introduced the main mathematical result and presented the basic logic of how it can be experimentally exploited using synthetically engineered gene expression reporters. Next, we describe how *Equation 2* generalizes to the broader class of systems necessary to analyze single-cell experimental data.

### *Equation 2* constrains abundances in growing and dividing cells

The class of stochastic processes presented in *Equation 1* constrains the dynamics of cellular abundances in stationary processes. However, experimental data typically report measurements of growing and dividing cells. Under the assumption that during cell division, molecular abundances are divided on average proportional to cell size, we can show (see Appendix 1) that *Equation 2* must be satisfied by the dual reporter abundances when $Z_k$ is a component not affected by *geneX*. In this analysis of cellular growth and division, we allow for partitioning noise, division time fluctuations, asymmetric divisions, and arbitrary growth-rate dynamics (see *Figure 2B*).

### *Equation 2* constrains concentrations in growing and dividing cells

Chemical reaction rates typically depend on molecular concentrations and not absolute abundance numbers. Under general assumptions (see Appendix 1 Section 12), we can show that the covariance constraint of *Equation 2* describes molecular concentrations in growing and dividing cells, see *Figure 2*. This result assumes that the abundance of *X* does not affect cell volume or growth rate. This requirement can be intuitively understood, because if *X* affects cell volume, then it causally affects the concentration of $Z_k$ which depends on volume (see Appendix 1 Section 12).

### *Equation 2* applies to reporters with fluctuating degradation rates and fluctuating translation rates

The class of systems defined in *Equation 1* assumes that the dual reporters are degraded in a first-order process with a constant rate parameter $\beta$, see *Equation 1*. This assumption can be relaxed: the invariant of *Equation 2* generalizes to the class of systems in which the degradation rate constant is an arbitrary function of all the cellular components that are not affected by the dual reporters, as long as it does not decay to zero (see Appendix 1 Section 4). As a result, the degradation constant can vary in time with arbitrary extrinsic fluctuations. Additionally, for the class of co-regulated fluorescent proteins, the translation rate parameters can also vary with arbitrary extrinsic fluctuations.

### A rule for detecting causality in the face of measurement uncertainty

Under the null hypothesis that there is no causal interaction from *X* to *Z*, *Equation 2* is equivalent to $r = 1$, where $r = \eta_{xz}/\eta_{yz}$ is the covariability ratio. Given 95% confidence intervals for an experimentally measured covariability ratio, we can conclude at the 2.5% significance level that there is a causal interaction from *X* to *Z* if $r = 1$ falls outside of the 95% confidence interval of the data (see *Appendix 1— figure 2* for an example).

### In silico validation of analytical results

In order to illustrate the generality of the above results and to perform a numerical sanity check, we simulated several stochastic birth-death systems in growing and dividing cells, see *Figure 2*. We modeled 10 systems made up of 6 network topologies in which a component *Z* is correlated with *X* and *Y* but is not affected by *X* , see *Figure 2A*. These systems include cellular components with non-linear reaction rates and feedback loops, fluctuating degradation rates, and confounding variables that affect all observed components. We analyzed each system subject to three different cellular growth dynamics: periodic exponential growth, periodic linear growth, and linear growth with fluctuating cell-division times and division sizes, see *Figure 2B*.

 To numerically test *Equation 2*, we generated random sample paths for abundances and concentrations using the Gillespie algorithm (*Gillespie, 1977*; *Voliotis et al., 2016*). System parameters were varied over several orders of magnitude. The numerical data verify that the normalized covariances of cellular abundances as well as cellular concentrations satisfy *Equation 2* in growing and dividing

cells, see *Figure 2C*. The same does not hold for the corresponding Pearson correlation coefficients, see *Figure 2C*. Note, due to finite sampling, any numerical simulation will necessarily show small deviations from the exact equality of *Equation 2*. Through statistical analyses and re-running systems for increased sampling, we confirmed that the (minuscule) deviations observed in numerical simulations were consistent with finite sampling; see Materials and methods.

In Appendix 1 Section 14 we simulate a larger example network made of 10 components in which *X* regulates a cascade of components. The results suggest that the relative distance down a regulatory cascade can be inferred from the degree of violation of *Equation 2*.

### *Equation 2* constrains data with significant measurement noise

Experimental techniques to measure mRNA abundances or fluorescence levels potentially introduce significant measurement noise. For example, when counting mRNA abundances, smFISH can lead to probabilistic undercounting noise and fluorescence microscopy can report fluorescence levels that include photon noise, read noise, and segmentation errors, while flow cytometry introduces Poisson noise when used with bacteria (*Galbusera et al., 2020*).

The invariant of *Equation 2* holds in the face of measurement noise as long as the noise is symmetric in *X* and *Y* measurements. We can show (Appendix 1 Section 16) that *Equation 2* holds in systems with arbitrary multiplicative noise that is independent of the *X* and *Y* signals, and additive noise that is independent of $Z_k$. Additionally, the invariant holds in the face of systematic undercounting that introduces a binomial readout of the signal of interest, along with Poisson-Gaussian noise. If, however, the experiment introduces a different type of noise in *X* as compared to *Y*, then *Equation 2* is no longer valid. It is thus important to choose similar measurement techniques while measuring the abundances of *X* and its passive reporter *Y*.

## Experimental proof-of-principle

Whether the above theoretical approach works in practice depends on two crucial questions: Can we reliably build dual reporter systems that satisfy the assumptions underlying *Equation 2*? Are experimental violations of *Equation 2* larger than measurement uncertainties for current experimental techniques?

As a first test to address these questions, we present an experimental proof-of-principle using synthetically engineered gene regulatory circuits in *E. coli*. These circuits consist of variants of the 'repressilator', a celebrated synthetic control circuit in which three genes respectively repress each other (*Elowitz and Leibler, 2000*; *Potvin-Trottier et al., 2016*).

### Causally connected genes that break the invariant

Inherent to our fluctuation approach is the direction of inference, that is violations of *Equation 2* imply the presence of a causal connection, whereas agreement with *Equation 2* does not imply the absence of causal interactions. The question thus becomes whether a biologically relevant genetic circuit with known causal connections ever breaks our covariance relation.

We engineered four synthetic circuits in which a TetR-YFP fusion protein (*X*) represses an RFP reporter (*Z*), and a passive CFP reporter (*Y*) is under the same transcriptional control as *X*, see *Figure 3A*. While the circuits differ in dynamics and connections, $X$ affects $Z$ in all cases. *Appendix 1—figure 17* for details of the synthetic circuits #1–4. Using time-lapse fluorescent microscopy with cells growing in a microfluidic device (see *Figure 3C*), we measured the normalized covariances from populations of genetically identical cells with high accuracy while discarding all temporal information.

We found that circuits #1 and #2 clearly broke the constraint, circuit #4 did not violate it, and circuit #3 showed significant deviations right at the limit of what can be reliably detected, see *Figure 3D*. Using the estimated error bars and a null hypothesis test at the 2.5% significance level, we find that the data from circuit #3 does not reject the null hypothesis of *Equation 2*, see *Appendix 1—figure 2*.

The above repressilator test circuits were chosen for their well-characterized genetic interactions and because oscillating abundances make for a convenient test of the constructs. However, large oscillations in Z in also make it much more challenging to detect violations of *Equation 2* because, in those circuits, the intrinsic noise we exploit is small compared to the periodically varying dynamics of the circuit. Indeed, the two clear violators correspond to altered versions of the repressilator circuit that do not oscillate (see *Appendix 1—figure 17*). These non-oscillating test cases may in fact be the

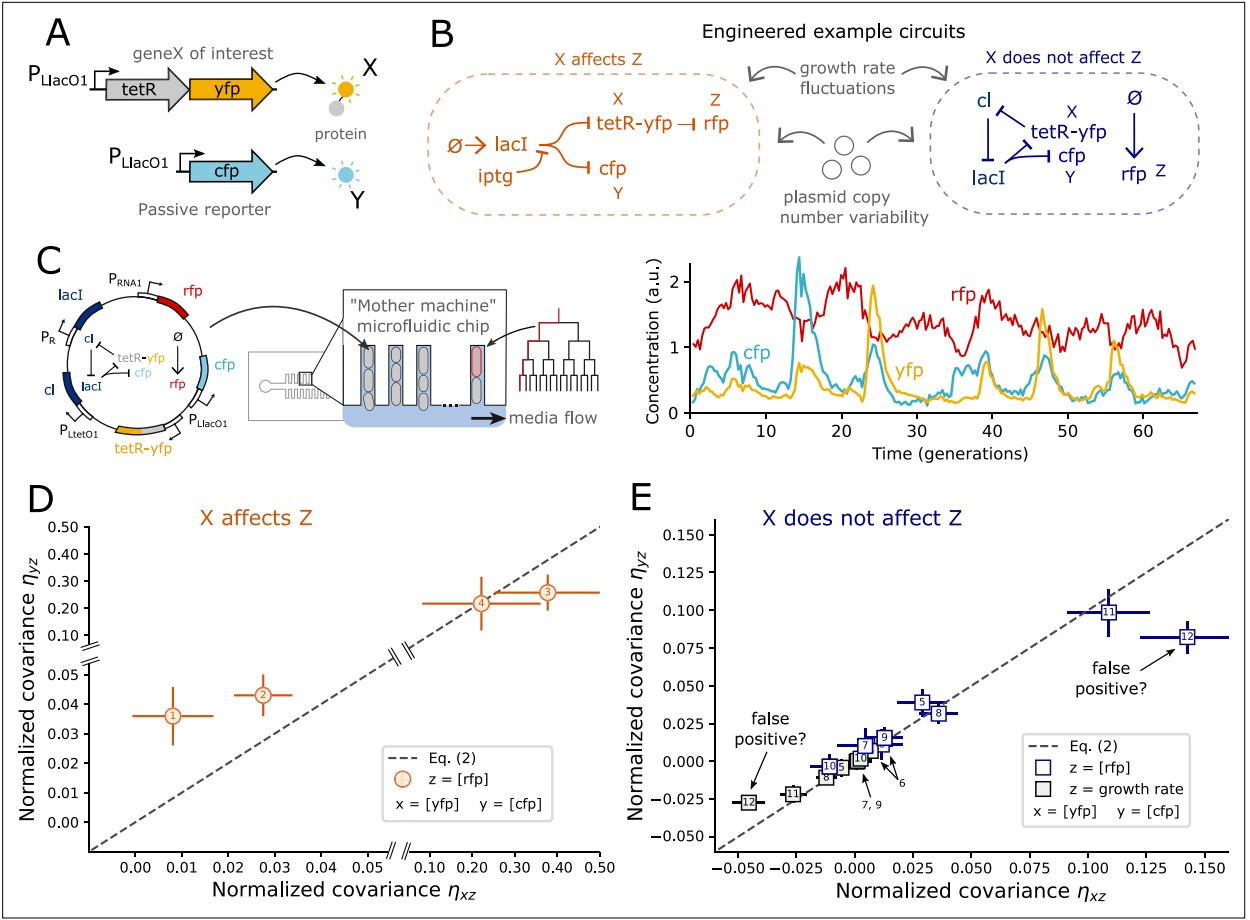

**Figure 3.** Experimental data from all but one synthetic regulatory circuit are consistent with the theory. (**A**) In all synthetic circuits, we considered TetR as our protein of interest ($X$) fused to YFP to allow for quantification through fluorescence microscopy. CFP ($Y$) was used as a passive read-out of the transcriptional control of *tetR* by placing it under the control of a copy of the same $P_{LlacO1}$ promoter as *tetR*. In all synthetic circuits $X$ and $Y$ are thus co-regulated by the LacI protein. (**B**) We constructed two different types of synthetic circuits using the repressilator motif (**Elowitz and Leibler, 2000**) as a basis. Left: example circuit in which TetR ($X$) causally affects a RFP reporter ($Z$). Right: negative control example circuit in which RFP was expressed constitutively and thus expected to be independent of TetR levels. In this circuit, $X$ does not causally affect $Z$ but $X$ and $Z$ are correlated due to plasmid copy number fluctuations. (**C**) *E. coli* cells with the synthetic circuit encoded on the pSC101 plasmid were grown in a microfluidic device and observed over hundreds of cell divisions while daughter cells were washed away. Fluorescence levels of YFP, CFP, and RFP were measured simultaneously for hundreds of mother cells, along with cell area, cell length, and the growth rate. For each strain, the time-lapse data of all cells were combined into a population distribution from which the normalized covariances were computed. All temporal information was thus discarded and not used in the analysis. (**D**) Two out of four causal interactions were clearly detected through violations of **Equation 2**. Of the other two, one was clearly not detected, whereas one was situated right at the limit of experimental detectability but did not violate the null hypothesis test at a 2.5% significance level (see **Appendix 1—figure 2**). See Materials and methods for details of our error analysis. Numbers indicate the synthetic circuits as listed in **Appendix 1—figure 17**. (**E**) All but one negative control circuits led to data consistent with **Equation 2** (dashed line). Numbered synthetic circuits are listed in **Appendix 1—figures 18 and 19**. The two inconsistent data points corresponding to the same synthetic circuit (number 12), which is presented in detail in **Figure 4**. The false positives either imply that our method works imperfectly or that we detected an unexpected causal interaction from TetR onto growth rate in this circuit. We present evidence for the latter interpretation in the next section.

most relevant assessment of using **Equation 2** in natural gene regulatory circuits. All the above circuits were made of genes encoded on a low copy number plasmid. We have not shown that our method can successfully detect causal interactions in a chromosomally encoded gene regulatory circuit. However, plasmid copy number fluctuations will reduce the degree of violation of **Equation 2** by reducing the fraction of reporter variability that is due to the intrinsic noise we exploit (see Appendix 1 Section 15). A priori, causal connections between chromosomally encoded genes are thus expected to be easier rather than harder to detect with our method.

Overall, the experimental data confirm that **Equation 2** has the fundamental power to experimentally detect causal interactions. However, the precision of current experimental techniques lies at the

edge of what is necessary to detect physiologically relevant interactions when genes show large variability relative to the intrinsic noise we exploit. This highlights the importance of correctly estimating measurement uncertainty in our approach. Our error bars estimate sampling error and errors from non-even sample illumination, temporal drift, background fluorescence, and autofluorescence. These were estimated using a bootstrapping approach where the corrections and the normalized covariances were computed recursively to samples of the data, see Materials and methods, *Appendix 1—figure 6*.

## Negative controls

Our crucial theoretical result is that *Equation 2* must be satisfied by any circuit in which *X* does not affect *Z*. To experimentally test this prediction, we engineered test cases in which *X* and *Y* are co-regulated, while *Z* is a component that is not affected by *X*. This took one of two forms: First, we constructed eight different circuits in which RFP (*Z*) was expressed constitutively using the pRNA1 promoter as previously used as a segmentation marker in *E. coli* (*Potvin-Trottier et al., 2016*; *Lord et al., 2019*). We used various synthetic circuits, and the RFP reporter was either integrated chromosomally or included on the circuit plasmid (the latter leading to correlations between RFP and our synthetic components without introducing a causal interaction). Second, as an additional control, for each circuit, we also considered the cellular growth rate as the unaffected cellular property Z. While the growth rate affects reporter concentrations through dilution, we did not expect cellular growth to be affected by our synthetic circuits, which made growth rates a convenient additional negative control.

We found the data from all but one synthetic circuit consistent with our predicted invariant; see *Figure 3E*. Note, the false positives correspond to the same synthetic circuit: one data point is from taking *Z* as the RFP concentration, while the other data point is from taking *Z* as the growth rate.

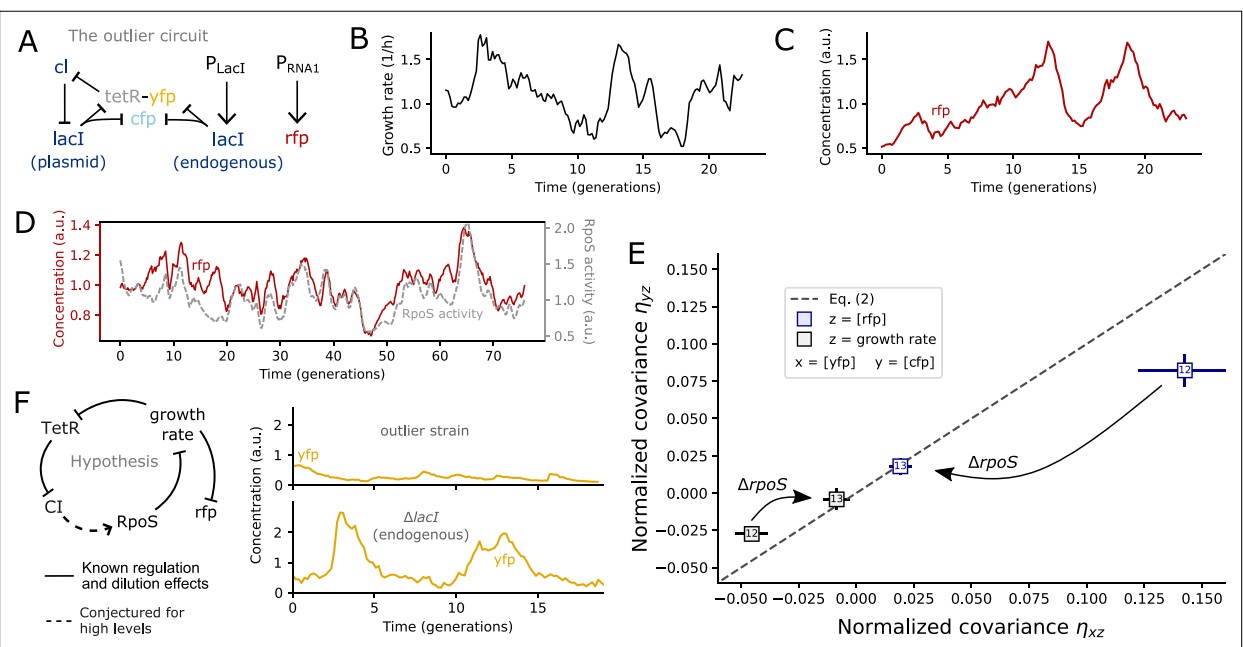

**Figure 4.** In the outlier strain, the bacterial stress response is triggered, which in turn regulates the 'constitutive' RFP reporter. (**A**) The outlier strain consists of the repressilator circuit encoded on the pSC101 plasmid with an endogenous copy of the *lacI* gene encoded on the chromosome. This endogenous source of LacI disrupts the regular oscillations of the repressilator circuit. RFP was chromosomally expressed under the control of the pRNA1 promoter (*Lin-Chao and Bremer, 1987*). (**B**) The outlier strain exhibited large growth rate variability. (**C**) The outlier strain exhibited large variability in RFP levels with the highest peaks occurring after the slowest growth periods. (**D**) RFP levels were strongly correlated with RpoS activity as quantified through the known RpoS target gadX (*Sampaio et al., 2022*). (**E**) Deletion of the *rpoS* gene made the outlier strain consistent with *Equation 2*. (**F**) Expressing LacI endogenously leads to long periods of low TetR-YFP levels rather than regular repressilator oscillations. Because TetR is the only repressor of the *cI* gene in the synthetic circuit, we hypothesize that during periods of low TetR, CI expression is so high that resource competition with the also highly expressed RFP triggers the bacterial stress response. This interaction, hypothesized to be present in cells with high CI and RFP expression levels, is indicated with a dashed arrow. Solid arrows indicate interactions with direct experimental support (*Patange et al., 2018*).

To rule out experimental error, we repeated the experiment twice while swapping the YFP and CFP reporters, which confirmed the result (see *Appendix 1—figure 7*). Additionally, we confirmed the result using a different cell segmentation pipeline (see *Appendix 1—figure 8*).

Taken at face-value, our method thus produced two false positives (out of 16 tests), suggesting that our approach could be valuable but somewhat unreliable. However, next, we present experimental evidence that indicates the violating data points were not false positives but may have correctly identified an unexpected causal effect from *X* on *Z* in the outlier circuit.

## Evidence that RpoS-mediated stress response affected cellular growth in the outlier circuit

We observed that the outlier strain (*Figure 4A*) exhibited periods of slow growth (*Figure 4B*), suggesting that cells underwent periods of stress. Indeed, although RFP was thought to be constitutively expressed, RFP exhibited clear temporal dynamics that were negatively correlated with growth rate (*Figure 4C*, *Appendix 1—figure 9*). These fluctuations were larger than other strains expressing RFP from the same chromosomal gene (*Appendix 1—figure 10*), suggesting that the outlier circuit caused additional fluctuations in the RFP concentration and growth rate.

We hypothesized that in the outlier strain, the 'constitutive' RFP reporter was being affected by the bacterial stress response regulator RpoS (*Lange and Hengge-Aronis, 1991*; *Tanaka et al., 1993*). By measuring the transcriptional activity of the known RpoS target *gadX* (*Sampaio et al., 2022*), we observed variable RpoS activity with pulses that correlated with periods of slow cellular growth (*Appendix 1—figure 11*). Similar pulsing behavior of RpoS triggering periods of slow growth has been reported previously (*Sampaio et al., 2022*; *Patange et al., 2018*). RFP levels were in turn strongly correlated with RpoS activity (*Figure 4D*), as expected for genes for which transcription rates are constant but cell growth is slowed down by RpoS activity. Indeed, upon deleting RpoS, RFP fluctuations as well as growth rate fluctuations satisfy *Equation 2*, see *Figure 4E*.

Next, we discuss why the RpoS stress response was triggered in the outlier circuit in a TetR-dependent manner. In the outlier strain, LacI was chromosomally expressed, which represses TetR independent of the repressilator circuit. Instead of regular oscillations of TetR, we observed extended periods of very low concentrations, see *Figure 4F*. We conjecture that during those extended periods of low TetR levels, expression of CI is exceedingly high, which in combination with the already highly expressed RFP, leads to resource competition in cells that ultimately triggers RpoS-mediated stress response. Note, CI is under the control of the very strong pLtetO1 promoter, which can initiate the transcription of up to 0.3 mRNA/s in bacteria when fully induced (*Lutz and Bujard, 1997*) and has been shown to burden *E. coli* (*Shachrai et al., 2010*). RFP is under the control of another strong promoter, pRNA1, previously used as a bright segmentation marker (*Potvin-Trottier et al., 2016*; *Lord et al., 2019*), which we find to affect growth rate (*Appendix 1—figure 12*). This RpoS-mediated causal effect of TetR on RFP would explain the violations of *Equation 2* by the outlier circuit, why deletion of *rpoS* removes the outlier data, and why the same circuit without endogenous *lacI* (strain #5) did not violate the constraint, see *Figure 3E*.

The disappearance of the deviation from *Equation 2* in the outlier circuit with RpoS knockout provides evidence that the outlier data may not have been a false positive, but detected an unexpected causal interaction mediated by cellular burden and RpoS.

## Discussion

A ubiquitous problem in understanding gene regulatory networks is identifying which genes regulate the expression levels of which other genes. Stochastic fluctuations of molecular abundances within cells provide a natural source of information about such regulation. Here, we presented a mathematical identity that potentially allows for the translation of such single-cell variability data into causal interactions. Because correlations do not imply causation, this method requires experimental intervention, but crucially, it does not require perturbing the physiological state of the cells.

Note that agreement with *Equation 2* does not prove the non-existence of such a causal interaction because the lack of deviation can have two reasons: the causal interaction could be negligible or the effect of the intrinsic noise we exploit is not large enough compared with other sources of variability of the target gene. In other words, while deviations of *Equation 2* rigorously establish

causal interactions, 'absence of this evidence does not constitute evidence of absence' (*Oliver and Billingham, 1971*).

## Proposed additional tests

Our data from synthetic circuits in *E. coli* provide a proof-of-principle application in which our theoretical idea survived a first contact with experimental realities in specific synthetic circumstances. Synthetic gene circuits are useful for testing our network inference method because they provide reliably known interactions and can easily be modified to produce an array of circuits to test. However, synthetic circuits are limited in that they may not describe physiologically relevant interactions in naturally occurring systems. In addition, the preceding section suggests that synthetic circuits can put a metabolic burden on cells which can lead to unintended causal interactions. Further experiments are needed to demonstrate that the approach can detect causal interactions in natural gene regulatory circuits.

This requires applying the method on an endogenous network with reliably known natural interactions. For instance, the widely studied lac operon presents an exemplary test candidate: $X$ would correspond to LacI, which represses the expression of *lacZ*. Using standard cloning techniques in molecular biology, the endogenous *lacI* and *lacZ* genes in a wild-type strain of *E. coli* can be replaced with *lacI-FP$_1$ lacZ-FP$_2$* fusion genes, where $FP_1$ and $FP_2$ correspond to two fluorescent proteins of choice. A passive reporter $Y$ would correspond to a spectrally distinguishable fluorescent reporter gene $FP_3$ placed on the chromosome at similar gene loci to *geneX*. Violation of *Equation 2* would demonstrate that the method can detect the causal interaction between the LacI repressor and the lac operon expression. Additional negative control genes that are not regulated by LacI can then be studied to test the validity of *Equation 2* under broader conditions.

Note that to minimize the metabolic burden of fluorescent proteins, the invariant *Equation 2* can be tested sequentially, that is doing the experiment with visible $X$ and $Z$ first, followed by an experiment to measure $Y$ and $Z$ next. If hybridization or sequencing techniques such as smFISH or scRNAseq are used to measure transcript abundances, then endogenous *geneX* and *geneZ* transcript abundances can be measured directly, and the only metabolic burden comes from the expression of the *geneY* reporter.

## Limitations of this study

Our method relies on a handful of key assumptions about the molecular reporters used. In the absence of standardized synthetic parts, we may not know whether a given engineered system satisfies these assumptions. Only if the assumptions are satisfied, can violations of the invariant rigorously identify the existence of a directed causal interaction.

While the presented experimental study works as a proof-of-principle to illustrate the approach and establish its practical feasibility, at face value, it also produced a false positive result. Through additional perturbation experiments, we argue that the outlier circuit includes an unexpected causal connection. However, the evidence for this conclusion is weaker than the proven theoretical results presented. The fact that our interpretation of the perturbation experiments will be debated may serve as an illustration of how valuable a fluctuation-based approach would be that avoids perturbation experiments altogether.

The negative control circuits in this study were chosen because they lacked a causal interaction from $X$ to $Z$ when considering the known interactions between the used promoters and proteins. However, the false positive result suggests that unexpected causal interactions can still exist in these circuits. Further perturbation experiments and additional controls would thus be beneficial to directly verify that $X$ did not affect $Z$ in our synthetic negative control circuits.

Additionally, all experimental synthetic circuits were tested in *E. coli* and eukaryotic test cases remain to be investigated. In eukaryotic cells, promoter sequences may not be the sole factor determining transcription rates. For instance, dual reporter genes with identical promoters have been engineered in mammalian cells (*Raj et al., 2006*). The expression of these genes underwent bursts that are not coordinated when the genes are placed at distant gene loci (*Raj et al., 2006*). While the invariant of *Equation 2* holds in the face of such stochastic bursting, it only holds when, on average, the burst frequency is the same for each dual reporter (see Appendix 1 Section 8). Additional consistency checks should also be reported to test whether the passive reporter reads out the same signal as the

gene of interest. This could be done, for example, by testing the invariant of *Equation 2* with other cellular components that are known to not be affected by *geneX*, or by direct measurement of the transcription activity using a separate method (*Lenstra et al., 2016*). However, it is possible that the existence of some of the many other factors of gene regulation in eukaryotes, such as the presence of enhancers or epigenetic modifications, could make the proposed approach difficult to apply. It thus remains to be shown that a passive reporter can be engineered to read out the same signal as a naturally occurring gene of interest in the face of additional factors of gene regulation in eukaryotes.

# Materials and methods
## Numerical simulation details
Exact simulated single-cell time trajectories of the abundances were generated using the standard Gillespie algorithm (*Gillespie, 1977*), with an additional step to account for time-dependent rates and divisions (*Voliotis et al., 2016*) (see Appendix 1 Section 13 for the exact algorithm used). The time trajectories for the concentrations correspond to the abundance trajectories divided by the cell volume trajectories $V(t)$, with the latter being simulated independently of the species abundances. Simulations were performed with Python.

### Simulated cell growth and division models
Three cellular growth dynamics were simulated. The first and second are linear and exponential growth, respectively, with constant division times and symmetric cell divisions. Here, $V(t)$ trajectories were generated analytically as periodic functions. The volume is reduced by a factor $1/2$ at evenly spaced division times $\{\tau_i\}$. Between division times $\tau_i$, $\tau_{i+1}$ the volume is given by $V(t) = V_0 \cdot (1 + \frac{t-\tau_i}{t_d})$ for the linear case and $V(t) = V_0 \cdot 2^{(t-\tau_i)/t_d}$ for the exponential case, where $t_d$ is the time between divisions, and $V_0$ is the volume right after a division. The third simulated volume dynamics is linear growth with stochastic division times and asymmetric divisions. Here, a constant linear growth rate $\frac{dV}{dt} = u$ is used. Division times $\{\tau_i\}$ and division factors $\{a_i\}$ are picked recursively: The $\tau_{i+1}$ is taken from a normal distribution with mean set to the doubling time:

$$\tau_{i+1} = \mathcal{N}\left(\frac{V(\tau_i)}{u}, 0.2\right) + \tau_i.$$

where $\mathcal{N}(\mu, \sigma)$ is the normal distribution with mean μ and standard deviation $\sigma$. Until then the volume grows linearly with constant rate

$$V(t) = V(\tau_i) + u \cdot (t - \tau_i) \quad \forall \, \tau_i \leq t < \tau_{i+1}.$$

At $\tau_{i+1}$ the volume is reduced by factor $a_{i+1}$ taken from a normal distribution with mean set to the ratio of cell volumes at the beginning and end of the cycle

$$a_{i+1} = \mathcal{N}\left(\frac{V(\tau_i)}{V(\tau_i) + u(\tau_{i+1} - \tau_i)}, 0.2\right),$$

$$V(\tau_{i+1}) = a_{i+1} \cdot \left(V(\tau_i) + u(\tau_{i+1} - \tau_i)\right).$$

As a result, cells that grow more (less) than double in size during a cycle tend to divide with larger (smaller) division factors, ensuring the volume trajectories do not eventually expand (decay) to infinity (zero).

At division, molecular abundances are reduced according to a binomial splitting with probability given by the division factor. For example, if the volume is reduced by a factor of 0.4 at a cell division, that is $V \rightarrow 0.4V$, then each molecule has probability 0.4 to remain in the followed daughter cell.

### Computing normalized covariances from trajectories
Normalized covariances were computed by integrating over the trajectories to obtain time averages for first and second moments. This is equivalent to using the distribution given by calculating the fraction of the total system time spent in each sampled state. In the ergodic regime, this distribution converges to the stationary distribution of the ensemble.

## Simulated systems

We simulated ten groups of systems defined by their rate functions (see *Appendix 1—table 1* for details). For each group, each model parameter was picked randomly multiple times from the set {0.1, 1, 10}. This was done for each of the three volume dynamics we considered. If a simulated system gave an average abundance in one of the components less than 0.01 molecules, then the system was omitted to avoid numerical errors that arise from divisions of small numbers when computing the normalized covariances. In total, there are 6522 resulting simulations which are plotted in *Figure 2C*.

## Confidence intervals for finite sampling error

For each system, simulated trajectories were generated forty times, and then repeated until the percentage errors of the normalized covariances, taken as the standard error of the mean divided by the mean, reached less than 1% or the number of simulations reached 1000. Each trajectory ran for 20,000 cell divisions and started with a unique random number generator seed. Each component abundance was set to 1 at the beginning of each trajectory, and the cell cycle time was set to 0. We let the simulated cells reach 200 cell divisions before we start to compute the time average integrals for the moments, in order for the effect of the initial condition to dissipate, and we analyze the systems' cyclo-stationarity state. The final normalized covariance for each system corresponds to the average taken over the ensemble of simulated trajectories with 95% confidence intervals given by twice the standard error. Distributions for the uncertainties for each class of systems are plotted in *Appendix 1—figures 3 and 4*.

For a given system, *Equation 2* was tested by verifying that the ratio $\eta_{xz}/\eta_{yz}$ produced a 95% confidence interval that encompassed the predicted value of 1. The test was satisfied by 6369 out of 6522 systems (97.7%), consistent with the definition of confidence intervals for finite sampling. The percentage of outliers for each simulated process is plotted in *Appendix 1—figure 5*. Re-simulating the outliers for twice the number of simulations led to a reduction in the standard error by a factor of $1/\sqrt{2}$, with *Equation 2* being satisfied by 143 of the 153 outliers (93.5%). Re-simulating the remaining 10 violators with four times the number of simulations reduced the errors by a factor of 1/2, and all of the 10 outliers satisfied *Equation 2* when quantified at such high numerical accuracy.

## Strain and plasmid construction

All strains, plasmids, full construction details are provided in the Appendix 1. The base background strain used throughout the manuscript is *E. coli* MG1655.

The base plasmid used throughout the manuscript is the repressilator (*Elowitz and Leibler, 2000*; *Potvin-Trottier et al., 2016*). It consists of three genes: *tetR* from the Tn10 transposon, *cI* from bacteriophage $\lambda$, and *lacI* from the lactose operon, which have promoters that are repressed by LacI, TetR, and CI, respectively. All genes are placed on the low-copy pSC101 plasmid which is inserted into *E. coli* MG1655.

Using standard molecular biology techniques, the repressilator plasmid sequence was altered in a number of ways to construct the different circuits. The *tetR* repressor was set as the gene of interest *geneX*, and TetR levels were measured through fluorescence measurements of a TetR fusion to the yellow fluorescent protein *mVenus NB* (YFP) (*Balleza et al., 2018*; *Nagai et al., 2002*). We engineered the passive reporter *geneY* by expressing a different fluorescent protein *SCFP3A* (CFP) (*Balleza et al., 2018*; *Kremers et al., 2006*) under the control of the same pLacO1 promoter as *geneX* on the same pSC101 plasmid. Both *X* and *Y* levels are co-regulated by LacI concentrations. To ensure equal degradation times, we used a version of the repressilator in which the three repressing genes lack degradation tags so that circuit proteins are removed predominantly from cell division and dilution (*Potvin-Trottier et al., 2016*). The fluorescent proteins *mVenus NB* and *SCFP3A* were chosen since they both have a short maturation half-life (4.1 ± 0.3 min and 6.6 ± 0.5 min respectively at 37°C) compared to the time scale of the repressilator oscillations (~ 5 hr).

For the RFP, we used a modified *mkate2* hybrid (with improved translational efficiency) throughout all experiments, which consists of the *mCherry* N-terminal 11 amino acids followed by the *mKate2* sequence. In the experiment shown in *Figure 4D* in which GFP was used to measure *gadX* expression, we used *gfpmut2* as a transcriptional reporter for the *gadX* promoter, placed on a low-copy pSC101 plasmid (taken from the *E. coli* promoter library *Zaslaver et al., 2006*). For *Figure 4E*, we used a strain of *E. coli* MG1655 with *rpoS* deletion taken from the *E. coli* Keio Knockouts library (*Baba et al., 2006*).

We chose the repressilator as the base circuit because the circuit interactions are known and the resulting dynamics (periodic oscillations) can be used as a consistency check to ensure that the TetR-YFP fusion is functional (i.e. still represses the $\lambda$ gene).

## Mother machine experiment

We used a microfluidic device commonly called 'mother machine' to follow growing and dividing cells for hundreds of generations in controlled environments (*Wang et al., 2010*; *Allard et al., 2022*). Single cells are trapped in micrometer-wide trenches. As cells grow and divide, the mother cells remain trapped while newborn daughter cells are washed away by the constant flow of growth media. Automated time-lapse microscopy and cell segmentation software enables us to track hundreds of cells in each experiment, while precisely measuring cell fluorescence, growth rate, and cell size. The time-series data for each mother cell can be pooled into a distribution from which the normalized covariances can be computed.

f *Equation 2* does not require the use of time-series data, we used the additional temporal information as a consistency check for our assumptions. In particular, the highly correlated trajectories of the YFP and CFP fluorescence (see *Appendix 1—figures 18 and 19*) are consistent with our assumption that these reporters are co-regulated. Additionally, the observed oscillations in the closed loop circuits are indicative that the TetR-YFP fusion has not lost its function as a result of the added fluorescent protein. Moreover, the time-series data allows us to measure cellular growth rates which can be used to test *Equation 2* as shown in *Figure 3D*.

## Microfluidic chip preparation

Polydimethylsiloxane (PDMS; Sylgard 184 Silicon Elastomer, Thermo Fisher Scientific) was mixed at a 10:1 (monomer   curing agent) ratio, poured on top of a 1.0 μm tall wafer and degassed for 1 hr at room temperature before baking for an additional 1.5 hr at 65°C. After careful removal of the PDMS from the wafer, individual PDMS chips were cut out with a razor blade. The inlet and outlet holes were punched with a 0.75 mm biopsy puncher (World Precision Instruments). The PDMS chips were sonicated in isopropyl alcohol (Thermo Fisher Scientific) for 30 min and dried at 65°C for 15 min. Glass coverslips (Thermo Fisher Scientific: 22x40 mm #1.5) were cleaned with 1M potassium hydroxide (KOH, Sigma Aldrich) for 20 min. The PDMS chips were bonded to the glass coverslips using a plasma cleaner (Oxygen flow rate at 45 sccm, power at 30W for 15 s, Tergeo Plasma Cleaner, PIE Scientific). The completed microfluidic device was heated to reinforce the plasma bonding at 100°C for 10 min, then 65°C for 30 min.

## Cell preparation

*E. coli* strains were grown overnight in LB with appropriate antibiotics to select for cells containing the constructed plasmids. At ~ 3-4 hr prior to the experiment start time, the overnight cultures were diluted 1:100 in imaging media consisting of M9 salts, 10% (v/v) LB, 0.2% (w/v) glucose, 2 mM MgSO4, 0.1 mM CaCl2, 1.5 μM thiamine hydrochloride and 0.85 g/L Pluronic F-108 (Sigma Aldrich, surfactant to prevent cells sticking to the surface of a microfluidic device). At an OD600 of 0.2-0.4, cells were loaded into the main feeding channel of the microfluidic chip and centrifuged at 5000 × *g* for 10 min to push cells into the cell trenches. The feeding channels were then connected to syringes filled with imaging media using Tygon tubing, and the microfluidic chip was placed in a temperature-controlled incubation chamber set at 37°C. Media was pumped through the feeding channel using syringe pumps (New Era Pump System), first at a rate of 5 μL/min for ~ 0.5-1 hr to get the cells comfortable in the trenches, then at a high rate of 100 μL/min for 1 hr to clear the inlets and outlets. The rate of media flow was then set back to 5 μL/min for the duration of the experiment.

## Microscopy and image acquisition

Images were acquired using a Zeiss Axio Observer inverted microscope equipped with a 63x Plan-Apochromat M27 oil objective (NA 1.40), an Orca Flash 4.0 LT camera (Hamamatsu), and an LED epifluorescence illuminator (Zeiss Colibri 7). The experiments were performed inside a temperature-controlled incubation chamber set at 37°>C. To reduce photobleaching, the exposure time (100 ms) and light intensity (10-20%) were set low, with 16-bit CZI images taken every 5-8 min. Focal drift was corrected automatically with the Definite Focus 2 (Zeiss) monitoring and compensation system using

an infrared laser (850 nm). In all experiments in which CFP, YFP, and RFP are measured simultaneously (i.e. all experiments except for the one shown in *Figure 4D*), the following filter sets were used for acquisition: CFP (Semrock FF02-475/20-25), YFP (Chroma CT560/39bp), RFP (Zeiss Filter Set 91 HE LED), with respective Zeiss Colibri 7 illumination wavelengths set to 430 nm (CFP), 511 nm (YFP), and 590 nm (RFP), along with the Zeiss TBS 450/538/610 beam splitter. For the experiment in Fig. 4D in which a gfp reporter is used to measure gadX expression, the following filter sets were used for acquisition: GFP (Chroma ET525/50m) and cy5 (Zeiss BP 690/50) with Zeiss Colibri 7 illumination wavelengths set to 475 nm (GFP) and 630 nm (cy5), along with Zeiss QBS 405/493/575/653 beam splitter.

## Data analysis

### Segmentation

Because all three fluorescence channels were used to measure synthetic circuit components throughout, we used the bright-field channel for segmentation. This was achieved with the automated deep learning-based cell segmentation software DeLTA (*Lugagne et al., 2020*; *O'Connor et al., 2022*). The U-Net deep learning models used for channel identification and cell segmentation were trained with a large dataset built from 5 mother machine experiments performed in the Potvin Lab that were generated with the same microscopy and image acquisition setup. In these training datasets, a bright RFP is expressed, and a thresholding segmentation pipeline is performed on the RFP channel images to obtain segmentation masks. The U-Net deep learning models from DeLTA are trained with these segmentation masks in combination with bright-field images from the training dataset as described in *Lugagne et al., 2020*; *O'Connor et al., 2022*. The training dataset was segmented with the same thresholding pipeline used in previously described procedures (*Potvin-Trottier et al., 2016*; *Norman et al., 2013*). We also trained a DelTA model to use the RFP channel for segmentation and obtained similar results on the strains with bright RFP fluorescence. In Appendix 1, we include videos of a mother machine with segmentation boundaries obtained from our DeLTA models, along with the single-cell growth trajectories.

We estimated the YFP, CFP, and RFP single-cell concentrations as the average fluorescence intensities of all pixels in the segmentation mask of each cell. Cell area is measured as returned by opencv's contourArea() over the segmentation mask. Cell length is measured by fitting a rotated bounding box to the segmented cell. We only analyzed data from the mother cells trapped at the top of the growth chambers.

### Single-cell traces construction

For a given cell chamber with the Region of Interest identified by the U-Net model, a single-cell trace was constructed by selecting the segmented cell in each frame with highest average vertical pixel location. This, in effect, keeps track of the mother cells trapped at the top of the growth chambers.

Each single-cell trace went through a manual purging process. Traces with cell areas that are not growing and dividing were removed, as they are expected to correspond to dead cells. Traces with growing and dividing cell areas with very low constant fluorescence correspond to cells that lost the inserted plasmid due to random partitioning at cell division. These traces were also purged, but were analyzed separately to measure the cell autofluorescence as described in the next section. Moreover, segmentation and tracking errors were reduced by purging any parts of the traces that exhibited a clear non-biological anomaly.

Cell divisions were identified by a sudden decrease in the cell area. A division is called whenever the cell lengths dropped to less than 80% of their previous value.

### Temporal drift correction

Focal drift was reduced automatically with a 850 nm infrared laser (Zeiss Definite Focus 2). However, we used an oil objective, which can cause temporal drift from spreading of the oil over time. We thus applied an additional correction to the data as follows.

In a given mother machine experiment, we computed the mean YFP, CFP, and RFP signals of all the cells at each time frame. For example, the mean YFP time-trace is given by

$$\langle \text{YFP}_{avg} \rangle_t = \frac{1}{N_{\text{cells}}(t)} \sum_{i=1}^{N_{\text{cells}}(t)} \text{YFP}_{avg}^i(t), \tag{3}$$

where $\text{YFP}_{avg}^i(t)$ is the average YFP intensity taken over the segmented area of the $i^{th}$ cell at time frame $t$, and $N_{\text{cells}}(t)$ is the number of surviving cells at time frame $t$. The mean time-traces of each fluorophore are fitted with a second-degree polynomial according to a least-squares fit. To correct for the drift, we multiply a measured intensity with the reciprocal of the respective fitted polynomial (see **Appendix 1—figure 13**). For example, if $f(t)$ is the obtained fit of $\langle \text{YFP}_{avg} \rangle_t$ from **Equation 3**, we correct all the $\text{YFP}_{avg}$ measurements as follows

$$\text{YFP}_{avg}^i(t) \rightarrow \text{YFP}_{avg}^i(t) \cdot \frac{f(0)}{f(t)}$$

$$\forall i \in [1, N_{\text{cells}}(t)] \quad \& \quad \forall \text{timeframes } t,$$

where $f(0)$ is the fitted function $f(t)$ taken over the first time frame.

## Uneven illumination and background correction

A single image frame from our setup encompasses ~30 cell trenches with total spatial extension of ~ 100 μm. This results in uneven illumination onto the cells. We use the linear gain model (**Peng et al., 2017**; **Smith et al., 2015**) to correct for uneven illumination and background fluorescence:

$$I^{meas}(x) = I^{true}(x) \times S(x) + D(x), \tag{4}$$

where $I^m(x)$ and $I^{true}(x)$ are the measured and true intensities at horizontal pixel position $x$ respectively, the multiplicative term $S(x)$ models the uneven illumination, and the additive term $D(x)$ is any background noise that is present when no light is incident on the sensor. For our imaging setup, we decompose $I^{true}(x)$ as follows

$$I^{meas}(x) = \left( I^{FP}(x) + I^a(x) + I^{media}(x) + I^b(x) \right) \times S(x) + D(x), \tag{5}$$

where $I^{FP}(x)$ is the intensity from the fluorescent proteins, $I^a$ is the autofluorescence of the cell, $I^{media}$ is any fluorescence originating from the media, and $I^b$ is any fluorescence originating from the PDMS background of the microfluidic device. The total background is given by $b(x) = I^b(x) \times S(x) + D(x)$, in which case the imaging model becomes

$$I^{meas}(x) = \left( I^{FP}(x) + I^a(x) + I^{media}(x) \right) \times S(x) + b(x). \tag{6}$$

We use a 'prospective' approach to correct for $S(x)$ and $b(x)$ as follows. For each segmented cell in an image, we compute the average intensity in a box of dimensions 50 by 50 pixels located 15 pixels above the cell trench (see **Appendix 1—figure 15A**). This in effect estimates $b(x)$ at the $x$ position of just above each cell trench. In each image, this $b(x)$ is removed from the $I^{meas}(x)$ measurements of each segmented cell. To estimate the uneven illumination gain function $S(x)$, we pool all of the single cell fluorescence measurements according to their horizontal pixel position $x$. Data is binned into bins of size 50 pixels and averages are taken at each bin (see **Appendix 1—figure 15B**). The resulting curve gives an estimate of $\langle I^{FP} + I^{af} + I^{media} \rangle \times S(x)$. To estimate $S(x)$, we fit the curve with a 3rd degree polynomial $p_S(x)$. To correct for the uneven illumination gain function, we multiply the fluorescence measurements by the reciprocal of $p_S(x)$:

$$I^{FP}(x) + I^a(x) + I^{media}(x) = \left( I^{meas}(x) - b(x) \right) \cdot \frac{p_{S,max}}{p_S(x)},$$

where $p_{S,max}$ is the max value of the fit $p_S(x)$ over $x$.

## Autofluorescence and fluorescing media

The preceding section described how the background and the uneven illumination were corrected using the linear gain imaging model of **Equation 6**. Even with the background correction, the

fluorescence profile across growth chambers empty of cells is not negligible compared to the profiles of some cell-containing chambers, see *Appendix 1—figure 15*. This indicates that media fluorescence is not negligible in experiments with strains that produce low levels of fluorescence. Here, we show how we estimated the autofluorescence $I^a$ and the media fluorescence $I^{media}$ to obtain the sought-after fluorescent protein fluorescence $I^{FP}$.

In each mother machine experiment, there was a subset of cells that lost the synthetic circuit plasmid due to random partitioning at cell division. These cells were used to estimate $I^a + I^{media}$. Cells that lost the plasmid were identified manually by observing single-cell traces: When a cell loses the plasmid, the CFP and YFP fluorescence decay rapidly and reach a constant low signal level for the remainder of the experiment, see *Appendix 1—figure 14*. The segments of the time traces at constant low signal level were cut and saved, with the autofluorescence and the media fluorescence taken as the average fluorescence of the saved traces. Around 10–30 cases of plasmid loss occur for each strain in a mother machine experiment.

Note that $I^a$ and $I^{media}$ do not need to be corrected in the RFP channel measurements. This is because RFP is taken as the component either affected or not affected by *X*. For the systems in which RFP is not affected by *X*, any *function* of RFP (like adding autofluorescence and media fluorescence) will also not be affected by *X* and the invariant of *Equation 2* will be satisfied. Alternatively, if X affects RFP, then it will generally also affect any typically expected function of RFP. We did not correct for $I^a$ and $I^{media}$ in the RFP channel because a fraction of circuits have RFP located on the chromosome and not the plasmid.

## Estimating confidence intervals

The corrections from the preceding sections rely on the distribution of single-cell measurements. As a result, sampling error affects the accuracy of the corrections, along with the final estimators of the normalized covariances. We applied two methods to estimate normalized covariance confidence intervals by taking into account sampling error and its effect on the corrections.

First, we use bootstrapping, where the data corrections and the normalized covariances are computed over many samples of the data, allowing for replacement in sampling. If an experiment produces *N* single-cell traces of cells with plasmid and *M* single-cell traces of cells that lost the plasmid, random samples of size *N* and *M* of each respective ensemble are taken. The corrections from the previous sections are then applied using the samples, with the corrected data pooled into a distribution from which the normalized covariances are computed. This is repeated until 100 normalized covariances $\eta_{sample}$ have been computed for each sample. The final reported normalized covariance corresponds to the average over the $\eta_{sample}$, with confidence intervals given by twice the standard deviation. See *Appendix 1—figure 16* and Appendix 1 Section 17 for details.

Second, we use sampling without replacement, where the single-cell traces from each experiment are divided into 7 to 10 disjoint sets, and the corrections and the normalized covariances are computed for each set. The reported normalized covariances correspond to the average of those computed from the sets, with error bars being the standard error of the mean (see Appendix 1 Section 17 for details).

A single mother machine experiment typically produced 100–500 single-cell traces of cells with plasmid and 7–30 single-cell traces of cells that lost the plasmid due to random partitioning at division. The computed normalized covariances from the bootstrapping method are shown in *Figure 3D,E*, while those from the alternative splitting method are shown in *Appendix 1—figure 6*. The two methods give similar results.

## Growth rate estimation

We define growth rate in this work as the rate of change of the cell length normalized by cell length: $\frac{1}{L}\frac{dL}{dt}$. To estimate the growth rate from the single-cell traces, we assume exponential growth between division events.

In that case, if there is no division event that occurs at time frames $i-1$, $i$, and $i+1$ then the growth rate at time $t_i$ is computed as $\frac{\ln(L_{i+1})}{\ln(L_{i-1})}\frac{1}{2\Delta t}$, where $L_i$ is the cell length at time $t_i$, $L_{i-1}$ is the length at the preceding frame, $L_{i+1}$ is the length at the subsequent frame, and $\Delta t$ is the time between frames (5–8 min). If there is a division event at the subsequent frame $i+1$ then the growth rate is computed

as $\frac{\ln(L_i)}{\ln(L_{i-1})}\frac{1}{\Delta t}$ . If there is a division event at the current frame $i$ then the growth rate is computed as $\frac{\ln(L_{i+1})}{\ln(L_i)}\frac{1}{\Delta t}$ .

For illustration, we smoothed the growth rate measurements shown in **Figure 4B** using a moving average filter with a window of 5 frames. We did not smooth the data when computing the normalized covariances and CVs shown in **Figure 3**.

## Acknowledgements

We thank Raymond Fan, B Kell, Seshu Iyengar, Sid Goyal, and Josh Milstein for many helpful discussions. This work was supported by the Natural Sciences and Engineering Research Council of Canada and a New Researcher Award from the University of Toronto Connaught Fund. Simulations were performed on the Niagara, Beluga, and Narval supercomputers at the SciNet HPC Consortium. SciNet is funded by the Canada Foundation for Innovation; the Government of Ontario, Ontario Research Fund - Research Excellence, and the University of Toronto. EJS gratefully acknowledges a doctoral research visit grant to support their experimental work in LPT's lab at Concordia University.

## Additional information

### Funding

| Funder | Grant reference number | Author |
| --- | --- | --- |
| Natural Sciences and Engineering Research Council of Canada | DG RGPIN-2019-06443 | Euan Joly-Smith Mir Mikdad Talpur Andreas Hilfinger |
| Natural Sciences and Engineering Research Council of Canada | DG RGPIN-2019-07002 | Laurent Potvin-Trottier |
| University of Toronto | Connaught Fund | Andreas Hilfinger |
| Canada Foundation for Innovation | John R. Evans Leader Fund 38290 | Laurent Potvin-Trottier |
| Government of Ontario, Ontario Research Fund - Research Excellence University of Toronto | | Andreas Hilfinger |

The funders had no role in study design, data collection and interpretation, or the decision to submit the work for publication.

### Author contributions

Euan Joly-Smith, Conceptualization, Data curation, Formal analysis, Investigation, Visualization, Methodology, Writing – original draft, Writing – review and editing; Mir Mikdad Talpur, Paige Allard, Fotini Papazotos, Investigation, Methodology; Laurent Potvin-Trottier, Resources, Supervision, Funding acquisition, Writing – original draft; Andreas Hilfinger, Conceptualization, Formal analysis, Supervision, Funding acquisition, Investigation, Writing – original draft, Project administration, Writing – review and editing

### Author ORCIDs

Euan Joly-Smith ⓘ https://orcid.org/0000-0002-3792-2231
Andreas Hilfinger ⓘ https://orcid.org/0000-0002-2411-6775

Reviewer #2 (Public Review): https://doi.org/10.7554/eLife.92497.4.sa1
Author response https://doi.org/10.7554/eLife.92497.4.sa2

# Additional files

## Supplementary files
MDAR checklist

## Data availability
Numerical simulation code and data, code for analyzing the single-cell traces to compute the normalized covariances, python code used to run the DeLTA deep learning segmentation pipeline, as well as plasmid sequences, are all available on Github (https://github.com/ejolysmith/Exploiting-fluctuations-causal-interactions-manuscript, copy archived at *Joly-Smith, 2025*). The segmented and tracked single-cell traces along with our trained U-Net models used for segmentation and chamber identification are available on Zenodo (https://doi.org/10.5281/zenodo.15616830).

The following dataset was generated:

| Author(s) | Year | Dataset title | Dataset URL | Database and Identifier |
|---|---|---|---|---|
| Joly-Smith E, Talpur MM, Allard P, Papazotos F, Potvin-Trottier L, Hilfinger A | 2025 | Data for Exploiting fluctuations in gene expression to detect causal interactions between genes | https://doi.org/10.5281/zenodo.15616830 | Zenodo, 10.5281/zenodo.15616830 |

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

## Appendix 1

## 1. Supplementary figures

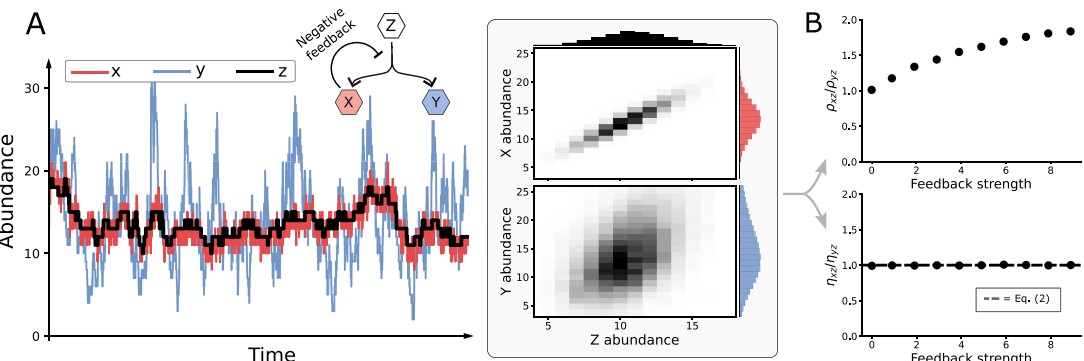

**Appendix 1—figure 1.** The dynamics of the 'dual reporters' *X* and *Y* are not necessarily symmetric despite the fact that *X* and *Y* have the same production rate. (**A**) Feedback in gene regulation can suppress or amplify fluctuations in the abundances of molecular species. Here we show an example system of the class of *Equation 1*, where *X* affects the shared transcription rate of *X* and *Y* but does not affect the shared upstream variable *Z*. In this example, the intrinsic fluctuations of *X* are suppressed through negative feedback, whereas those from *Y* are not. This leads to unequal dynamics and cell-to-cell variability in *X* and *Y* levels. See Section 10 for details of the simulated example system. (**B**) The invariant *Equation 2* does not hold in terms of Pearson correlation coefficients, i.e., $\rho_{xz} \neq \rho_{yz}$ even when *X* does not affect *Z*. In this example system, the divergence between $\rho_{xz}$ and $\rho_{yz}$ increases as the feedback strength increases (see Section 10). In contrast, *Equation 2* constrains all systems in which *X* does not affect *Z*, even when the *X* and *Y* levels have vastly different dynamics and cell-to-cell variability. Note, the covariances and correlations can be determined from the static snapshots of cell-to-cell variability indicated in panel A. The time-trace to indicate the *dynamics* of the reporters is for illustration purposes only.

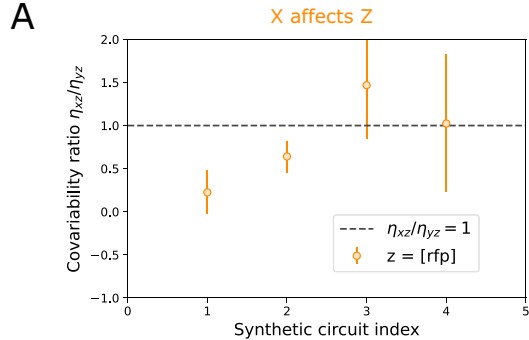

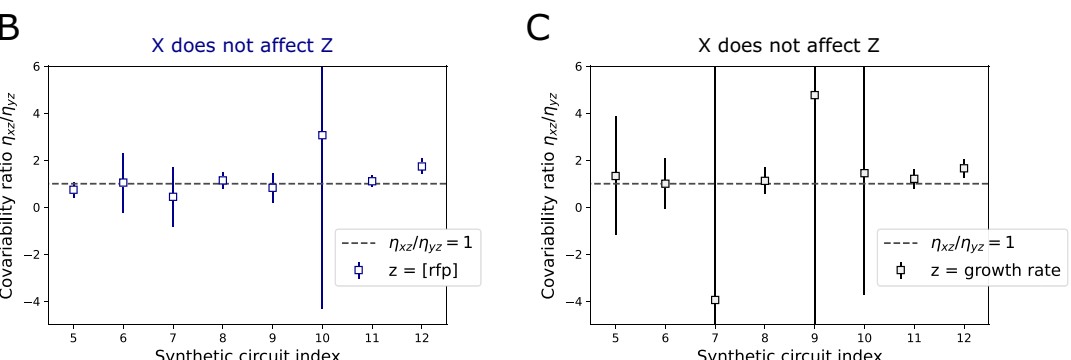

**Appendix 1—figure 2.** Using the null hypothesis test on the measured covariability ratios. Using the null hypothesis test on the measured covariability ratios. (**A**) Plotted are the measured covariability ratios $\eta_{xz}/\eta_{yz}$ for *Appendix 1—figure 2 continued on next page*

*Appendix 1—figure 2 continued*

synthetic circuits shown in *Figure 3D* and *Appendix 1—figure 17* in which *X* affects *Z*. Error bars correspond to estimated 95% confidence intervals, which were determined with error propagation (using the uncertainties python package) using the 95% confidence intervals for $\eta_{xz}$ and $\eta_{yz}$ that were estimated as detailed in Section 17 and the Materials and methods of the main text. Under the null hypothesis that there is no causal interaction from *X* to *Z*, the expected value of $\eta_{xz}/\eta_{yz} = 1$. If the measured covariability ratio is above (below) the expected value of 1, then there is a 2.5% probability that the expected value of 1 falls below (above) the lower (upper) error bar. As a result, under a significance level of 2.5%, we say there is a causal interaction from *X* to *Z* if the value of $\eta_{xz}/\eta_{yz} = 1$ does not fall within the measured 95% confidence intervals. Using this rule, our method detected two causal interactions (synthetic circuits #1 and #2) out of four. (**B**) Under the rule for detecting causality outlined in A, we detect a causal interaction in synthetic circuit #12 when we set the RFP concentration as the *Z* component of interest. Plotted are the measured covariability ratios $\eta_{xz}/\eta_{yz}$ for the synthetic circuits shown in *Figure 3E* and *Appendix 1—figures 18 and 19*. (**C**) Similarly to B, we detect a causal interaction in synthetic circuit # 12 when we set the growth rate as the *Z* component of interest. All other negative control circuits satisfy the null hypothesis test.

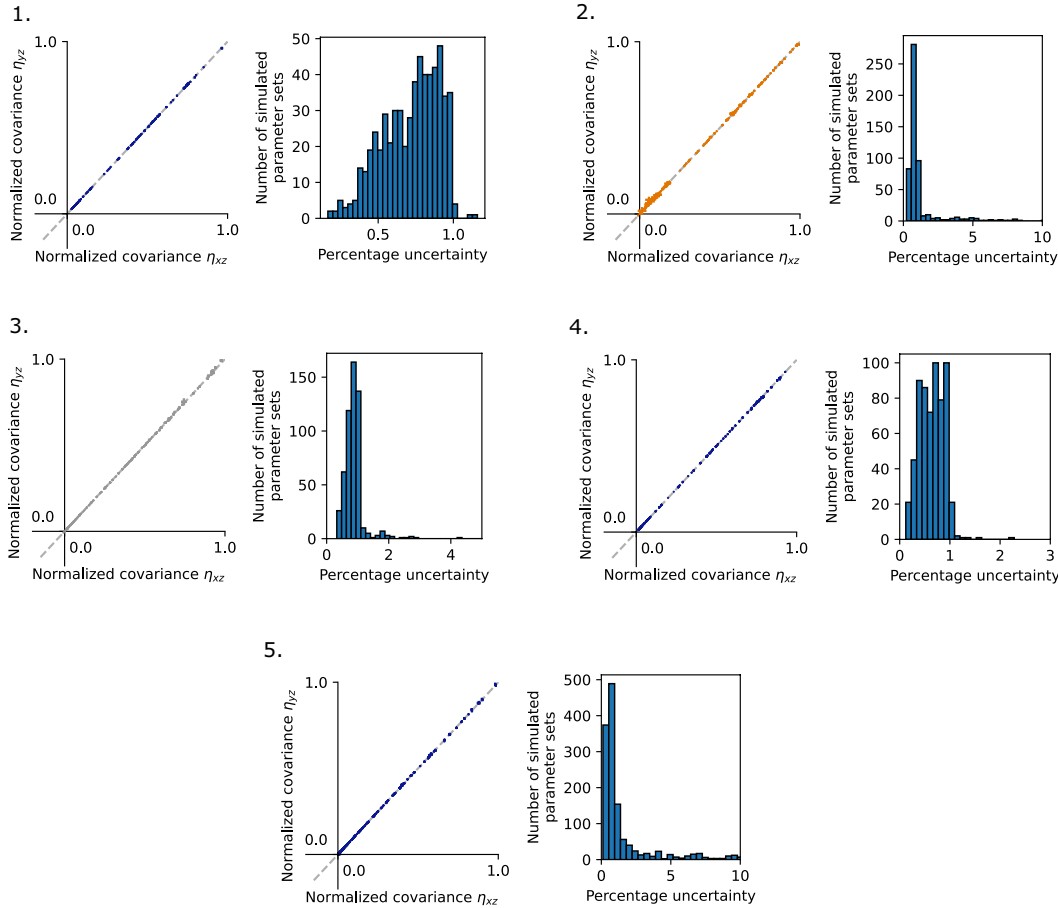

**Appendix 1—figure 3.** Percentage uncertainty distributions for individual simulated processes — part 1. Plotted are the computed normalized covariances from the simulated concentrations of growing and dividing cells in processes 1–5 (as listed in *Appendix 1—table 1*). Each number in the top left corners corresponds to the process index listed in *Appendix 1—table 1*. For each process, a number of random parameters were chosen and simulated as described in the Materials and methods section of the main text. On the left of each subplot are the normalized covariances for each of the parameter sets simulated in the respective process. Error bars are present but are almost always too small to be visible. On the right of each subplot is the distribution of percentage uncertainty, defined as $\sigma_{xz}/|\eta_{xz}| \times 100\%$ and $\sigma_{yz}/|\eta_{yz}| \times 100\%$, where $\sigma_{xz}$ is the estimated 95% confidence interval for $\eta_{xz}$. Most of the percentage errors fall below 5%. However, occasionally a parameter set is chosen that leads to either rare birth and death events in the stochastic process, which leads to low sampling, or the particular parameters lead to near zero normalized covariances, which can lead to large percentage errors. Colors of dots and error bars in the left plots correspond to the same colors plotted in the combined figure of *Figure 2*.

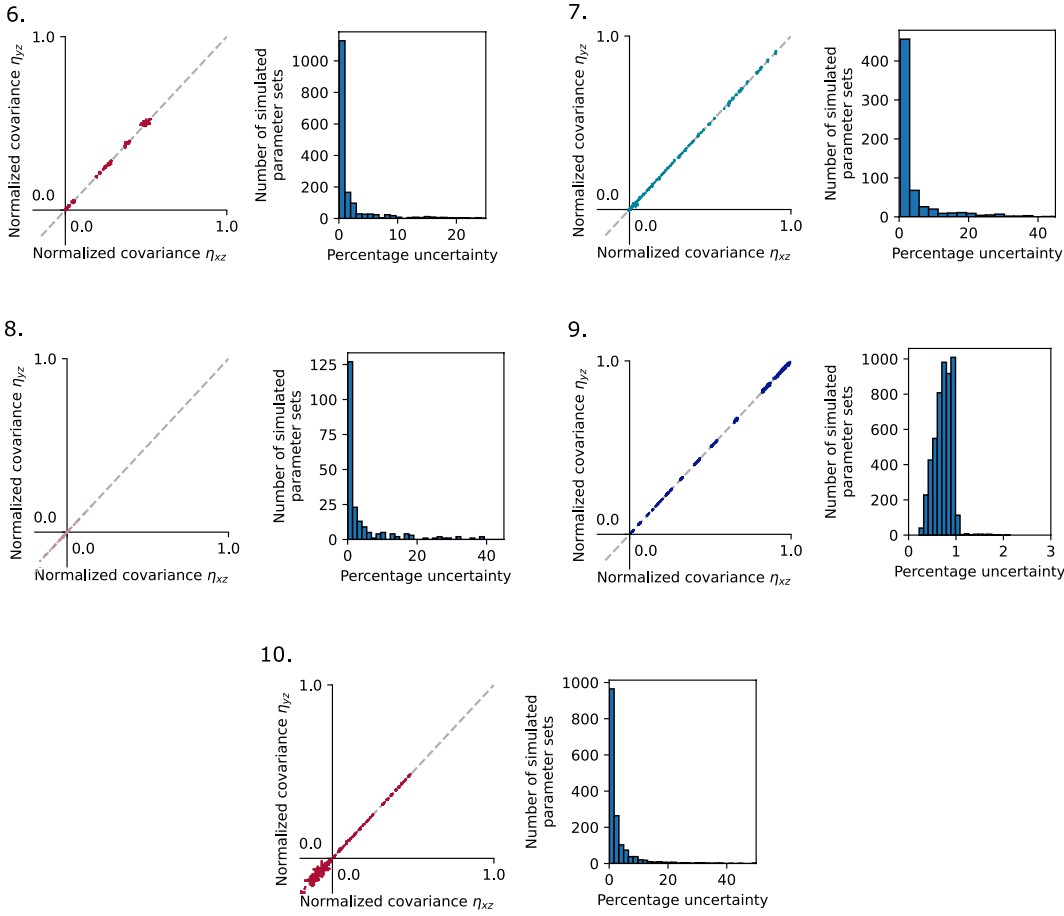

**Appendix 1—figure 4.** Percentage uncertainty distributions for individual simulated processes — part 2. Same as *Appendix 1—figure 3* but for processes 6 to 10 (as listed in *Appendix 1—table 1*).

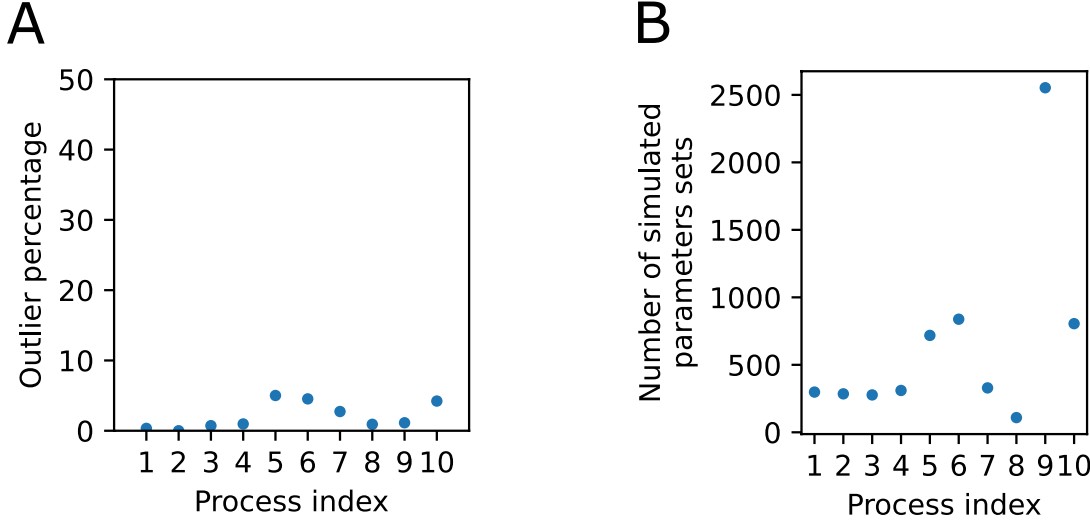

**Appendix 1—figure 5.** Percentage of simulations that deviate from the invariant of *Equation 2*. (**A**) Plotted is the percentage of simulated parameter sets for each simulated process (as listed in *Appendix 1—table 1*) that did not produce normalized covariances that satisfied *Equation 2*. Specifically, the 95% confidence interval for the ratio $\eta_{xz}/\eta_{yz}$ in these 'outlier' simulations did not capture the predicted value of 1. It is expected that not all simulations will agree with *Equation 2* due to sampling errors. The largest outlier percentage is for process 5, which has 5.01%
*Appendix 1—figure 5 continued on next page*

*Appendix 1—figure 5 continued*

of simulations that did not satisfy *Equation 2*. As described in the Materials and Methods section of the main text, all outlier systems were simulated again with additional sampling which resulted in agreement with *Equation 2*. (**B**) Plotted is the number of parameter sets simulated for each process. Some processes took less time to simulate and, as a result, resulted in a larger number of parameter sets. All processes were simulated for at least 100 parameter sets.

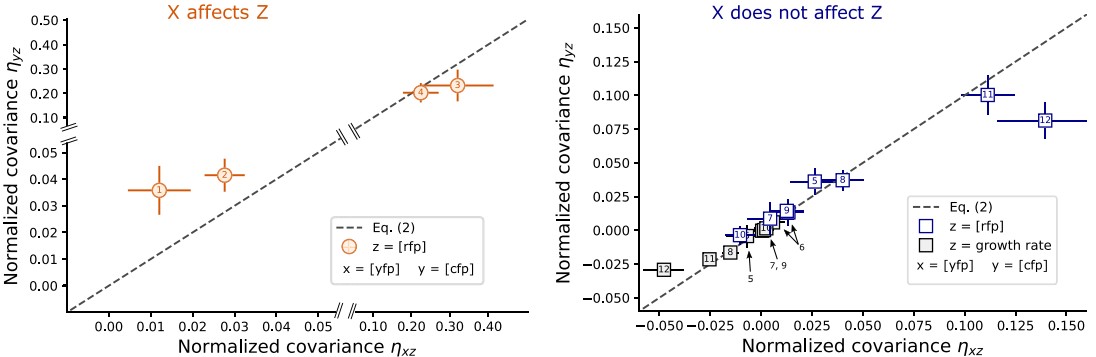

**Appendix 1—figure 6.** Estimating uncertainty with a second method does not change the results of *Figure 3D,E*. Shown are the measured normalized covariances for the synthetic positive control circuits (left) and negative control circuits (right), where error bars are computed by splitting the ensemble of single-cell traces into 7-10 disjoint sets. Final normalized covariances are taken as the averages of the sets, with 95% confidence intervals given by two times the standard error of the mean (plotted error bars). Data corrections such as non-even illumination correction and autofluorescence removal were done separately on each set in order to estimate the error of these corrections. Autofluorescence and imaging media fluorescence were corrected by analyzing cells that lost the synthetic plasmid due to random partitioning at cell division, as described in Materials and methods. If 10 or more cells from an experiment lost the plasmid, then the single-cell traces were randomly split into 10 disjoint sets, both with and without the plasmid. Otherwise, the number of disjoint sets was set as the number of cells that lost the plasmid (a minimum of 7), so that each set would contain at least one cell that lost the plasmid to estimate the autofluorescence. This is described in the Materials and methods section of the main text, as well as in *Appendix 1—figure 16* and Section 17.

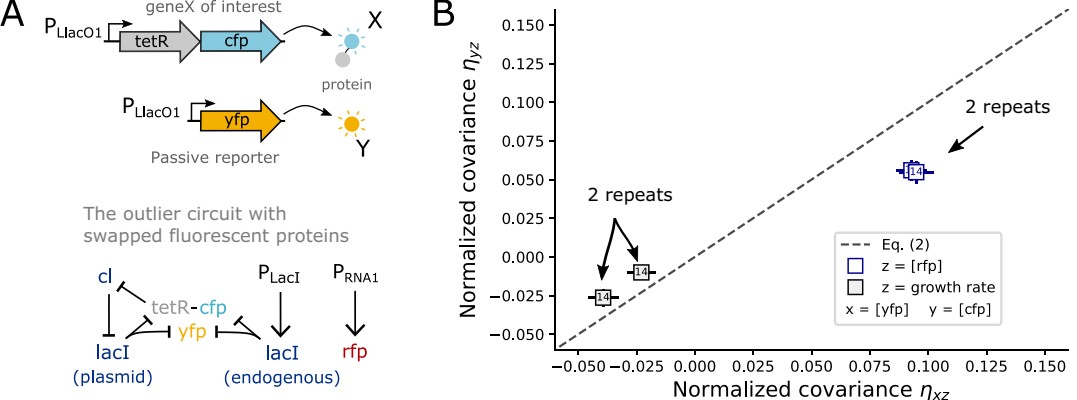

**Appendix 1—figure 7.** The outlier circuit does not satisfy *Equation 2* after switching the fluorescent proteins. (**A**) To rule out the possibility that the outlier in *Figure 3E* was caused by a systematic error from an asymmetry between measurements of the CFP and YFP fluorescence signals, we rebuilt the outlier circuit, but with *cfp* fused to *tetR* and *yfp* set as the transcriptional passive reporter. If the outlier was caused by such a systematic error, then the normalized covariance measurements would switch to the other side of the dashed line. (**B**) The normalized covariances still do not satisfy *Equation 2*, and they do not switch to the other side of the dashed line. Two repeats of the experiment are shown. Note that the numerical values for the normalized covariances are different than the outlier circuit in *Figure 3E*. This could be indicative that the CFP protein used changes the TetR function when used as a fusion.

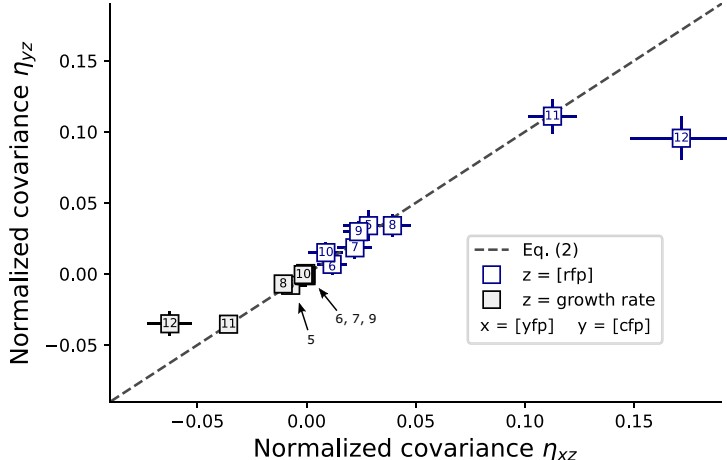

**Appendix 1—figure 8.** Using a different cell segmentation pipeline does not alter the main result of *Figure 3E* . For the results shown in *Figure 3* we used the deep-learning cell segmentation pipeline DeLTA (*O'Connor et al., 2022*), see Materials and Methods. We used the DeLTA pipeline to segment cells with the bright field channel because the three fluorescence channels (CFP, YFP, and RFP) were not always bright enough in the positive control strains to use as a segmentation marker. However, in all the negative control strains, RFP is under the control of the strong pRNA1 promoter which has been used as a segmentation marker previously (*Potvin-Trottier et al., 2016*; *Lord et al., 2019*). We thus applied the same segmentation pipeline used in *Potvin-Trottier et al., 2016*; *Lord et al., 2019* on the RFP channel of the negative control strains. The numerical values of the normalized covariances change slightly, but their relation to *Equation 2* remains intact. This second segmentation pipeline determines the edges of the cells using a thresholding algorithm, and it estimates background fluorescence by taking the median of the median of each image; see *Lord et al., 2019* for details. The autofluorescence of strains 6, 7, 8, 9, and 10 was not removed because these strains express RFP from the plasmid, so cells that lost the plasmid (which we use to estimate autofluorescence, see Materials and Methods) cannot be tracked with this pipeline.

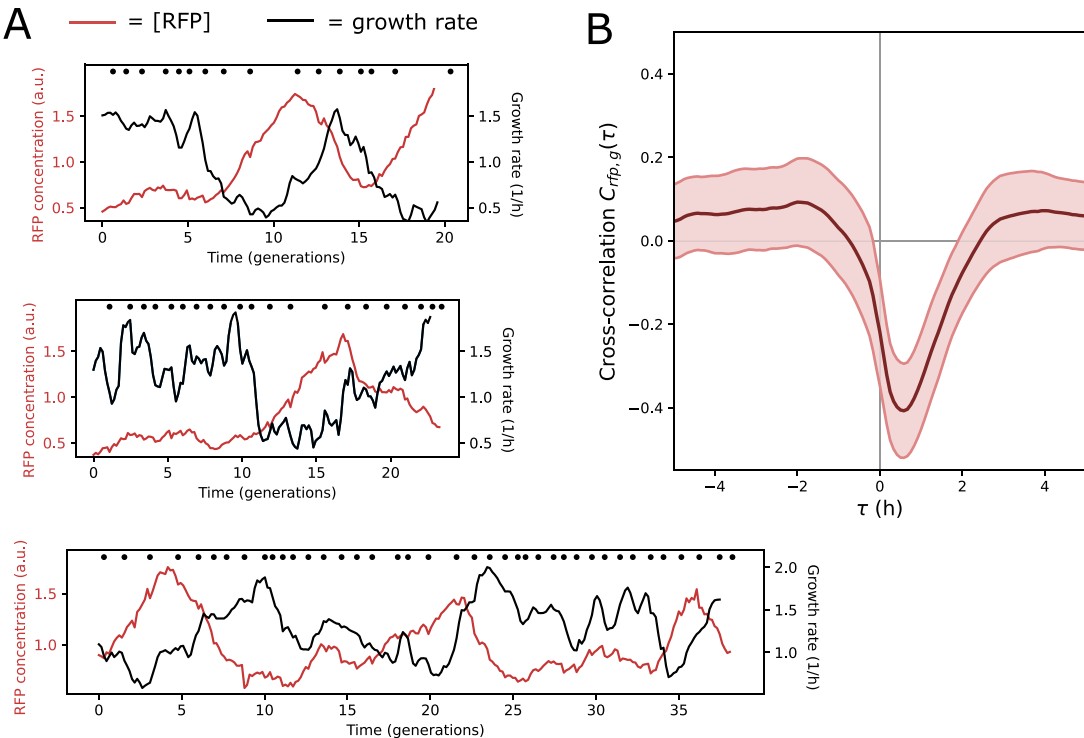

**Appendix 1—figure 9.** In the outlier circuit, large RFP fluctuations are often negatively correlated with growth rate fluctuations. We observed many instances in which RFP concentration was negatively correlated with fluctuations in the cellular growth rate. Shown are three sample single-cell traces. The top two are from the outlier circuit, whereas the bottom trace is from the outlier circuit with swapped fluorescent proteins (see **Appendix 1—figure 7A**). Dots correspond to times at which a cell division has been detected. (**B**) To quantify this trend over all lineages, we used the cross-correlation function $C_{\mathrm{rfp,g}}(\tau)$ between the RFP concentration and growth rate (**g**) in the outlier circuit. We observe a negative correlation time shifted by approximately 1 hr. $C_{\mathrm{rfp,g}}(\tau)$ quantifies how the RFP concentration is correlated with growth rate when the growth rate signal is shifted by time $\tau$ relative to the RFP signal. $C_{\mathrm{rfp,g}}(\tau)$ was computed using the python scipy function correlate on each single-cell trajectories while normalizing by $\sqrt{C_{\mathrm{rfp,rfp}}(0)C_{\mathrm{g,g}}(0)}$. The dark line corresponds to the average at time $\tau$ over all trajectories, with the width of the shaded region given by the standard deviation about the mean.

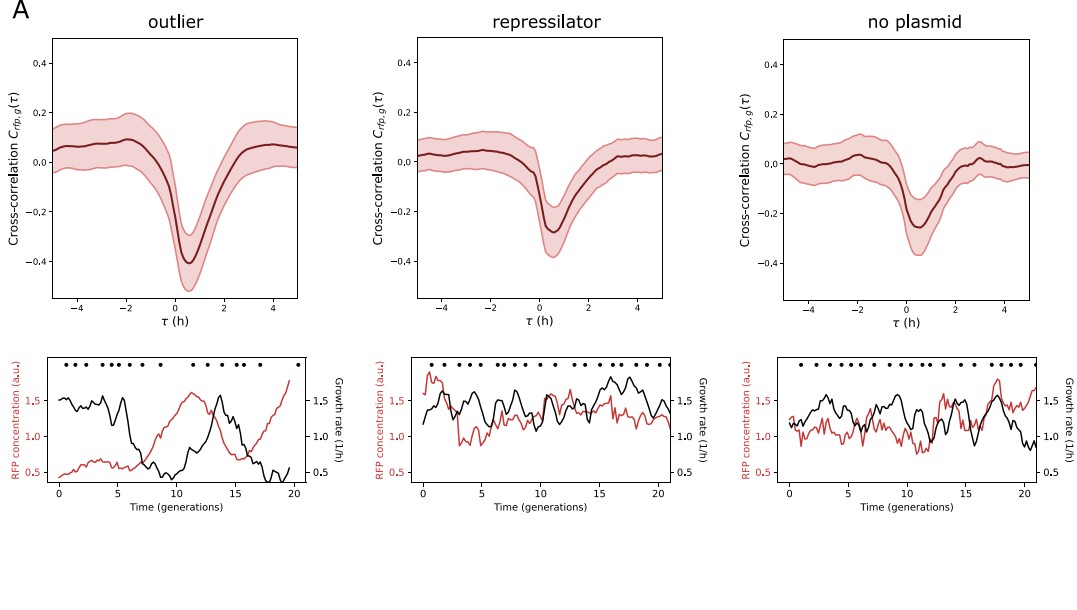

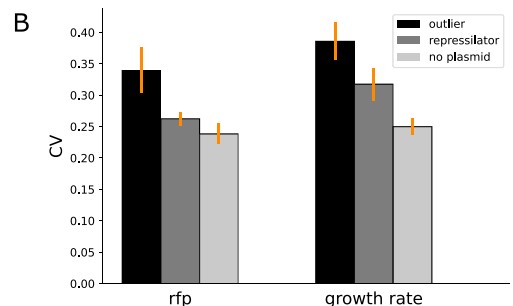

**Appendix 1—figure 10.** The outlier circuit shows more variability in RFP concentrations and growth rate measurements as compared to other strains that express RFP from the chromosome with the pRNA1 promoter. (**A**) The 'regular' repressilator circuit without endogenous LacI also exhibits negative correlation between RFP and growth rate, albeit with smaller magnitude. Shown are the cross correlations (defined in **Appendix 1—figure 9B**) for the outlier circuit (left), the strain with the regular repressilator (outlier circuit with *lacI* deletion, middle), and the background strain without a plasmid but with RFP still expressed (right). In all cases, RFP is expressed from the chromosome under the control of the same pRNA1 promoter. The outlier circuit exhibits slightly more negative correlation. Below each cross-correlation plot is a sample time-trace for a cell with the respective circuit. Dots correspond to times at which a cell division has been detected. (**B**) The 'regular' repressilator circuit without endogenous LacI does not exhibit the same degree of growth rate fluctuations and does not exhibit as pronounced RFP pulses. The size of the fluctuations of the growth rate and RFP concentration was measured as the coefficient of variability (CV), defined as $\mathrm{CV}_z = \sqrt{Var(z)}/\langle z \rangle$. The CVs were measured for the outlier circuit, the strain with the regular repressilator (outlier circuit with *lacI* deletion), and the background strain without a plasmid but with RFP still expressed. The outlier circuit exhibits a larger CV in both cases, suggesting that the outlier circuit causes the increased growth rate fluctuations. Orange error bars correspond to 95% confidence intervals estimated using the same bootstrapping approach used for the normalized covariances described in the Materials and Methods of the main text.

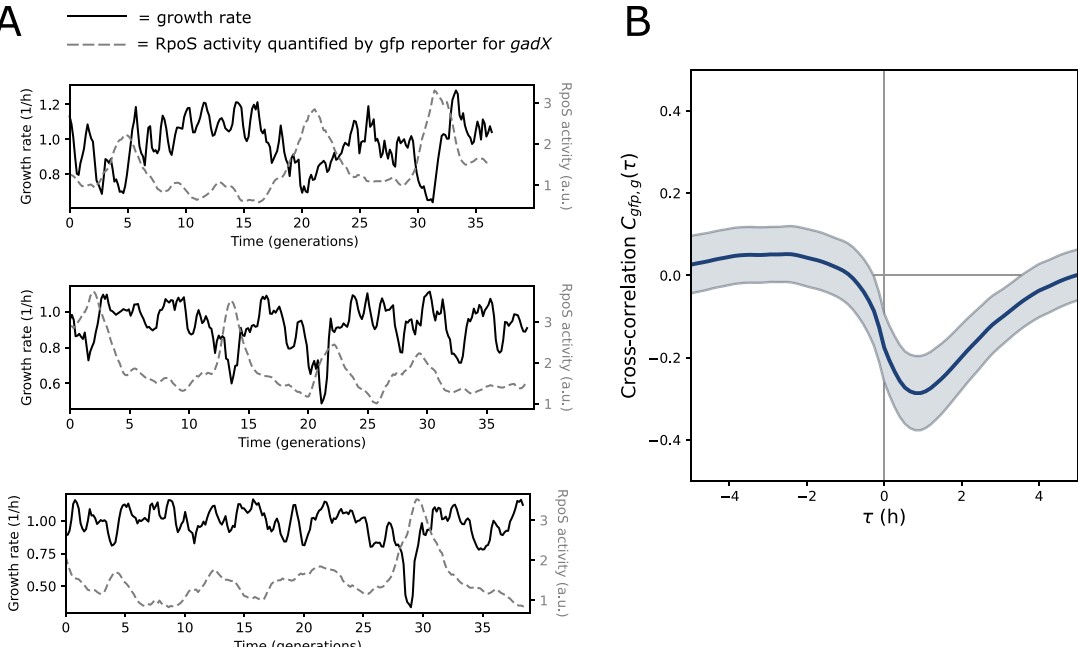

**Appendix 1—figure 11.** RpoS activity often displayed pulses that are negatively correlated with growth rate. We observed many instances in which RpoS activity, quantified by a transcriptional GFP reporter under the control of the *gadX* promoter, was negatively correlated with fluctuations in the cellular growth rate. Shown are three sample single-cell traces. (**B**) To quantify this trend over all lineages, we used the cross-correlation function $C_{\mathrm{gfp,g}}(\tau)$ between the GFP concentration and growth rate (**g**). We observe a negative correlation time shifted by approximately 1 hr. $C_{\mathrm{gfp,g}}(\tau)$ quantifies how the GFP concentration is correlated with growth rate when the growth rate signal is shifted by time $\tau$ relative to the GFP signal. It was computed as described in the *Appendix 1—figure 9* caption.

### 4. Repressilator cascade with endogenous lacI

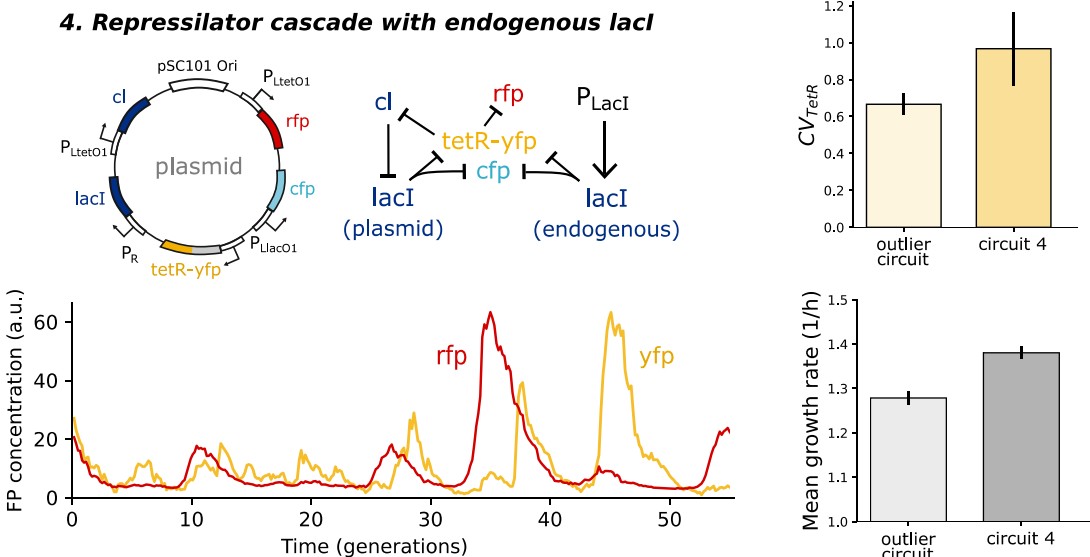

**Appendix 1—figure 12.** Removing the pRNA1 controlled *rfp* gene changes the TetR dynamics of the outlier circuit and increases the growth rate. One of the positive control circuits (4) contains the same synthetic circuit as the outlier circuit, but without the pRNA1 controlled *rfp* on the chromosome, and instead a pLtetO1 controlled *rfp* gene on the plasmid. We observed that the TetR dynamics, quantified by YFP, were different than in the outlier circuit as they displayed random large pulses. The RFP levels, in turn, also displayed random large pulses when the TetR levels were reduced. To quantify these observations over all lineages, we computed the CV for the TetR levels for both strains (top right). The strain 4 CV is larger. We also computed the average growth rate *Appendix 1—figure 12 continued on next page*

*Appendix 1—figure 12 continued*

for both strains and found that the strain 4 growth rate is approximately 9% larger. These results suggest that the pRNA1-controlled *rfp* gene affected the rest of the circuit elements, along with the cellular growth rate, which we hypothesize is from burden caused by resource competition. Error bars correspond to 95% confidence intervals using the bootstrapping approach used for the normalized covariances as discussed in the Materials and methods section.

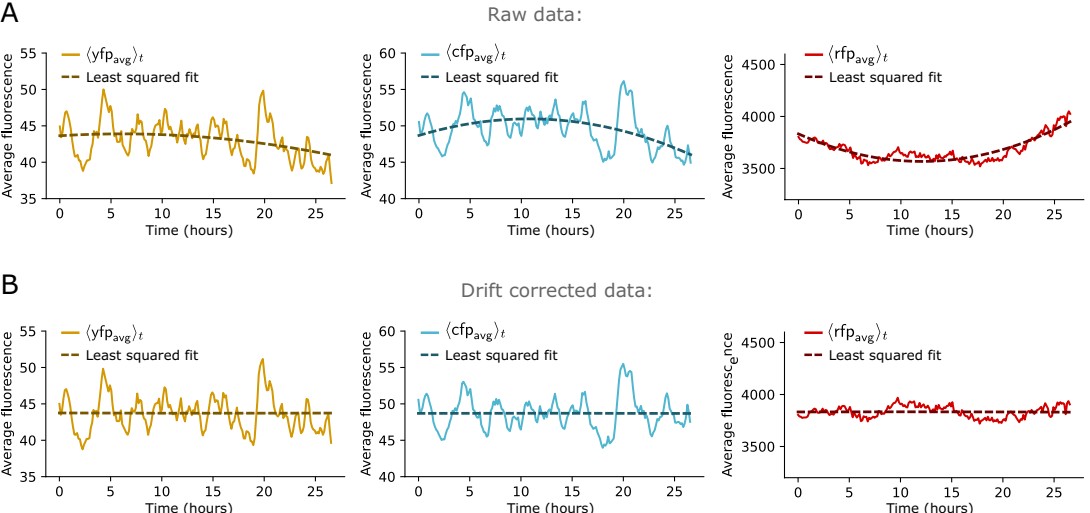

**Appendix 1—figure 13.** Correcting for temporal drift of the focal points. (**A**) Individual average fluorescence measurements (taken over cell area) are binned according to their timeframe. The average of the measurements at each timeframe is plotted. A second-degree polynomial is fitted to the resulting time-traces using a least squares fit. Shown is data from the strain 6 in **Appendix 1—figure 18**. The YFP and CFP time-traces fluctuate more than the RFP time-trace due to the large pulses that they display from the Repressilator. (**B**) Temporal drift is corrected by multiplying all of the individual average fluorescence measurements with the reciprocal of the fits. Plotted is the average time-traces taken from the corrected data, along with new fits that display constant trends.

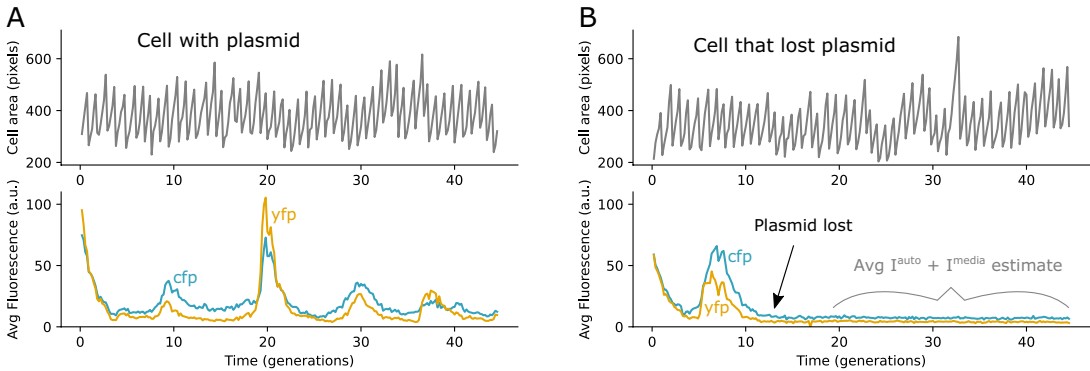

**Appendix 1—figure 14.** Estimating autofluorescence and media fluorescence with cells that lost plasmid. (**A**) A single-cell time trace from a mother machine experiment in which the plasmid is kept for the entire duration of the experiment. On the top is the cell area over time displaying cellular growth and division, and on the bottom are the average YFP and CFP signals taken over the cell area. Note that autofluorescence and media fluorescence have not been corrected for here. (**B**) A single-cell time trace in which the plasmid containing the synthetic circuit is lost. The signals decay to constant values just above zero. The cell area continues to grow and divide, indicating that the decay is not the result of segmentation error or loss of cell.

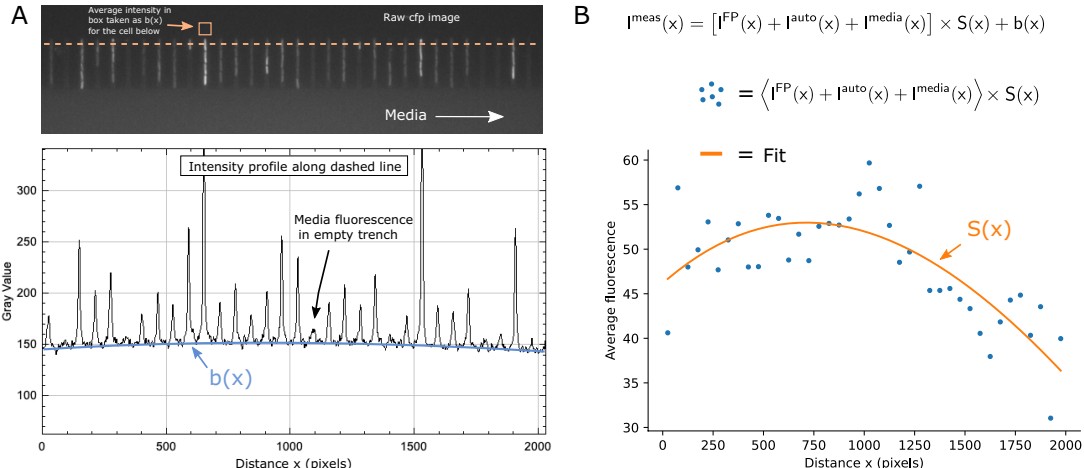

**Appendix 1—figure 15.** Correcting for background and uneven illumination. (**A**) Plotted is the fluorescence intensity profile across the dashed orange line shown in a single image of the CFP channel. The background $b(x)$ is traced by the light blue solid line in the bottom plot, to illustrate the slight $x$ dependence. To estimate and remove the background, in each image the average fluorescence across a 50x50 pixel box located 15 pixels above each mother cell was removed from the fluorescence measurement of each respective cell in that image. Note also that an empty trench produces a significant amount of fluorescence when compared to cell-containing trenches. This indicates that fluorescence originating from the media itself needs to be considered. This will be discussed in the next section. The fluorescence profile was created using the plot profile function in Fiji ImageJ. (**B**) With the background $b(x)$ removed, it remains to correct for $S(x)$. We bin all of the fluorescence measurement of all cells and over all time according to pixel position $x$ in bins of size 50 pixels and take the average of each bin (blue dots). A single frame has a width of 2048 pixels. This gives an estimate of $\langle I^{FP} + I^{auto} + I^{media} \rangle \times S(x)$, with deviations resulting from finite sampling. The binned averages are fit using the least squares method with a third-degree polynomial (solid orange line). The uneven illumination gain function $S(x)$ is corrected by multiplying all the fluorescence measurements with the reciprocal of the fitted polynomial. Data shown is from the strain 6 system from *Appendix 1—figure 18*.

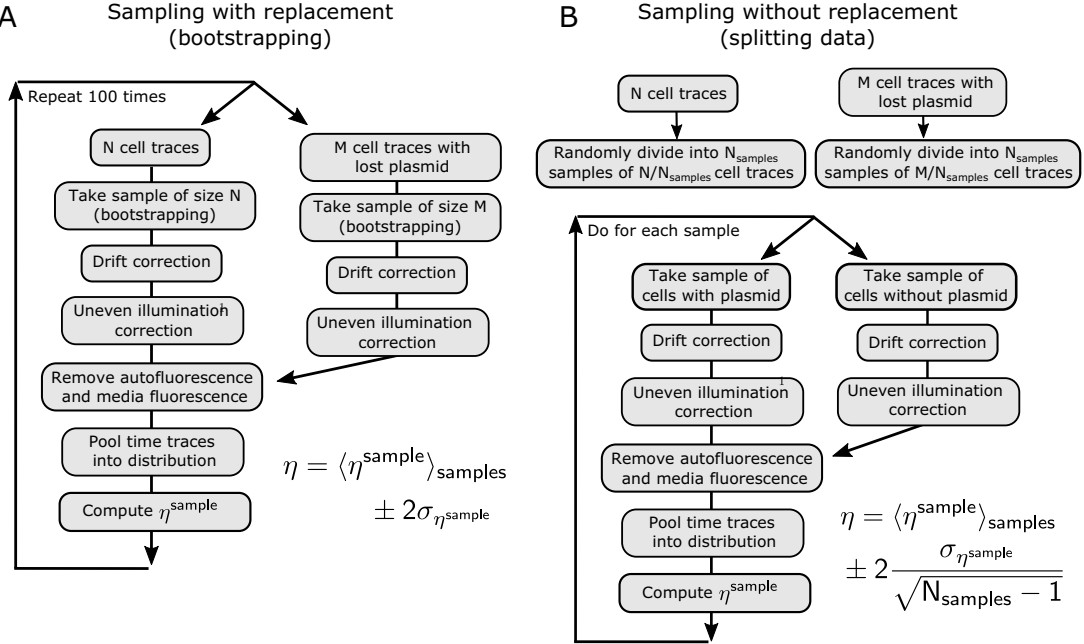

**Appendix 1—figure 16.** Estimating confidence intervals for normalized covariance measurements using two methods. (**A**) Imaging error corrections and the normalized covariance are computed 100 times for 100 bootstrap samples of the experimental data. The final normalized covariance is given as the average over the samples, with *Appendix 1—figure 16 continued on next page*

*Appendix 1—figure 16 continued*

confidence intervals given by two times the standard deviation. (**B**) The data is divided into $N_{samples} = \min(M, 10)$ disjoint sample sets, where $M$ is the number of cells that lost the plasmid. Most experiments produced $M > 10$, but a few produced $M < 10$ (with a minimum of 7). Imaging error corrections and the normalized covariance are computed for each sample. The final normalized covariance is given by the average over the $N_{samples}$ samples, with confidence intervals given by two times the standard error of the mean.

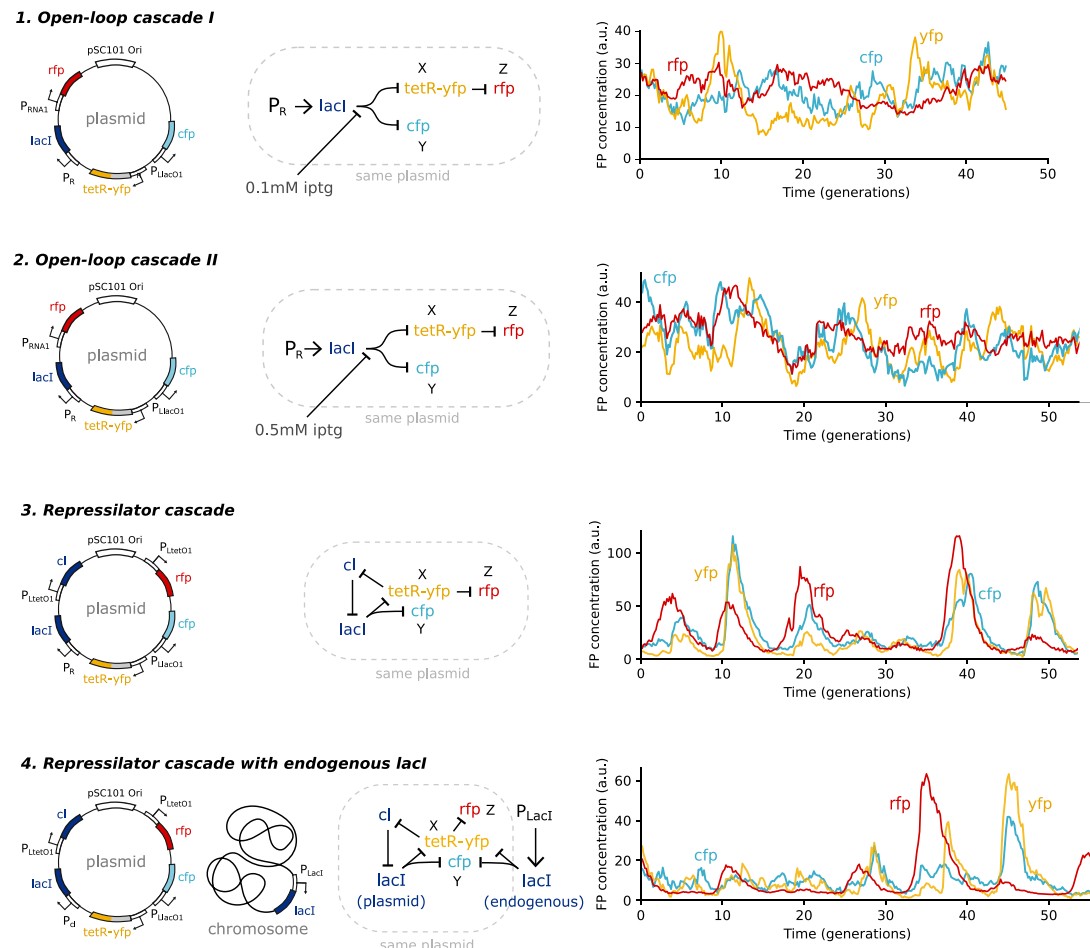

**Appendix 1—figure 17.** Positive control synthetic circuits with sample single-cell time traces. Plotted fluorescence units are not consistent from sample to sample.

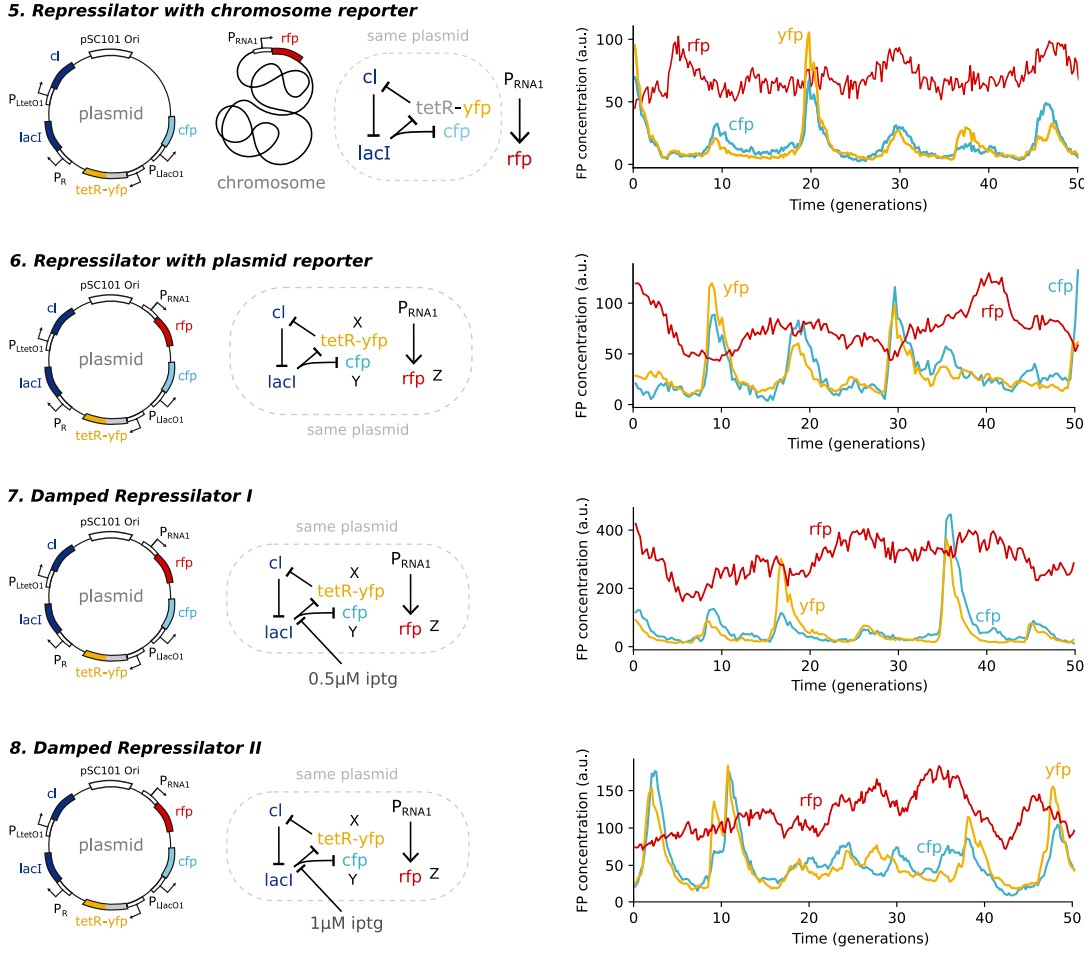

**Appendix 1—figure 18.** Negative control synthetic circuits with sample single-cell time traces – part 1. Plotted fluorescence units are not consistent from sample to sample.

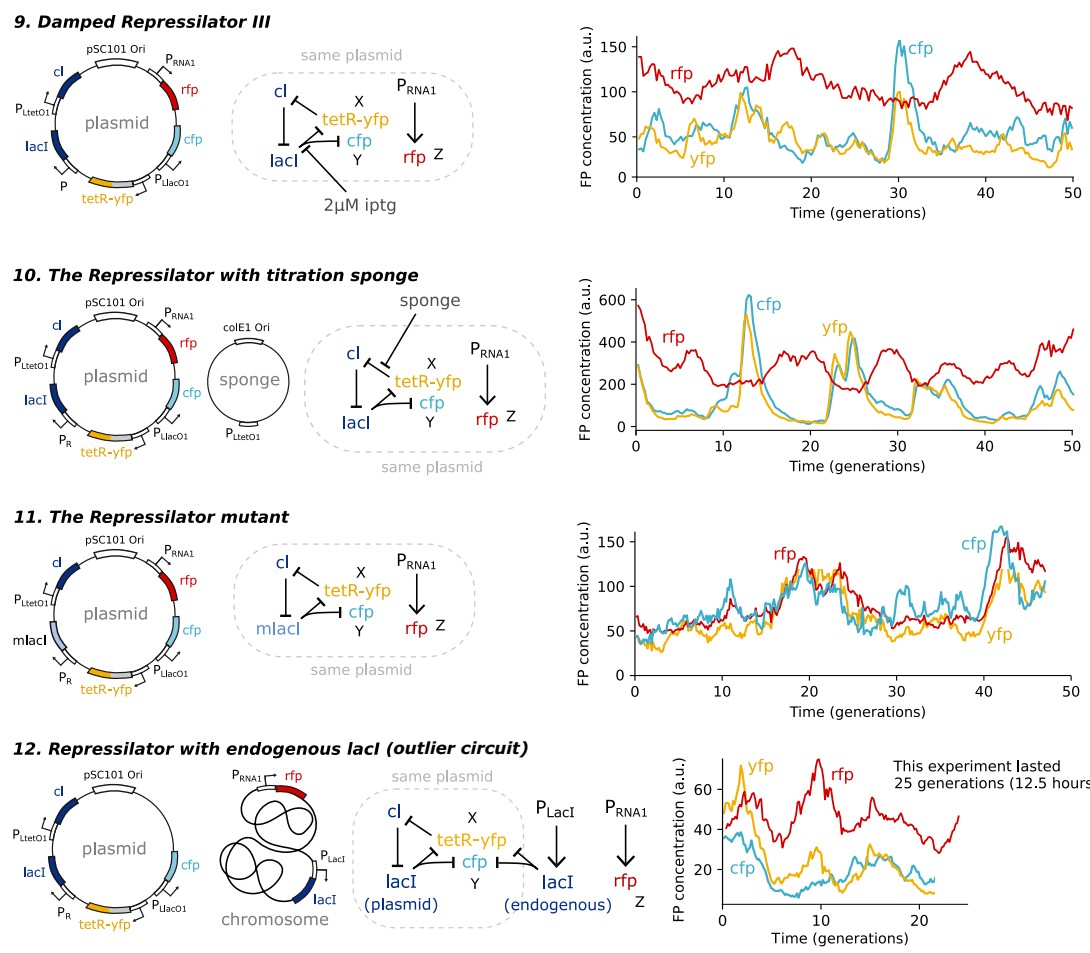

**Appendix 1—figure 19.** Negative control synthetic circuits with sample single-cell time traces – part 2. Plotted fluorescence units are not consistent from sample to sample.

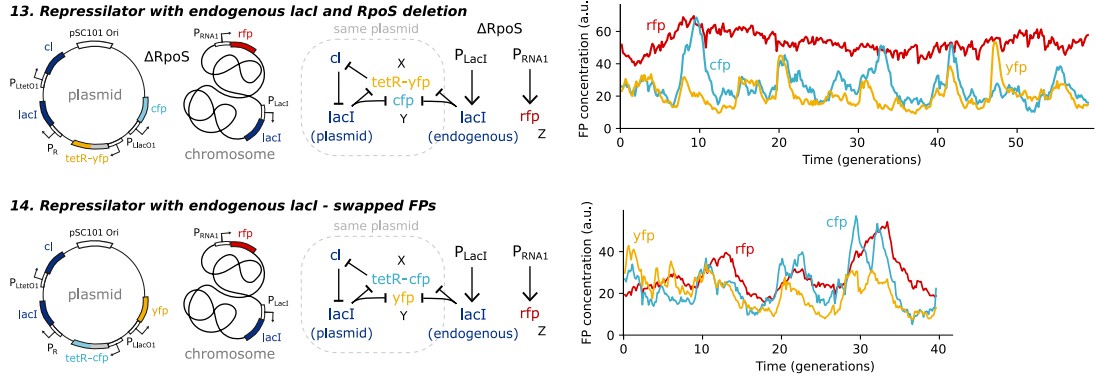

**Appendix 1—figure 20.** Other synthetic circuits.

## 2. Supplementary tables

General rates for four component system

$$x \xrightarrow{r_x^+} x+1 \qquad y \xrightarrow{r_y^+} y+1 \qquad z \xrightarrow{r_z^+} z+1 \qquad w \xrightarrow{r_w^+} w+1$$

$$x \xrightarrow{r_x^-} x-1 \qquad y \xrightarrow{r_y^-} y-1 \qquad z \xrightarrow{r_z^-} z-1 \qquad w \xrightarrow{r_w^-} w-1$$

**Appendix 1—table 1.** Simulated birth-death processes in growing and dividing cells. Additional details in Section 11. Top box shows a general four-component birth-death process. Below the box shows the respective rates for the 10 processes simulated. For each process, three different single-cell volume trajectories $V(t)$ were simulated according to the Materials and methods of the main text, and the rate parameters shown above were varied randomly multiple times. In processes 1–3, 8, and 9, $\lambda$ was picked randomly from the set {10, 100, 1000}, whereas in processes 4, 5, and 7, it was picked randomly from the set {1,10, 100}. The parameter δ is set to 0.01 throughout to avoid divisions by 0. In processes 4, 5, 7, and 9, α is respectively set to (2, 0.1, 1),(2, 0.5, 0.1), (2, 1, 0.5), (1, 2, 0.5) for all simulations with respective volume dynamics (1, 2, 3). For example, in system 4 with volume dynamics 2, α = 0.1. All other parameters, namely, $\beta, \beta_z, \beta_w, \lambda_z, \lambda_w, K, A, B, C, \omega$ were picked randomly from the set {0.1, 1, 10}. In all simulations, the average division time is set to 1. Simulations were performed using the Gillespie algorithm, with adjustments to account for time-dependent rates, as described in Section 14.

| Process index | $r_x^+$ | $r_x^-$ | $r_y^+$ | $r_y^-$ | $r_z^+$ | $r_z^-$ | $r_w^+$ | $r_w^-$ |
|---|---|---|---|---|---|---|---|---|
| 1 | $\frac{\lambda z}{K+x}$ | $\beta x$ | $\frac{\lambda z}{K+x}$ | $\beta y$ | $\lambda_z$ | $\beta_z z$ | $0$ | $0$ |
| 2 | $\frac{\lambda(x+\delta)}{\left(K+x+\left(\frac{y}{z+\delta}\right)^2\right)}$ | $\beta x$ | $\frac{\lambda(x+\delta)}{\left(K+x+\left(\frac{y}{z+\delta}\right)^2\right)}$ | $\beta y$ | $\lambda z$ | $\beta_z z$ | $0$ | $0$ |
| 3 | $\frac{\lambda w}{K+x^2}$ | $\beta x$ | $\frac{\lambda w}{K+x^2}$ | $\beta y$ | $\lambda_z w$ | $\beta_z z$ | $\lambda_w$ | $\beta_w w$ |
| 4 | $\frac{\lambda\left(\frac{z}{V(t)}\right)}{1+\left(\frac{x}{V(t)}\right)^2}$ | $\beta x$ | $\frac{\alpha\lambda\left(\frac{z}{V(t)}\right)}{1+\left(\frac{x}{V(t)}\right)^2}$ | $\beta y$ | $\lambda_z$ | $\beta_z z$ | $0$ | $0$ |
| 5 | $\frac{\alpha\lambda V(t)}{1+\left(\frac{x/V(t)}{\delta+z/V(t)}\right)^3}$ | $\beta x$ | $\frac{\lambda V(t)}{1+\left(\frac{x/V(t)}{\delta+z/V(t)}\right)^3}$ | $\beta y$ | $\lambda_z V(t)$ | $\beta_z z$ | $0$ | $0$ |
| 6 | $A\left(\sin(\omega t)+1\right)V(t)$ | $\beta x$ | $B\left(\sin(\omega t)+1\right)V(t)$ | $\beta y$ | $C\left(\sin(\omega t)+1\right)V(t)$ | $\beta_z z$ | $0$ | $0$ |
| 7 | $\frac{\alpha\left(\frac{y}{V(t)}+\delta\right)}{\frac{y}{V(t)}+\delta+\left(\frac{x/V(t)}{\delta+w/V(t)}\right)^2}$ | $\beta x$ | $\frac{\left(\frac{y}{V(t)}+\delta\right)}{\frac{y}{V(t)}+\delta+\left(\frac{x/V(t)}{\delta+w/V(t)}\right)^2}$ | $\beta y$ | $\lambda_z w/V(t)$ | $\beta_z z^2$ | $\lambda_w V(t)$ | $\beta_w w$ |
| 8 | $\frac{\lambda}{1+\left(\frac{x/V(t)}{\delta+w}\right)^3}$ | $\beta x$ | $\frac{\lambda}{1+\left(\frac{x/V(t)}{\delta+w}\right)^3}$ | $\beta y$ | $\frac{\lambda_z}{1+\left(\frac{w/V(t)}{k}\right)^3}$ | $\beta_z z$ | $\frac{\lambda_z}{1+\left(\frac{z/V(t)}{k}\right)^3}$ | $\beta_w w$ |
| 9 | $\frac{\alpha\lambda z}{K+x}$ | $xz$ | $\frac{\lambda z}{K+x}$ | $yz$ | $\lambda_z$ | $\beta_z z$ | $0$ | $0$ |
| 10 | $A\left(\sin(\omega t)+1\right)V(t)$ | $\beta xz$ | $B\left(\sin(\omega t)+1\right)V(t)$ | $\beta yz$ | $C\left(\sin(\omega t)+1\right)V(t)$ | $\beta_z z$ | $0$ | $0$ |

**Appendix 1—table 2.** Strains and plasmids.
See *Appendix 1—figures 17–20* for circuits.

| Strain | Details |
|---|---|
| EJS1 | Synthetic circuit 5. Background strain is MG1655 *E. coli* attB::lacYA177C, ΔaraCBAD, ΔlacIZYA, ΔaraE, ΔaraFGH, ΔrhaSRT, ΔrhaBADM, attn7::mCherry-mKate2-hybrid. Contains pEJS1. |
| EJS2 | Synthetic circuit 12. Background strain is MG1655 *E. coli* with glmS::PRNA1-mCherry-mKate2- hybrid and motility deletion ΔmotA. Contains pEJS1. |
| EJS3 | Synthetic circuits 1 and 2. Background strain is MG1655 *E. coli* with attB::lacYA177C, ΔaraCBAD, ΔlacIZYA, ΔaraE, ΔaraFGH, ΔrhaSRT, and ΔrhaBADM. Contains plasmid pEJS3. |
| EJS4 | Synthetic circuit 3. Background strain is MG1655 *E. coli* with attB::lacYA177C, ΔaraCBAD, ΔlacIZYA, ΔaraE, ΔaraFGH, ΔrhaSRT, and ΔrhaBADM. Contains plasmid pEJS4. |

*Appendix 1—table 2 Continued on next page*

*Appendix 1—table 2 Continued*

| Strain | Details |
| --- | --- |
| EJS6 | Synthetic circuit 4. Background strain is MG1655 *E. coli* with motility deletion ΔmotA. Contains plasmid pEJS4. |
| EJS7 | Synthetic circuit 11. Background strain is MG1655 *E. coli* with attB::lacYA177C, ΔaraCBAD, ΔlacIZYA, ΔaraE, ΔaraFGH, ΔrhaSRT, and ΔrhaBADM. Contains plasmid pEJS5m. |
| EJS9 | Synthetic circuit 14. Background strain is MG1655 *E. coli* with glmS::PRNA1-mCherry-mKate2- hybrid and motility deletion ΔmotA. Contains plasmid pEJS10. |
| PA12 PA52 | Strain with no plasmid in *Appendix 1—figure 10*. Background strain is MG1655 *E. coli* attB::lacYA177C, ΔaraCBAD, ΔlacIZYA, ΔaraE, ΔaraFGH, ΔrhaSRT, ΔrhaBADM, attn7::mCherry-mKate2-hybrid. Synthetic circuits 6, 7, 8, and 9. Background strain is MG1655 *E. coli* with attB::lacYA177C, ΔaraCBAD, ΔlacIZYA, ΔaraE, ΔaraFGH, ΔrhaSRT, and ΔrhaBADM. Contains plasmid pEJS5. |
| PA53 | Synthetic circuit 10. Background strain is MG1655 *E. coli* with attB::lacYA177C, ΔaraCBAD, ΔlacIZYA, ΔaraE, ΔaraFGH, ΔrhaSRT, and ΔrhaBADM. Contains plasmids pEJS5 and pLPT41. |
| PA64 | Used in *Figure 4D* Background strain is MG1655 *E. coli* with glmS::PRNA1-mCherry-mKate2-hybrid and motility deletion ΔmotA. Contains pSC101 plasmid with Kanamycin resistance and gfpmut2 reporter gene with gadX promoter taken from *Zaslaver et al., 2006*. |
| PA66 | Synthetic circuit 13. Background strain is MG1655 *E. coli* with glmS::PRNA1-mCherry-mKate2- hybrid, motility deletion ΔmotA, and DRpos::FRT Kan frt. Contains plasmid pEJS1. |

| Plasmid | Details |
| --- | --- |
| pEJS1 | Used in circuits 5, 12, and 13. Repressilator with no degradation tags, with *tetR* replaced with *tetR-mVenusNB fusion*, along with an integrated *SCFP3A* reporter. pSC101 origin. Ampicillin resistance. |
| pEJS3 | Used in circuit 1 and 2. Repressilator with no degradation tags with *cI* replaced with *mCherry-mKate2-hybrid*, *tetR* replaced with *tetR-mVenusNB fusion*, and an integrated *SCFP3A* reporter. pSC101 origin. Ampicillin resistance. |
| pEJS4 | Used in circuit 3 and 4. Repressilator with no degradation tags, with *tetR* replaced with *tetR-mVenusNB fusion*, and integrated *SCFP3A* and *mCherry-mKate2-hybrid* reporters. pSC101 origin. Ampicillin resistance. |
| pEJS5 | Used in circuits 6, 7, 8, and 9. Repressilator with no degradation tags, with *tetR* replaced with *tetR-mVenusNB fusion*, and integrated *SCFP3A* and *mCherry-mKate2-hybrid* reporters. pSC101 origin. Ampicillin resistance. |
| pEJS5m | Used in circuit 11. Same as pEJS5, with a single missense point mutation on *lacI* that results in no oscillations forming (see 11 in *Appendix 1—figure 19*). LacI function must be affected by the mutation. pSC101 origin. Ampicillin resistance. |
| pEJS10 | Used in circuit 14. Same as pEJS1 but *SCFP3A* forms the fusion with *tetR* while *mVenusNB* is integrated as a transcriptional reporter. pSC101 origin. Ampicillin resistance. |
| pLPT41 | Used in circuit 10. ColE1 origin plasmid with *PLtetO*1 promoter, taken from *Potvin-Trottier et al., 2016*. Acts as a "titration sponge" for small numbers of TetR levels using repressor-binding sites, see *Potvin-Trottier et al., 2016*. Kanamycin resistance. |

## 3. Cloning details

### A. Bacterial strains and growth conditions

A complete list of the synthetic circuits can be found in *Appendix 1—figures 17–20* . A complete list of strains is in *Appendix 1—table 2.*

The MG1655 *E. coli* strain used (SKA360 from *Aoki et al., 2019*) for the microfluidic experiments with circuits 1, 2, 3, 6, 7, 8, 9, 10, and 11 had gene attB::*lacYA177C* and the following deletions: Δ*araCBAD*, Δ*lacIZYA*, Δ*araE*, Δ*araFGH*, Δ*rhaSRT*, and Δ*rhaBADM*. The strain used for circuit 5 was identical with an added *attn7::PRNA1-mCherry-mKate2-hybrid* chromosomal gene. This last strain was also used in *Appendix 1—figure 10* for the strain with no plasmid. The MG1655 *E. coli* strain used (NDL162 from *Luro et al., 2020*) for the microfluidic experiments with circuits 12, 14, and 15 had chromosomal gene *glmS::PRNA1-mCherry-mKate2-hybrid* and a motility deletion Δ*motA*. The strain used for circuit 13 was identical with the additional Δ*rpoS::FRT-Kan-frt*. The strain used for circuit 4 was identical to NDL162 but without the chromosomal gene *glmS::PRNA1-mCherry-mKate2-hybrid*.

The standard heat shock procedure was used to transform plasmid DNA into chemically competent *E. coli*. During cloning procedures, *E. coli* strains were grown in LB medium (1% tryptone, 0.5% yeast extract, 1NaCl) with aeration at 250 rpm or LB agar plates at 37°C with appropriate antibiotics at standard concentrations: Ampicillin 100 µg/ml (Amp) and Kanamycin 50 µg/ml (Kan) (antibiotics and LB medium sourced from Fisher Scientific).

Prior to a mother machine experiment, cells were grown overnight in LB medium with appropriate antibiotics from glycerol stocks. At ~3-4 hr prior to the experiment start time, the overnight cultures were diluted 1:100 in imaging media consisting of M9 salts, 10% (v/v) LB, 0.2% (w/v) glucose, 2 mM MgSO4, 0.1 mM CaCl2, 1.5 µM thiamine hydrochloride and 0.85 g/L Pluronic F-108 (Sigma Aldrich, surfactant to prevent cells sticking to the surface of a microfluidic device). For experiments with circuits 2, 7, 8, and 9 in which IPTG is used, the respective concentrations of IPTG shown in *Appendix 1—figures 17–19* are added to the imaging media given to the cells at this point and throughout the remainder of the experiment.

## B. Plasmid construction

A complete list of plasmids can be found in *Appendix 1—table 2*. Snapgene files are available on GitHub. Primers and PCR fragments are labeled in the Snapgene files.

Plasmids on the left side of *Appendix 1—figures 17–20* were constructed using Gibson assembly (NEBuilder HiFi DNA Assembly Master Mix, NEB, MA, USA). Reactions were performed according to the manufacturers protocols, with Gibson assembly reactions incubated for 15 min at 50°C.

Primers were ordered from Thermo Fisher Scientific and PCR were performed with Accuprime Pfx (Thermo Fisher) or Phusion polymerase (New England BioLabs) according to manufacturers protocols. All plasmids were verified by DNA sequencing, either Sanger sequencing by Quintara Biosciences (MA, USA) or whole plasmid nanopore sequencing with Plasmidsaurus certified by Oxford Nanopore Technologies.

Various fragments from plasmids built in *Potvin-Trottier et al., 2016* were amplified by PCR with the ordered primers. Most notably is pLPT119, which formed the backbone for all the plasmid constructs. It consists of the repressilator with no degradation tags on the *lacI*, *cI*, and *tetR* genes (*Potvin-Trottier et al., 2016*).

To assemble pEJS1 and pEJS10 by phusion reaction, two fragments were amplified from pLPT119 and made up the three repressors from the repressilator, SCFP3A with PLlacO-1 promoter was amplified from pLPT107, and mVenus NB was amplified from pTP85.

To assemble pEJS3 and pEJS4, SCFP3A with PLlacO-1 promoter was amplified from pLPT107, mCherry-mKate2-hybrid with PLtetO-1 promoter was amplified from pLPT27, mVenus fusion from pEJ1, and repressilator fragments were amplified from pLPT119.

To assemble pEJS5, SCFP3A with PLlacO-1 promoter was amplified from pLPT107, mCherry-mKate2-hybrid with PRNA1 promoter was amplified from pLPT26, mVenus fusion from pEJS1, and repressilator fragments were amplified from pLPT119.

For the experiment shown in *Figure 4D*, we used a pSC101 plasmid with Kanamycin resistance and a *gfpmut2* reporter gene with *gadX* promoter taken from *Zaslaver et al., 2006* (ordered from the *E. coli* Promoter Collection from Horizon Discovery).

The *tetR-yfp* fusion was made by removing the stop codon from the *tetR* sequence and replacing it with a short linker sequence followed by the mVenus NB sequence. The first nucleotide from the mVenus NB start codon is not added. The linker sequence is GGGTCTGACATCCTCGAGT.

All repressilator gene sequences and fluorescent protein sequences are followed by transcription terminator T1 from the *E. coli rrnB* gene.

# 4. Class of systems

We consider an arbitrarily complex time-continuous Markov chain of birth-death processes with unspecified reaction rates, corresponding to a biochemical reaction network in a cell. The state of the system at a given moment in time is given by the numbers $\mathbf{z} = \{z_k\}$, consisting of the abundances of all the chemical species in the cell, and all stochastic processes that influence their chemical reaction rates. The dynamics of the system varies stochastically according to unspecified reaction rates:

$$\mathbf{z} \xrightarrow{W_i(\mathbf{z})} \mathbf{z} + \mathbf{d}_i \quad \text{for} \quad i = 1, 2, 3 \ldots, \tag{4.1}$$

where $W_i(\mathbf{z})$ is the probability per unit time that the i-th reaction occurs, and $\mathbf{d}_i$ is the step size of the i-th reaction.

The reaction rate functions $W_i(\mathbf{z})$ can depend on all the system variables or on a subset of them. The Markov chain can be drawn as a network, where nodes correspond to the components $\{z_k\}$, and directional links correspond to explicit parameter dependencies in the rates $W_i(\mathbf{z})$. For instance, if the component $z_n$ ever changes with a reaction rate that depends explicitly on $z_m$, then an arrow is drawn from the $z_m$ node to the $z_n$ node, see *Appendix 1—figure 21A*.

Note that because molecular abundances cannot be negative, any biochemical system must have the property that the propensity of a reaction must be equal to zero when the system is in a state in which an occurrence of that reaction would take one of the abundances to negative numbers. Stochastic networks that do not have this property cannot correspond to biochemical reaction networks. As a result, the abundances $\{z_k\}$ are bounded to be non-negative through reaction rate dependencies. For example, if $P(\mathbf{z}, t)$ and $z_k$ correspond to the abundance of molecular species $Z_j$ and $Z_k$ respectively which undergo the following conversion reaction $(z_j, z_k) \xrightarrow{W_i(\mathbf{z})} (z_j - 1, z_k + 1)$, then the transition probability function $W_i$ must have an explicit $z_i$ dependence in that the following must be satisfied: $W_i(\mathbf{z}) = 0$ when $z_j = 0$. Even in the particular case where the conversion rate is approximately independent of the educt concentration when the latter is non-zero, the reaction cannot physically occur when $z_j = 0$, and so $W_i$ must have an explicit $z_i$ dependence.

The system probability distribution $P(\mathbf{z}, t)$ for this network evolves according to the chemical master equation (*Van Kampen, 1992*; *Lestas et al., 2008*):

$$\frac{d}{dt} P(\mathbf{z}, t) = \sum_k \left[ W_k(\mathbf{z} - \mathbf{d}_k) P(\mathbf{z} - \mathbf{d}_k, t) - W_k(\mathbf{z}) P(\mathbf{z}, t) \right]. \tag{4.2}$$

Time evolution equations for the moments $\langle u_k^n \rangle$ of each component follow directly from the master equation .

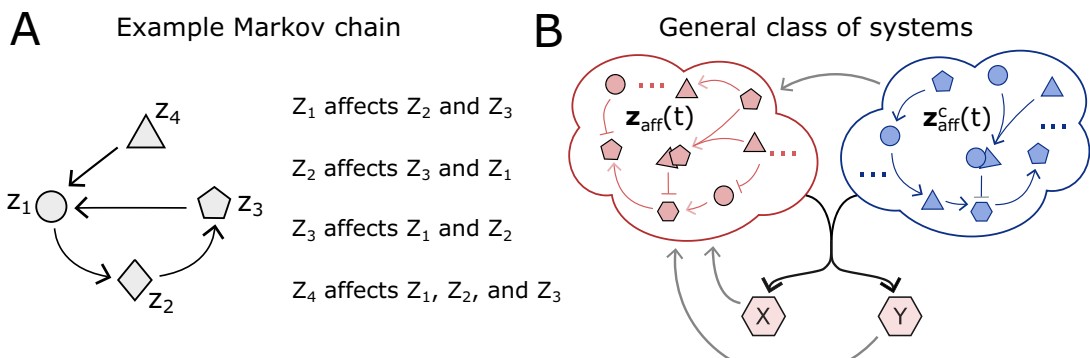

**Appendix 1—figure 21.** Definition of the class of systems modeling a gene and its passive reporter. (**A**) An example Markov chain with the state of the system given by $\mathbf{z} = (z_1, z_2, z_3, z_4)$. Nodes correspond to system variables, whereas arrows correspond to reaction rate dependencies. For instance, if $z_2$ changes under any

*Appendix 1—figure 21 continued*

transition with a reaction rate that depends explicitly on $z_1$, then we draw an arrow from $z_1$ to $z_2$. We say that a variable affects another variable if a path can be drawn in the network topology from the first variable to the second. (**B**) We consider all stochastic processes in which two cellular components, *X* and *Y*, are made with an identical, but unspecified, rate. This rate can depend in any way on a cloud of unknown time-varying components **z**, which in turn can depend in arbitrary ways on the number of *X* and *Y* molecules, denoted by *x* and *y*. The birth and death reactions of *X* and *Y* in *Equations 4.3; 4.4* are the only specified parts within this arbitrarily large network. We define two disjoint sets that make up all the unspecified components in the networks. The variables affected by *X* or *Y*, denoted by $\mathbf{z}_{\mathrm{aff}}$ are all the components in the network that *X* or *Y* affects. All other components in the system are denoted by $\mathbf{z}^c_{\mathrm{aff}}$, which can affect the components affected by *X* in an arbitrary way.

## A. mRNA reporters

We let *X* correspond to the transcript of a gene of interest, with $x$ corresponding to the number of *X* molecules. Functional interactions in this network are specified by the dependencies of the reaction rates in the network. We say that *X* affects another component *Z* if a path can be drawn from *X* to *Z* in the topology of the network, see *Appendix 1—figure 21A*. The set of all components in the network that are affected by *X* or *Y* is denoted by $\mathbf{z}_{\mathrm{aff}}$. All other components in the cell are denoted as $\mathbf{z}^c_{\mathrm{aff}}$, see *Appendix 1—figure 21B*. We let the *X* number dynamics be described by a stochastic birth-death process, with production and degradation reactions of *X* to be given by

$$x \xrightarrow{R(\mathbf{z}_{\mathrm{aff}},\, \mathbf{z}^c_{\mathrm{aff}})} x + 1 \qquad x \xrightarrow{x\beta(\mathbf{z}^c_{\mathrm{aff}})} x - 1, \tag{4.3}$$

where the production rate *R* can depend in any way on the sets of unspecified components $\mathbf{z}_{\mathrm{aff}}$ and $\mathbf{z}^c_{\mathrm{aff}}$, and the degradation rate $\beta$ can depend in any way on all components not affected by *X* or *Y*, as long as any possible realization of $\beta_t$ does not decay to zero over time (i.e., $\langle\beta\rangle_t = \lim_{T\to\infty} \frac{1}{T}\int_0^T \beta_t dt > 0$). If *X* is the transcript of *geneX*, then the rate *R* corresponds production rate corresponds to the transcription rate of *geneX*.

We next introduce into the system a passive reporter *Y*, engineered to read out the same signal as *X* that degrades with the same rate

$$y \xrightarrow{\alpha R(\mathbf{z}_{\mathrm{aff}},\, \mathbf{z}^c_{\mathrm{aff}})} y + 1 \qquad y \xrightarrow{y\beta(\mathbf{z}^c_{\mathrm{aff}})} y - 1, \tag{4.4}$$

where $\alpha$ is an arbitrary proportionality constant. We extend the definition of $\mathbf{z}_{\mathrm{aff}}$ to include all components affected by *X* or *Y*. These components can in turn depend in arbitrary ways on the number of *X* and *Y* molecules. The dynamics of all other cellular components in the cell thus remain unspecified, and the resulting dynamics need not be Markovian or ergodic in *X* and *Y*. Note that the sets of unknown components can include an unbounded number of nonphysical mock variables, which can model the different spatial compartments that *X* can find itself in as well as any complex history-dependent dynamics. We only specify that the total *X* abundance undergoes first-order degradation, where the degradation rate can depend in an arbitrary way on any of the variables not affected by *X* or *Y*.

## B. Fluorescent protein reporters

We can also consider a different class of systems in which *X* denotes the protein of a gene of interest that is fused to a fluorescent protein. This fluorescent protein must undergo a maturation step before it can fluoresce. We denote the immature fluorescent protein as *X*, and the transcript for this protein as $M_x$. The mRNA abundances are described by the following stochastic birth-death processes modeling transcription and mRNA degradation

$$m_x \xrightarrow{R(\mathbf{z}_{\mathrm{aff}},\, \mathbf{z}^c_{\mathrm{aff}})} m_x + 1 \qquad m_x \xrightarrow{m_x\beta(\mathbf{z}^c_{\mathrm{aff}})} m_x - 1 \tag{4.5}$$

The 'dark' fluorescent proteins $X_{\mathrm{d}}$ are translated from the mRNA and then mature into 'bright' fluorescent proteins *X*, which are then degraded:

$$x_d \xrightarrow{\lambda(\mathbf{z}^c_{\mathrm{aff}})m_x} x_d + 1 \qquad (x_d, x) \xrightarrow{x_d/\tau_{mat}} (x_d - 1, x + 1) \qquad x \xrightarrow{x\beta_p(\mathbf{z}^c_{\mathrm{aff}})} x - 1, \tag{4.6}$$

where $\lambda(\mathbf{z}_{\text{aff}}^c)$ is an unspecified translation rate, $\tau_{mat}$ is the average maturation time, and $\beta_p(\mathbf{z}_{\text{aff}}^c)$ is an unspecified protein degradation factor. The translation rate $\lambda$ and degradation rate $\beta_p$ can fluctuate and can depend in arbitrary ways on all the components not affected by $X$ or $M_x$ or $Y$ or $M_y$. We assume that any possible realization of $\beta_p$ does not decay to zero over time (i.e. $\langle\beta_p\rangle_t = \lim_{T\to\infty}\frac{1}{T}\int_0^T \beta_{p,t}dt > 0$). The fluorescent proteins are assumed to undergo first-order maturation. In *Balleza et al., 2018*, the authors show that approximately a third of a large set of common fluorescent proteins undergo first-order maturation kinetics in bacteria.

We next introduce into the system a passive reporter $Y$ corresponding to a second fluorescent protein reading out the same transcription rate:

$$
\begin{aligned}
m_y &\xrightarrow{\alpha R(\mathbf{z}_{\text{aff}}, \mathbf{z}_{\text{aff}}^c)} m_y + 1 \qquad m_y \xrightarrow{m_y \beta(\mathbf{z}_{\text{aff}}^c)} m_y - 1 \\
y_d &\xrightarrow{\alpha_p \lambda(\mathbf{z}_{\text{aff}}^c) m_y} y_d + 1 \qquad (y_d, y) \xrightarrow{y_d/\tau_{mat}} (y_d - 1, y + 1) \qquad y \xrightarrow{y \beta_p(\mathbf{z}_{\text{aff}}^c)} y - 1,
\end{aligned}
\tag{4.7}
$$

where the transcription and translation rates are proportional to those of *geneX* with unknown proportionality constants $\alpha$ and $\alpha_p$. The average maturation and degradation times are equal to those of $X$.

## 5. The invariant relation — Derivation of *Equation 2*

**Definition 1**. A co-transition event is a transition event in the system where two or more components change simultaneously. That is, a reaction where the step size $\mathbf{d}$ has more than one vector component.

An example would be a conversion event, where a chemical species $z_m$ converts to another chemical species $z_n$

$$
(z_m, z_n) \xrightarrow{W(z_m)} (z_m - 1, z_n + 1)
$$

According to our framework, an arrow would be drawn from $z_m$ to $z_n$ in the network if the above reaction was part of the system. Another example would be two molecules $z_m$, $z_n$ that bind to form a complex $z_l$

$$
(z_m, z_n, z_l) \xrightarrow{W(z_m, z_n)} (z_m - 1, z_n - 1, z_l + 1).
$$

If the above reaction was part of the system, an arrow would be drawn from $z_m$ to $z_n$, from $z_n$ to $z_m$, and from both $z_m$, $z_n$ to $z_l$.

In order to derive *Equation 2*, we need to assume that no components in $\mathbf{z}_{\text{aff}}^c$ are part of a co-transition event with any variable in $\mathbf{z}_{\text{aff}}$. We thus begin by proving the following lemma.

**Lemma 1.** Let $Z_k$ be a component that is not affected by $X$ or $Y$. For any network of the class in *Appendix 1—figure 21B* (*Equations 4.3; 4.4*), there exists another network with the exact same dynamics in $X$, $Y$, and $Z_k$, but where no components in $\mathbf{z}_{\text{aff}}^c$ are part of a co-transition event with any component in $\mathbf{z}_{\text{aff}}$.

**Proof of Lemma 1.** Let the following co-transition reaction be part of the system, where $\{a_k\}$ are variables in $\mathbf{z}_{\text{aff}}$ and $\{b_k\}$ are variables in $\mathbf{z}_{\text{aff}}^c$

$$
(\mathbf{a}, \mathbf{b}) \xrightarrow{W(\mathbf{b})} (\mathbf{a} + \mathbf{d}_a, \mathbf{b} + \mathbf{d}_b).
\tag{5.1}
$$

By definition of $\mathbf{z}_{\text{aff}}$ and $\mathbf{z}_{\text{aff}}^c$, the reaction rate $W$ in the above reaction cannot depend on the variables $\{a_k\}$. However, because $\mathbf{a}$ is part of $\mathbf{z}_{\text{aff}}$, the $\{a_k\}$ variables must be affected by $X$ or $Y$. Therefore, there must exist one or more reactions, labeled with $i \in \{1, 2, \dots\}$, in the system, that change $\mathbf{a}$ with reaction rates that depend on the variables affected by $X$ or $Y$

$$
\mathbf{a} \xrightarrow{W_i(\mathbf{z}_{\text{aff}}, \mathbf{z}_{\text{aff}}^c)} \mathbf{a} + \mathbf{d}_i \quad \text{for} \quad i = 1, 2, \dots
\tag{5.2}
$$

Note that the above reactions cannot make changes to $\mathbf{b}$; otherwise, the $\{b_k\}$ variables would not be in the variables not affected by $X$ or $Y$. We now consider an exact copy of the whole system $\mathbf{z}$ and the system reactions, with the change that the $\mathbf{a}$ variables are decomposed into two sets of mock variables. Specifically, we define the variables $\{a_k^{int}\}$ and $\{a_k^b\}$ that undergo the following reactions

$$(\mathbf{a}^b, \; \mathbf{b}) \xrightarrow{W(\mathbf{b})} \left( \mathbf{a}^b + \mathbf{d}_a, \; \mathbf{b} + \mathbf{d}_b \right)$$

$$\mathbf{a}^{int} \xrightarrow{W_i\left(\mathbf{z}_{\mathrm{aff}}, \, \mathbf{z}_{\mathrm{aff}}^c\right)} \mathbf{a}^{int} + \mathbf{d}_i \quad \text{for} \quad i = 1, 2, \ldots,$$

which correspond to the reactions in *Equations 5.1; 5.2*. We replace all the explicit $\mathbf{a}$ dependencies in the reaction rates of the system with $\mathbf{a}^{int} + \mathbf{a}^b$. As a result, all the reactions that depended on $\mathbf{a}$ are unchanged, but now we can put the $\mathbf{a}^b$ variables in $\mathbf{z}_{\mathrm{aff}}^c$, and we can put the $\mathbf{a}^{int}$ variables in $\mathbf{z}_{\mathrm{aff}}$. We are left with a system where the $\mathbf{a}^{int}$ are not part of a co-transition event with variables in $\mathbf{z}_{\mathrm{aff}}^c$, and the dynamics of $X$ and $Y$ remain unchanged. Moreover, none of the reaction rates that govern the dynamics of any component in the original $\mathbf{z}_{\mathrm{aff}}^c$ have been altered, meaning the dynamics of $Z_k$ remain unchanged. Such a decomposition can be done for any co-transition reaction that involves components from $\mathbf{z}_{\mathrm{aff}}$ and $\mathbf{z}_{\mathrm{aff}}^c$.

We now let $Z_k$ correspond to another component of interest in the system, as a continuous-time Markov process. It can be any stochastic process that can be measured, like the abundance of a molecular species in the network, the size of the cell, or any parameter that can influence the reaction rates of the system.

**Theorem 1**. Let $Z_k$ be a component in $\mathbf{z}_{\mathrm{aff}}^c$. If the averages $\langle x_t \rangle$, $\langle y_t \rangle$ and the covariances $\mathrm{Cov}(x_t, z_{k,t})$, $\mathrm{Cov}(x_t, z_{k,t})$ over the ensemble have reached a stationary state (i.e. they have become constant over time), then $\eta_{xz_k} = \eta_{yz_k}$.

**Proof of Theorem 1**. We condition on the components not affected by $X$ or $Y$, $\mathbf{z}_{\mathrm{aff}}^c[-\infty, t]$. This corresponds to a hypothetical system where all the variables in $\mathbf{z}_{\mathrm{aff}}^c$ become deterministic time-varying signals $\{z_k(t)\}$. The reactions governing X and Y in the conditional system become

$$x \xrightarrow{R(\mathbf{z}_{\mathrm{aff}}, \, t)} x + 1 \qquad y \xrightarrow{\alpha R(\mathbf{z}_{\mathrm{aff}}, \, t)} y + 1$$

$$\tag{5.3}$$

$$x \xrightarrow{x\beta(t)} x - 1 \qquad y \xrightarrow{y\beta(t)} y - 1$$

where $R\left(\mathbf{z}_{\mathrm{aff}}, t\right) = R\left(\mathbf{z}_{\mathrm{aff}}, \, \mathbf{z}_{\mathrm{aff}}^c(t)\right)$ and $\beta(t) = \beta\left(\mathbf{z}_{\mathrm{aff}}^c(t)\right)$ now have an explicit time dependence from the conditioned history $\mathbf{z}_{\mathrm{aff}}^c[-\infty, t]$. We let $A$ be the set of all integers $k$ such that the $k$-th reaction in *Equation 4.1* leads to a change in at least one of the components in $\mathbf{z}_{\mathrm{aff}}$. In the conditional probability space, the components affected in $\mathbf{z}_{\mathrm{aff}}$ follow the following reactions

$$\mathbf{z}_{\mathrm{aff}} \xrightarrow{W_k(\mathbf{z}_{\mathrm{aff}}, \, t)} \mathbf{z}_{\mathrm{aff}} + \mathbf{d}_k \quad \forall k \in A, \tag{5.4}$$

where $W_k(\mathbf{z}_{\mathrm{aff}}, \; t) = W_k\left(\mathbf{z}_{\mathrm{aff}}, \; \mathbf{z}_{\mathrm{aff}}^c(t)\right)$ now has an explicit time dependence from the conditioned history of the variables not affected by $X$ or $Y$. Note that if some components in $\mathbf{z}_{\mathrm{aff}}^c$ were part of a co-transition event with components in $\mathbf{z}_{\mathrm{aff}}$, then *Equations 5.3; 5.4* would not hold. This is because conditioning on the history of those extrinsic variables effectively conditions on those birth events in $\mathbf{z}_{\mathrm{aff}}$ that are caused by those co-transitions. However, from Lemma 1, we can always work with another network in which there are no such co-transition events, and where the dynamics of $X$ and $Y$ remain unchanged.

This conditional system follows the following master equation

$$\frac{d}{dt} \; P\left(x, y, \mathbf{z}_{\mathrm{aff}}, t \mid \mathbf{z}_{\mathrm{aff}}^c[-\infty, t]\right) =$$
$$\sum_{k \in A} \left[ W_k\left(\mathbf{z}_{\mathrm{aff}} - \mathbf{d}_k, t\right) P\left(x, y, \mathbf{z}_{\mathrm{aff}} - \mathbf{d}_k, t \mid \mathbf{z}_{\mathrm{aff}}^c[-\infty, t]\right) - W_k\left(\mathbf{z}_{\mathrm{aff}}, t\right) P\left(x, y, \mathbf{z}_{\mathrm{aff}}, t \mid \mathbf{z}_{\mathrm{aff}}^c[-\infty, t]\right) \right]$$
$$+ R\left(x - 1, y, \mathbf{z}_{\mathrm{aff}}, t\right) P\left(x - 1, y, \mathbf{z}_{\mathrm{aff}}, t \mid \mathbf{z}_{\mathrm{aff}}^c[-\infty, t]\right) - R\left(x, y, \mathbf{z}_{\mathrm{aff}}, t\right) P\left(x, y, \mathbf{z}_{\mathrm{aff}}, t \mid \mathbf{z}_{\mathrm{aff}}^c[-\infty, t]\right)$$
$$+ \alpha R\left(x, y - 1, \mathbf{z}_{\mathrm{aff}}, t\right) P\left(x, y - 1, \mathbf{z}_{\mathrm{aff}}, t \mid \mathbf{z}_{\mathrm{aff}}^c[-\infty, t]\right) - \alpha R\left(x, y, \mathbf{z}_{\mathrm{aff}}, t\right) P\left(x, y, \mathbf{z}_{\mathrm{aff}}; t \mid \mathbf{z}_{\mathrm{aff}}^c[-\infty, t]\right)$$
$$+ (x + 1)\beta(t) P\left(x + 1, y, \mathbf{z}_{\mathrm{aff}}, t \mid \mathbf{z}_{\mathrm{aff}}^c[-\infty, t]\right) - x\beta(t) P\left(x, y, \mathbf{z}_{\mathrm{aff}}, t \mid \mathbf{z}_{\mathrm{aff}}^c[-\infty, t]\right)$$
$$+ (y + 1)\beta(t) P\left(x, y + 1, \mathbf{z}_{\mathrm{aff}}, t \mid \mathbf{z}_{\mathrm{aff}}^c[-\infty, t]\right) - y\beta(t) P\left(x, y, \mathbf{z}_{\mathrm{aff}}, t \mid \mathbf{z}_{\mathrm{aff}}^c[-\infty, t]\right).$$

We consider the averages of $X$ and $Y$ conditioned on the upstream history, $\bar{x}(t) = E\left[x_t | \mathbf{z}_{\mathrm{aff}}^c[-\infty, t]\right]$ and $\bar{y}(t) = E\left[y_t | \mathbf{z}_{\mathrm{aff}}^c[-\infty, t]\right]$, where $x_t$ and $y_t$ are the $X$ and $Y$ abundances at time $t$. From the above

master equation, the time-evolution for these first moments can be derived (*Joly-Smith et al., 2021*; *Hilfinger and Paulsson, 2011*)

$$\frac{d\bar{x}}{dt} = \bar{R}(t) - \bar{x}\bar{\beta}(t) \quad \& \quad \frac{d\bar{y}}{dt} = \alpha\bar{R}(t) - \bar{y}\bar{\beta}(t),$$
(5.5)

where $\bar{R}(t) = E\left[R(x_t, y_t, \mathbf{z}_{\text{aff}}, t)|\mathbf{z}_{\text{aff}}^c[-\infty, t]\right]$ and $\bar{\beta}(t) = E\left[\beta(\mathbf{z}_{\text{aff}}^c)|\mathbf{z}_{\text{aff}}^c[-\infty, t]\right]$ are the average production and degradation rates conditioned on the history of the variables not affected by X or Y. Note that $\bar{\beta}(t) = E\left[\beta(\mathbf{z}_{\text{aff}}^c)|\mathbf{z}_{\text{aff}}^c[-\infty, t]\right] = \beta(\mathbf{z}_{\text{aff}}^c(t)) = \beta(t)$, because the time trajectory of $\mathbf{z}_{\text{aff}}^c$ is set through the conditioning on the upstream history. We can then take the expectation of $\bar{x}$ over all possible histories of $\mathbf{z}_{\text{aff}}^c$ to get

$$E[\bar{x}(t)]_{\text{histories}} = E\left[E\left[x_t \mid \mathbf{z}_{\text{aff}}^c[-\infty, t]\right]\right]_{\text{histories}} = \langle x_t \rangle,$$
(5.6)

which follows from the law of total expectation. We now let $Z_k$ correspond to any component in the network that is not affected by X or Y. It can be a molecular abundance, concentration, or another stochastic cellular variable like the growth rate of the cell. It follows that

$$E\left[\bar{x}(t)z_k(t)\right]_{\text{histories}} = E\left[E\left[x_t \mid \mathbf{z}_{\text{aff}}^c[-\infty, t]\right] \cdot E\left[z_{k,t} \mid \mathbf{z}_{\text{aff}}^c[-\infty, t]\right]\right]_{\text{histories}},$$
$$= E\left[E\left[x_t z_{k,t} \mid \mathbf{z}_{\text{aff}}^c[-\infty, t]\right]\right]_{\text{histories}} = \langle x_t z_{k,t} \rangle$$
(5.7)

where the second step comes from the fact that conditioning on the history of $\mathbf{z}_{\text{aff}}^c$ effectively also conditions on the history of $Z_k$ with $z_k(t) = E\left[z_{k,t}|\mathbf{z}_{\text{aff}}^c[-\infty, t]\right]$ (so x and $z_k$ are independent when conditioning on the $\mathbf{z}_{\text{aff}}^c$ history), the last step follows from the law of total expectation, and $z_{k,t}$ is the measured amount of $Z_k$ at time t. From *Equations 5.6; 5.7*, it follows that

$$\text{Cov}\left(x_t, z_{k,t}\right) = \text{Cov}\left(\bar{x}(t), \bar{z}_k(t)\right),$$
(5.8)

where $\bar{z}_k(t) = E\left[z_{k,t}|\mathbf{z}_{\text{aff}}^c[-\infty, t]\right] = z_k(t)$, where the last step comes from the fact that the time trajectory of $z_k$ is set through the conditioning of the upstream histories. Intuitively, *Equation 5.8* says that when X does not affect $Z_k$, the stochastic fluctuations of X average out when taking the covariance between X and $Z_k$. Strikingly, this is independent of any type of feedback that X may impose through interactions in the cloud of components $\mathbf{z}(t)$. As a result, the same should hold for Y, and since $\bar{y}(t)$ is governed by the same differential equation as $\bar{x}$, the intrinsic fluctuations that differentiate X and Y will average out when taking the covariances with $Z_k$.

That is, dividing the right equation in *Equation 5.5* with α, we write the general solution for $\bar{x}(t)$ and $\bar{y}(t)/\alpha$, and find

$$\bar{x}(t) = \bar{x}(0)e^{-\int_0^t \beta(u)du} + \int_0^t e^{-\left(\int_0^t \beta(u)du - \int_0^{t'} \beta(v)dv\right)} \bar{R}(t')dt'$$
(5.9)

$$\bar{y}(t)/\alpha = \bar{y}(0)e^{-\int_0^t \beta(u)du}/\alpha + \int_0^t e^{-\left(\int_0^t \beta(u)du - \int_0^{t'} \beta(v)dv\right)} \bar{R}(t')dt'$$
(5.10)

Taking the average over all histories and subtracting the equations we have

$$E[\bar{x}(t)]_{\text{histories}} - E[\bar{y}(t)]_{\text{histories}} = E\left[\left(\bar{x}(0) - \bar{y}(0)/\alpha\right)e^{-\int_0^t \beta(u)du}\right]_{\text{histories}}$$
$$\Rightarrow \langle x_t \rangle - \langle y_t \rangle/\alpha = E\left[\left(\bar{x}(0) - \bar{y}(0)/\alpha\right)e^{-\int_0^t \beta(u)du}\right]_{\text{histories}},$$
(5.11)

where in the second step we used *Equation 5.6* which also holds for y by symmetry. We now invoke the requirement that the averages $\langle x_t \rangle$ and $\langle y_t \rangle$ are stationary, meaning they are constant over time. In that case, $\langle x_t \rangle - \langle y_t \rangle/\alpha$ is constant over time, and so

$$\langle x_t \rangle - \langle y_t \rangle/\alpha = \lim_{t \to \infty}\left(\langle x_t \rangle - \langle y_t \rangle/\alpha\right).$$
(5.12)

Substituting *Equation 5.11*, we thus have

$$\langle x_t \rangle - \langle y_t \rangle / \alpha = E \left[ \left( \bar{x}(0) - \bar{y}(0)/\alpha \right) e^{-\int_0^\infty \beta(u)du} \right]_{\text{histories}}. \tag{5.13}$$

Now, we must have $\int_0^\infty \beta(u)du \to \infty$ when $\langle \beta \rangle_t = \lim_{T \to \infty} \frac{1}{T} \int_0^T \beta(u)du > 0$. Therefore, the right-hand side of *Equation 5.13* becomes 0:

$$\langle x_t \rangle - \langle y_t \rangle / \alpha = E \left[ \left( \bar{x}(0) - \bar{y}(0)/\alpha \right) e^{-\int_0^\infty \beta(u)du} \right]_{\text{histories}} = 0, \tag{5.14}$$

Similarly, we now multiply *Equations 5.10; 5.9* with $z_k(t)$, average over all histories, and subtract to obtain

$$llE[\bar{x}(t)z_k(t)]_{\text{histories}} - E[\bar{y}(t)z_k(t)]_{\text{histories}} = E \left[ \left( \bar{x}(0) - \bar{y}(0)/\alpha \right) z_k(t) e^{-\int_0^t \beta(u)du} \right]_{\text{histories}}$$
$$\Rightarrow \langle x_t z_{k,t} \rangle - \langle y_t z_{k,t} \rangle / \alpha = E \left[ \left( \bar{x}(0) - \bar{y}(0)/\alpha \right) z_k(t) e^{-\int_0^t \beta(u)du} \right]_{\text{histories}} = 0, \tag{5.15}$$

where in the second step we used *Equation 5.7* which also holds for $y$ by symmetry. The last step follows when $\langle x_t z_{k,t} \rangle$ and $\langle y_t z_{k,t} \rangle$ have reached stationarity and are constant over time, along with $\langle \beta \rangle_t = \lim_{T \to \infty} \frac{1}{T} \int_0^T \beta(u)du > 0$.

It then follows from *Equations 5.14; 5.15* that

$$\langle x \rangle = \langle y \rangle / \alpha \quad \& \quad \text{Cov}(x, z_k) = \text{Cov}(y, z_k)/\alpha. \tag{5.16}$$

Dividing $\text{Cov}(x, z_k)$ by $\langle x \rangle \langle z \rangle$, and $\text{Cov}(y, z_k)/\alpha$ by $\langle y \rangle \langle z \rangle / \alpha$, we find $\eta_{xz_k} = \eta_{yz_k}$.

If the reporter $Y$ is engineered to be passive (i.e., it does not affect components in the network), then a violation of *Equation 2* would imply that $X$ affects $Z_k$. Otherwise, such a violation would imply that $X$ or $Y$ affect $Z_k$.

Thus far, we assumed that all the components not affected by $X$ or $Y$ are part of a continuous-time Markov chain. This was in order to make a rigorous definition of causal interaction in our framework as a path in the topology of the transition rates. Alternatively, if we relax the requirement that the components not affected by $X$ or $Y$ be Markov chains (i.e. they can be a set of arbitrary stochastic processes), we can operationally define 'no causal interaction from $X$ or $Y$' to mean that we can condition on the history of those stochastic processes and write down *Equations 5.3; 5.4*. We can then operationally define any violation of *Equation 2* as a 'causal interaction from $X$ or $Y$' to a stochastic process $Z_k$.

## 6. Derivation of the invariant relation for fluorescent proteins

The derivation follows analogously to the proof of Theorem 1. We condition on the history of the components not affected by $X$ or $Y$, from which the following differential equations follow like those in *Equation 5.5*:

$$ll\frac{d\bar{m}_x}{dt} = \bar{R}(t) - \bar{m}_x \beta(t) \quad \& \quad \frac{d\bar{m}_y}{dt} = \alpha \bar{R}(t) - \bar{m}_y \beta(t)$$
$$\frac{d\bar{x}_d}{dt} = \lambda(t)\bar{m}_x - \bar{x}_d/\tau_{mat} \quad \& \quad \frac{d\bar{y}_d}{dt} = \alpha_p \lambda(t)\bar{m}_y - \bar{y}_d/\tau_{mat} \tag{6.1}$$
$$\frac{d\bar{x}}{dt} = \bar{x}_d/\tau_{mat} - \bar{x}\beta_p(t) \quad \& \quad \frac{d\bar{y}}{dt} = \bar{y}_d/\tau_{mat} - \bar{y}\beta_p(t),$$

where $\lambda(t) = \lambda(\mathbf{z}_{\text{aff}}^c(t))$ and $\beta_p = \beta_p(\mathbf{z}_{\text{aff}}^c(t))$ . When the degradation rates $\beta$, $\beta_p$ do not decay to zero for any possible realization, and when the dual reporter ensemble averages and their covariances with $Z_k$ have reached a time-independent state, it follows from *Equation 6.1* analogously to the derivation from *Equation 5.5* to *Equation 5.16* that $\eta_{xz_k} = \eta_{yz_k}$.

## 7. The X and Y production rates can differ by certain types of fluctuations

In Section 5, we assumed that the production rates of $X$ and $Y$ were proportional with an arbitrary proportionality constant. This requirement can be relaxed. Here, we show that the $X$ and $Y$ production rates can differ by certain types of fluctuations. We do this by introducing a cloud of components in

the network that model fluctuations in the *X* or *Y* production rates that cause them to not be exactly the same.

In particular, consider the following class of systems

$$x \xrightarrow{R_x\left(\mathbf{z}_{\text{aff}}, \mathbf{z}_{\text{aff}}^c\right)} x+1 \quad y \xrightarrow{R_y\left(\mathbf{z}_{\text{aff}}, \mathbf{z}_{\text{aff}}^c\right)} y+1$$
$$x \xrightarrow{x\beta\left(\mathbf{z}_{\text{aff}}^c\right)} x-1 \quad y \xrightarrow{y\beta\left(\mathbf{z}_{\text{aff}}^c\right)} y-1 \tag{7.1}$$

where $R_x$ and $R_y$ are now different production rates for *X* and *Y*. Given a component $Z_k$ that is not affected by *X* or *Y*, we define the following two disjoint sets: **A** and **B**, see *Appendix 1—figure 22*. The set of cellular components **A** corresponds to all components that affect $Z_k$, along with all components that affect the degradation rate of *X* and *Y*, along with the component of interest $Z_k$. The set **B** corresponds to all components that are not affected by *X* or *Y* but that also do not affect $Z_k$. Note that both sets together, $\mathbf{A} \cup \mathbf{B}$, form the set of components not affected by *X* or *Y* (which we refer to as $\mathbf{z}_{\text{aff}}^c$ in previous sections). The cloud of components **B** can model fluctuations in the *X* or *Y* production rates.

## General class of systems

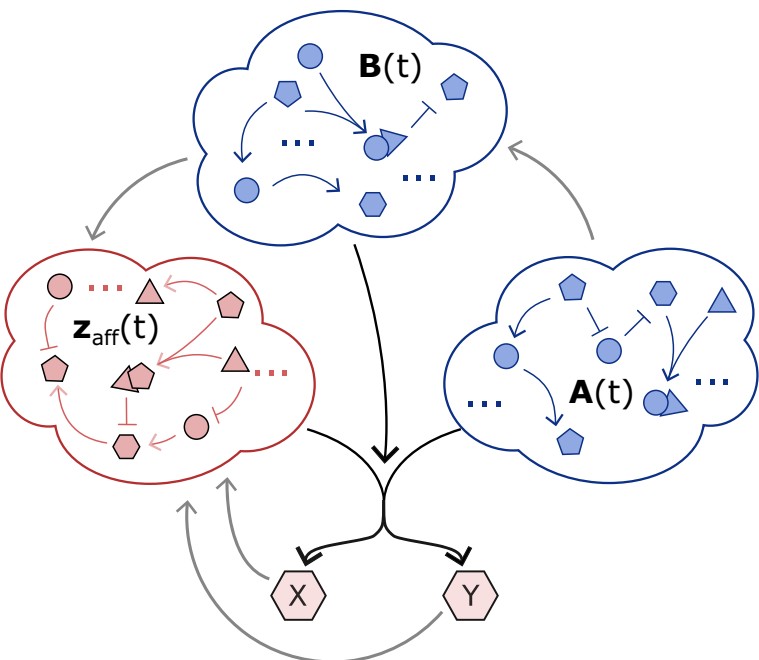

**Appendix 1—figure 22.** Definition of the class of systems modeling a gene and its passive reporter. Given a component $Z_k$ that is not affected by *X* or *Y*, we define **A** to be all the components in the network that affect $Z_k$, along with all the components that affect the degradation rates of *X* and *Y*, along with the component $Z_k$ itself. We define **B** to be all the components that are not affected by *X* or i and that are not in the set **A**. The union of **A** and **B** make up the cloud of components $\mathbf{z}_{\text{aff}}^c$ in *Appendix 1—figure 21* that are not affected by *X* or *Y*. We show in this section that the proof of *Equation 2* still holds when the production rates of *X* and *Y* differ by fluctuations that arise from the cloud of components **B**.

Let $R_x(x, y, \mathbf{z}_{\text{aff}}, \mathbf{A}, \mathbf{B})$ and $R_y(x, y, \mathbf{z}_{\text{aff}}, \mathbf{A}, \mathbf{B})$ be the production rates of *X* and *Y*, respectively. We can condition on the history of **A**, in which case the following differential equations follow similarly to *Equation 5.5*:

$$\frac{d\bar{\bar{x}}}{dt} = \bar{\bar{R}}_x(t) - \bar{\bar{x}}\bar{\bar{\beta}}(t) \quad \& \quad \frac{d\bar{\bar{y}}}{dt} = \bar{\bar{R}}_y(t) - \bar{\bar{y}}\bar{\bar{\beta}}(t), \tag{7.2}$$

$\bar{\bar{x}}(t) = E\left[x_t | \mathbf{A}[-\infty, t]\right]$ where , $\bar{\bar{y}}(t) = E\left[y_t | \mathbf{A}[-\infty, t]\right]$, and $\bar{\bar{\beta}}(t) = E\left[\beta(\mathbf{A}(t)) | \mathbf{A}[-\infty, t]\right]$, and where

$$\bar{\bar{R}}_x(t) = E\left[R_x(x_t, y_t, \mathbf{z}_{\text{aff}}, \mathbf{A}(t), \mathbf{B}(t)) | \mathbf{A}[-\infty, t]\right],$$
$$\bar{\bar{R}}_y(t) = E\left[R_y(x_t, y_t, \mathbf{z}_{\text{aff}}, \mathbf{A}(t), \mathbf{B}(t)) | \mathbf{A}[-\infty, t]\right].$$

These are the average production rates conditioned on the history of the components **A**. Note that *Equation 7.2* is identical to *Equation 5.5* if the average conditional rates are proportional to one another:

$$\bar{\bar{R}}_x(t) \propto \bar{\bar{R}}_y(t). \tag{7.3}$$

As long as *Equation 7.3* holds, the derivation that follows *Equation 5.5* applies here and it follows that $\eta_{xz_k} = \eta_{yz_k}$.

As a result, $R_x$ and $R_y$ can differ by any fluctuations that come out of the components in **B** or those in $\mathbf{z}_{\text{aff}}^c$ that average out when taking the average conditioned on the history of the components in **A**.

## Additive fluctuations

The covariability relation of *Equation 2* still holds if the production rates differ by some unspecified additive fluctuations that average out to zero. That is, if we have an unspecified additive noise term δ:

$$R_x(x_t, y_t, \mathbf{z}_{\text{aff}}(t), \mathbf{A}(t), \mathbf{B}(t)) = R_y(x_t, y_t, \mathbf{z}_{\text{aff}}(t), \mathbf{A}(t), \mathbf{B}(t)) + \delta(x_t, y_t, \mathbf{z}_{\text{aff}}, \mathbf{A}, \mathbf{B}) \tag{7.4}$$

such that $E\left[\delta(x_t, y_t, \mathbf{z}_{\text{aff}}, \mathbf{A}, \mathbf{B})|\mathbf{A}[-\infty, t]\right] = 0$. This noise term $\delta$ can depend in an arbitrary way on components in the system, as long as the fluctuations average out to zero when conditioned on the upstream history of **A**.

For example, we can consider the following birth-death process:

$$w \xrightarrow{\lambda} w+1 \quad p \xrightarrow{\lambda_p} p+1 \quad x \xrightarrow{w+K(p-\delta\langle p\rangle)} x+1 \quad y \xrightarrow{w} y+1 \quad z \xrightarrow{w} z+1$$
$$w \xrightarrow{w/\tau} w-1 \quad p \xrightarrow{p/\tau_p} p-1 \quad x \xrightarrow{x/\tau} x-1 \quad y \xrightarrow{y/\tau} y-1 \quad z \xrightarrow{z/\tau} z-1 \tag{7.5}$$

Here, $X$ and $Y$ have production rates given by the abundance of $W$, with $X$ having an additional additive term $K(p - \delta\langle p\rangle)$ which depends on the fluctuations in $p$ abundances. Both $W$ and $p$ fluctuate according to a Poisson process. Crucially, when $\delta = 1$ the additive noise term will average to zero, in which case the invariant of *Equation 2* must still be satisfied because $X$ and $Y$ do not affect $Z$, see *Appendix 1—figure 23A*. However, when $\delta = 0$ the additive noise term does not average to zero, which can lead to a deviation of *Equation 2* even when $X$ and $Y$ do not affect $Z$, see *Appendix 1—figure 23B*. This indicates that deviations from *Equation 2* can occur when the $X$ and $Y$ production rates differ by certain types of fluctuations even when $X$ and $Y$ do not affect $Z$.

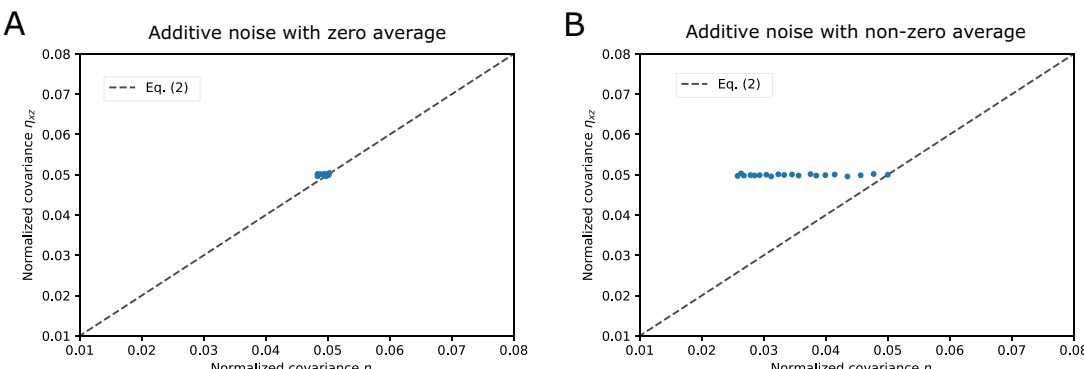

**Appendix 1—figure 23.** An example system shows that the invariant holds when the rates differ by additive fluctuations that average to zero. (**A**) Simulations of the birth-death process of *Equation 7.5* with $\delta = 1$, which ensures that the additive noise term averages to zero. Each dot corresponds to a single iteration of the Gillespie algorithm that was run until each reaction occurred at least $10^7$ times. In each dot, the parameters are $\tau = 1$, $\tau_p = 0.1$, $\lambda_p = 5/\tau_p$, $\lambda = 10$. The $K$ parameter quantifies the strength of the additive noise and was varied from 0 to 40 in increments of 2, with each dot having a particular $K$ value. As can be seen, the dots all lie on or very near the line given by *Equation 2*. Small deviations from the line are inevitable and are due to numerical errors brought on by finite sampling of the stochastic process. (**B**) Same as in A except now $\delta = 0$, which means the additive noise term does not average to zero. Here as the $K$ parameter is increased and more noise is added to the $X$ production rate, there is further deviation from *Equation 2* even as $X$ and $Y$ do not affect $Z$.

As an example, an additive noise term of this type can be the result of a linear cascade of an unspecified number of variables upstream of $X$ and $Y$. That is, consider the following system

$$x_1 \xrightarrow{R(\mathbf{z}_{\text{aff}}, \mathbf{z}_{\text{aff}}^c)} x_1 + 1 \quad x_2 \xrightarrow{\lambda_2 x_1} x_2 + 1 \quad \ldots \quad x_n \xrightarrow{\lambda_n x_{n-1}} x_n + 1$$

$$x_1 \xrightarrow{\beta_1(\mathbf{z}_{\text{aff}}^c)} x_1 - 1 \quad x_2 \xrightarrow{\beta_2(\mathbf{z}_{\text{aff}}^c)} x_2 - 1 \quad \ldots \quad x_n \xrightarrow{\beta_n(\mathbf{z}_{\text{aff}}^c)} x_n - 1$$

$$y_1 \xrightarrow{\alpha_1 R(\mathbf{z}_{\text{aff}}, \mathbf{z}_{\text{aff}}^c)} y_1 + 1 \quad y_2 \xrightarrow{\alpha_2 \lambda_2 y_1} y_2 + 1 \quad \ldots \quad y_n \xrightarrow{\alpha_n \lambda_n y_{n-1}} y_n + 1$$

$$y_1 \xrightarrow{\beta_1(\mathbf{z}_{\text{aff}}^c)} y_1 - 1 \quad y_2 \xrightarrow{\beta_2(\mathbf{z}_{\text{aff}}^c)} y_2 - 1 \quad \ldots \quad y_n \xrightarrow{\beta_n(\mathbf{z}_{\text{aff}}^c)} y_n - 1$$

$$x \xrightarrow{\lambda x_n} x + 1 \qquad\qquad y \xrightarrow{\alpha \lambda y_n} y + 1$$

$$x \xrightarrow{x \beta(\mathbf{z}_{\text{aff}}^c)} x - 1 \qquad\qquad y \xrightarrow{y \beta(\mathbf{z}_{\text{aff}}^c)} y - 1$$

where the $\{\lambda_k\}$, $\{\alpha_k\}$, $\lambda$, and $\alpha$ parameters are unspecified constants. The result of this system is that the production rates of $X$ and $Y$ will differ by additive fluctuations, yet Theorem 1 still holds.

## Multiplicative noise that is independent of the production rates:

The covariability relation of *Equation 2* still holds if the production rates differ by some unspecified multiplicative fluctuations that are independent of the production rates:

$$R_x = n_x R(x_t, y_t, \mathbf{z}_{\text{aff}}(t), \mathbf{A}(t), \mathbf{B}(t)) \quad \text{and} \quad R_y = n_y R(x_t, y_t, \mathbf{z}_{\text{aff}}(t), \mathbf{A}(t), \mathbf{B}(t)) \tag{7.6}$$

where $n_x$ and $n_y$ are multiplicative random variables such that

$$E\left[n_x R(x_t, y_t, \mathbf{z}_{\text{aff}}, \mathbf{A}, \mathbf{B}) | \mathbf{A}[-\infty, t]\right] = E\left[n_x\right] \cdot E\left[R(x_t, y_t, \mathbf{z}_{\text{aff}}, \mathbf{A}, \mathbf{B}) | \mathbf{A}[-\infty, t]\right]$$

$$E\left[n_y R(x_t, y_t, \mathbf{z}_{\text{aff}}, \mathbf{A}, \mathbf{B}) | \mathbf{A}[-\infty, t]\right] = E\left[n_y\right] \cdot E\left[R(x_t, y_t, \mathbf{z}_{\text{aff}}, \mathbf{A}, \mathbf{B}) | \mathbf{A}[-\infty, t]\right]$$

and where $E\left[n_x | \mathbf{A}[-\infty, t]\right] = E\left[n_x | \mathbf{A}[-\infty, t]\right]$. These noise terms $n_x$ and $n_y$ can depend in an arbitrary way on components in the system, as long as they remain independent of the production rate $R$ when conditioned on the upstream history of $\mathbf{A}$.

## 8. Bursty gene expression

In Eukaryotes, a popular model of gene expression assumes that transcription occurs in bursts (*Raj et al., 2006*; *Raj and van Oudenaarden, 2008*; *Bahar Halpern et al., 2015*). The dynamics of such bursts are then described by the burst frequency and the burst size. Here we show that the invariant *Equation 2* holds in the face of transcription bursting, as long as the burst frequency is identical for both *geneX* and the passive reporter *geneY*.

### Coordinated bursting

Dual reporter genes with identical promoters have been engineered in mammalian cells (*Raj et al., 2006*), where it was observed that they underwent bursts in gene expression. Of note, the bursts between these dual reporters were highly correlated when the genes were placed at similar gene loci. On the other hand, the bursts were not correlated when placed at distant gene loci. We first consider the case where the dual reporters undergo perfectly coordinated bursting, which would model the former case.

In fact, the class of systems described in Section 4.1 encompasses such coordinated bursting transcription rates. The rate $R$ is left unspecified and can vary by switching stochastically between different states, which could model sudden bursts in the production of mRNA. This class of systems can include transcription rates with arbitrary bursting dynamics, frequencies, and burst sizes. The key assumption, however, is that a burst in *geneX* will occur at the same time as a burst in *geneY*.

### Un-coordinated bursting with constant burst sizes

Bursts in mRNA abundances due to bursty transcription rates can be modeled by adding a burst size $b$ (a positive integer) to the class of systems:

$$x \xrightarrow{R(\mathbf{z}_{\text{aff}}, \mathbf{z}_{\text{aff}}^c)} x + b_x \qquad x \xrightarrow{x\beta(\mathbf{z}_{\text{aff}}^c)} x - 1,$$
$$y \xrightarrow{\alpha R(\mathbf{z}_{\text{aff}}, \mathbf{z}_{\text{aff}}^c)} y + b_y \qquad y \xrightarrow{y\beta(\mathbf{z}_{\text{aff}}^c)} y - 1. \tag{8.1}$$

Here, $R$ corresponds to the burst frequency, while $b_x$ and $b_y$ are the sizes of the bursts in the $X$ and $Y$ abundances, respectively. Of note, a burst in the $X$ abundances can occur at a different time as a burst in $Y$ abundances (and have a different size). The key assumption here is that the probability that a burst occurs at any given time in the $Y$ abundance is proportional to the respective probability at the same given time for $X$.

The derivation described in Section 5 follows for this class of systems, with the exception that a factor of $b_x$ and $b_y$ will go on front of the $\bar{R}(t)$ terms in *Equation 5.5*. Nevertheless, the normalized covariances are not affected by these burst sizes, and so the final result remains identical: the invariant of *Equation 2* holds for this class of transcriptional bursting.

It has been hypothesized that chromatin remodeling is a cause of transcriptional bursting in Eukaryotes (*Raj and van Oudenaarden, 2008*). Under this model, transcription of a gene is free to occur when the surrounding chromatin is in an open state. When the chromatin is in a condensed state, however, transcription of the gene cannot occur. Under this hypothesis, the class of systems of *Equation 8.1* could correspond to two genes located at different loci that each have the same probability at a given moment in time to suddenly be exposed due to stochastic chromatin remodeling.

## Un-coordinated bursting with distributed burst sizes

In the previous section and in the class of systems described by *Equation 8.1*, the burst sizes $b_x$ and $b_y$ are constant. Here, we show that the invariant of *Equation 2* still holds when the burst sizes are distributed. Such distributed burst sizes can be modeled through a series of stochastic reactions that can occur for each possible burst size:

$$x \xrightarrow{R_k(\mathbf{z}_{\text{aff}}, \mathbf{z}_{\text{aff}}^c)} x + k \qquad \text{for } k = 1, 2, 3, \ldots \tag{8.2}$$

Here, a burst of $k$ molecules can be produced with probabilistic rate $R_k$. The production dynamics of the $Y$ molecules are described by

$$y \xrightarrow{\alpha R_k(\mathbf{z}_{\text{aff}}, \mathbf{z}_{\text{aff}}^c)} y + b_y k \qquad \text{for } k = 1, 2, 3, \ldots \tag{8.3}$$

Here, the burst frequencies for each possible burst size can be multiplied by an arbitrary factor α, and the burst sizes can also be multiplied by an arbitrary integer $b_y$. Following the same derivation as described in Section 5, it follows that the invariant of *Equation 2* holds for this class of systems that models un-coordinated bursting with distributed burst sizes. Note that since the individual rates $R_k$ for each burst size are not specified, any distribution of burst sizes is included.

In this class of systems, a burst is modeled as occurring instantaneously. It remains to be shown that the invariant of *Equation 2* holds in the face of slow bursts in transcription that are un-coordinated between the dual reporters.

## 9. Simulated example systems from *Figure 1B*

To illustrate *Equation 2* in *Figure 1B* of the main text, we simulated a number of four-component stochastic birth-death processes with the following general rates

$$x \xrightarrow{r_x^+} x + 1 \quad y \xrightarrow{r_y^+} y + 1 \quad z_k \xrightarrow{r_z^+} z_k + 1 \quad w \xrightarrow{r_w^+} w + 1$$
$$x \xrightarrow{r_x^-} x - 1 \quad y \xrightarrow{r_y^-} y - 1 \quad z_k \xrightarrow{r_z^-} z_k - 1 \quad w \xrightarrow{r_w^-} w - 1$$

For cases in which $X$ affects $Z$ (circles in *Figure 1B*), we simulated the following systems.

1. $r_x^+ = \lambda w$, $r_y^+ = \alpha \lambda w$, $r_z^+ = \lambda_z x$, $r_w^+ = \lambda_w$, $r_x^- = x/\tau$, $r_y^- = y/\tau$, $r_z^- = x/\tau_z$, $r_w^- = w/\tau_w$
2. $r_x^+ = \lambda w$, $r_y^+ = \alpha \lambda w$, $r_z^+ = \lambda_z \frac{w^2}{0.01 + 0.1x}$, $r_w^+ = \lambda_w$, $r_x^- = x/\tau$, $r_y^- = y/\tau$, $r_z^- = x/\tau_z$, $r_w^- = w/\tau_w$
3. $r_x^+ = \lambda w$, $r_y^+ = \alpha \lambda w$, $r_z^+ = \lambda_z \frac{w^2}{0.01 + 0.001x}$, $r_w^+ = \lambda_w$, $r_x^- = x/\tau$, $r_y^- = y/\tau$, $r_z^- = x/\tau_z$, $r_w^- = w/\tau_w$
4. $r_x^+ = \lambda w$, $r_y^+ = \alpha \lambda w$, $r_z^+ = \lambda_z \frac{x}{0.01 + w}$, $r_w^+ = \lambda_w$, $r_x^- = x/\tau$, $r_y^- = y/\tau$, $r_z^- = x/\tau_z$, $r_w^- = w/\tau_w$

5. $r_x^+ = \lambda w,\quad r_y^+ = \alpha\lambda w,\quad r_z^+ = \lambda_z \frac{w^2}{0.01+0.5x+w},\quad r_w^+ = \lambda_w,\quad r_x^- = x/\tau,\quad r_y^- = y/\tau,\quad r_z^- = x/\tau_z,\quad r_w^- = w/\tau_w$

For cases in which $X$ does not affect $Z$ (squares in *Figure 1B*), we simulated the following systems.

1. $r_x^+ = \lambda w,\quad r_y^+ = \alpha\lambda w,\quad r_z^+ = \lambda_z w,\quad r_w^+ = \lambda_w,\quad r_x^- = x/\tau,\quad r_y^- = y/\tau,\quad r_z^- = x/\tau_z,\quad r_w^- = w/\tau_w.$

2. $r_x^+ = \lambda w,\quad r_y^+ = \alpha\lambda w,\quad r_z^+ = \lambda_z \frac{1}{1+w},\quad r_w^+ = \lambda_w,\quad r_x^- = x/\tau,\quad r_y^- = y/\tau,\quad r_z^- = x/\tau_z,\quad r_w^- = w/\tau_w.$

For each system, 1000 simulations were performed, each time randomly picking the following parameters $\alpha$, $\tau_z$, $\tau_w$, $\lambda$, $\lambda_z$, and $\lambda_w$ from the respective sets [0, 2], [0, 2], [0, 2], [0, 10], [0,10], and [0, 10], where [$a$, $b$] is defined as the set of all real numbers $n$ that satisfy $a \leq n \leq b$. The parameter $\tau$ was set to 1 throughout. Simulations were performed using the Gillespie algorithm (*Gillespie, 1977*) using C. Trajectories were simulated for $10^7$ reaction events. Normalized covariances were computed by integrating over the trajectories to obtain time averages for first and second moments. This is equivalent to using the distribution given by calculating the fraction of the total system time spent in each sampled state. In the ergodic regime, this distribution converges to the stationary distribution of the ensemble.

No violations of *Equation 2* were found, with plotted circles and squares corresponding to a representative sub-sample, with arbitrarily chosen sampling density, of our numerical simulations to indicate the accessible space

## 10. Example system with feedback from *Appendix 1—figure 1*

For *Appendix 1—figure 1*, we simulated the following stochastic birth-death process

$$
\begin{aligned}
z &\xrightarrow{\lambda_z} z + 1 & x &\xrightarrow{R(x,z)} x + 1 & y &\xrightarrow{R(x,z)} y + 1 & \text{with} \quad R(x, z) = \lambda \left(1 + \left(\frac{x}{z+b}\right)^n\right)^{-1} \\
z &\xrightarrow{z\beta_z} z - 1 & x &\xrightarrow{x\beta} x - 1 & y &\xrightarrow{y\beta} y - 1
\end{aligned}
\tag{10.1}
$$

where $\lambda_z = 0.02$, $\beta_z = 0.001$, $\lambda = 10$, $b = 0.01$, $\beta = 0.01$, and $n$ was varied to tune the feedback strength. In this system, $Z$ affects both $X$ and $Y$ because the rate $R$ depends on $z$. On the other hand, $X$ is not affected by $X$ and $Y$. The rate $R$ also depends on the $X$ numbers, leading to negative feedback in $X$ but not in $Y$. This system was simulated using the Gillespie algorithm (*Gillespie, 1977*).

For *Appendix 1—figure 1a* we set $n=20$, and simulated the system for 100,000 reaction events. The initial conditions at $t=0$ were $z=x=y=0$. The time-trace shown in *Appendix 1—figure 1a* is a small segment of the total simulated time-trace, whereas the shown distributions are made by pooling the entire time-trace into distributions (the system is ergodic). The Z abundance was given a constant offset of 3 in *Appendix 1—figure 1a* for illustrative purposes to align the time-traces.

As a measure of feedback strength in this system, we use the absolute value of the logarithmic gain taken at the averages

$$
\text{Feedback strength} = \left| \left( \frac{\partial \ln\left(R(x,z)\right)}{\partial \ln\left(x\right)} \Big|_{\langle x \rangle, \langle z \rangle} \right) \right| = n \left( \frac{\langle x \rangle^n}{\left(\langle z \rangle + b\right)^n + \langle x \rangle^n} \right).
$$

This quantifies how the rate $R$ changes with $x$ (*Paulsson, 2004*), and for this system is approximately equal to $n$. For *Appendix 1—figure 1b* we varied the feedback strength by simulating systems with different $n$ from 0 to 9. Each simulation ran for $10^7$ reaction events, and the Pearson correlation coefficients and the normalized covariances were computed from the distributions pooled from the sample paths as described above.

## 11. Details on the models simulated in *Figure 2*

Here we provide details of the stochastic birth-death processes that were simulated to generate the normalized covariances plotted in *Figure 2*. The full model rates with selected parameters are described in *Appendix 1—table 1*. Additional details on the computation of the normalized covariances, as well as sampling error estimations, are discussed in the Materials and Methods of the main text. The algorithm used to simulate these systems is discussed in Section 14

In each of the process descriptions below, we first show the network topology, followed by the stochastic birth-death reactions. Note that some processes share the same topology but differ in the rate functions. Moreover, we denote $[x]$ as the concentration of $X$, defined as $x/V(t)$, where $V(t)$ is the cell volume at time $t$ (similarly for $[y]$, $[z]$, and $[w]$). Some of the rates depend directly on the cell volume $V(t)$. In these reactions, we write $V(t)$ as $V$ in the rate functions to shorten the notation, but it should be noted that when $V$ is written in a rate function, it still corresponds to the cell volume at time $t$. The cell volume varies over time along with the molecular abundances, and three different volume dynamics are simulated as described in the Materials and methods section of the main text.

Note that in most of the systems below, we included a feedback mechanism. This is because feedback can make the dynamics of $X$ and $Y$ vastly different even though they are under the same control (see *Appendix 1—figure 1*). We thus added feedback in most of these systems to demonstrate that the invariant of *Equation 2* holds even when $X$ and $Y$ have different dynamics.

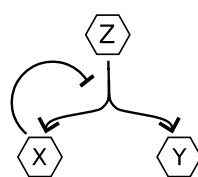

$$x \xrightarrow{\frac{\lambda z}{K+x}} x+1 \quad y \xrightarrow{\frac{\lambda z}{K+x}} y+1 \quad z \xrightarrow{\lambda_z} z+1$$
$$x \xrightarrow{\beta x} x-1 \quad y \xrightarrow{\beta y} y-1 \quad z \xrightarrow{\beta_z z} z-1$$

**Appendix 1—figure 24.** Process 1 — Negative feedback in X but not Y. Here the *X* and *Y* production rate depends linearly on the *Z* abundances, but is also suppressed by *X* abundances through negative feedback. The *Z* component is produced according to a Poisson process. This could model, for example, a gene regulated by an upstream transcription factor that also has an auto-repression mechanism. All components in this model are degraded with a first-order reaction with constant degradation constants.

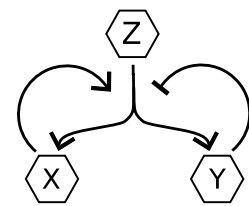

$$x \xrightarrow{\frac{\lambda(x+\delta)}{\left(K+x+\left(\frac{y}{z+\delta}\right)^2\right)}} x+1 \quad y \xrightarrow{\frac{\lambda(x+\delta)}{\left(K+x+\left(\frac{y}{z+\delta}\right)^2\right)}} y+1 \quad z \xrightarrow{\lambda_z} z+1$$
$$x \xrightarrow{\beta x} x-1 \quad y \xrightarrow{\beta y} y-1 \quad z \xrightarrow{\beta_z z} z-1$$

**Appendix 1—figure 25.** Process 2 — Asymmetric feedback. Though we state in the main text that *Y* is a passive reporter that does not affect other components in the network, in the proof of the invariant of *Equation 2* we only assume that *Y* does not affect the *Z* component of interest. Therefore, as long as *Y* does not affect *Z*, the invariant of *Equation 2* should hold even if *X* and *Y* are involved in a feedback mechanism that depends on different ways on the *X* and *Y* abundances. To test this, we simulated the process above. Here, the *X* and *Y* production rate is a hill function that increases with *X* and *Z* abundances, but has noise suppression in *Y* abundances. The δ parameter is a small number that is added to stop the rate from ever reaching zero when x=0 (otherwise the rate would stay at zero at the first moment that the *X* abundances hit zero). There is also a δ in the denominator to stop the denominator from ever becoming zero which would lead to numerical error.

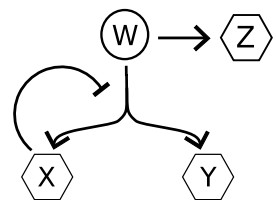

$$x \xrightarrow{\frac{\lambda w}{K+x^2}} x+1 \quad y \xrightarrow{\frac{\lambda w}{K+x^2}} y+1 \quad z \xrightarrow{\lambda_z w} z+1 \quad w \xrightarrow{\lambda_w} w+1$$

$$x \xrightarrow{\beta x} x-1 \quad y \xrightarrow{\beta y} y-1 \quad z \xrightarrow{\beta_z z} z-1 \quad w \xrightarrow{\beta_w w} w-1$$

**Appendix 1—figure 26.** Process 3 — Negative feedback with a confounding variable. Next, we test the invariant on a process in which $Z$ is correlated with $X$ and $Y$ through a confounding variable $W$. Here, $X$, $Y$, and $Z$ are produced with rates that are proportional to the $W$ abundances. However, the $X$ and $Y$ production rate is also suppressed by the $X$ abundances through negative feedback. This could model, for example, a transcription factor that regulates $X$, $Y$, and $Z$, in which $X$ is involved in an auto-repression mechanism.

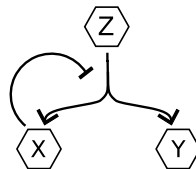

$$x \xrightarrow{\frac{\lambda[z]}{1+[x]^2}} x+1 \quad y \xrightarrow{\alpha\frac{\lambda[z]}{1+[x]^2}} y+1 \quad z \xrightarrow{\lambda_z} z+1$$

$$x \xrightarrow{\beta x} x-1 \quad y \xrightarrow{\beta y} y-1 \quad z \xrightarrow{\beta_z z} z-1$$

**Appendix 1—figure 27.** Process 4 — Negative feedback with concentration dependence 1. In the previous systems, the rates depend on the abundances of the components. However, reaction rates can depend on the concentration of components instead of the abundances, which will exhibit different dynamics in growing and dividing cells. In order to demonstrate that the invariant of *Equation 2* holds when the reaction rates depend on concentrations, we simulated the above process. Here, the production rate of $X$ and $Y$ increases linearly with the $Z$ concentration, but is suppressed by the $X$ concentration. The $Z$ production rate is a Poisson process, and all components undergo first-order degradation with constant degradation constant.

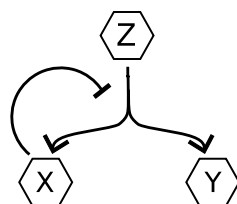

$$x \xrightarrow{\alpha\frac{\lambda V}{1+\left(\frac{[x]}{\delta+[z]}\right)^3}} x+1 \quad y \xrightarrow{\frac{\lambda V}{1+\left(\frac{[x]}{\delta+[z]}\right)^3}} y+1 \quad z \xrightarrow{\lambda_z V} z+1$$

$$x \xrightarrow{\beta x} x-1 \quad y \xrightarrow{\beta y} y-1 \quad z \xrightarrow{\beta_z z} z-1$$

**Appendix 1—figure 28.** Process 5 — Negative feedback with concentration dependence 2. Here we simulated another process with rates that depend on the component concentrations. However, here the rates scale with the cell volume $V$. This is to model a transcription rate that scales with the cell volume. In growing and dividing cells, this scaling in the transcription rate can allow for the mRNA concentration to be on average constant throughout the cell cycle, which has been observed in many genes. The $\delta$ parameter is added so that the denominator does not ever go to zero, which would cause numerical error.

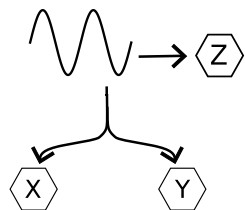

$$x \xrightarrow{A(\sin(\omega t)+1)V} x+1 \quad y \xrightarrow{B(\sin(\omega t)+1)V} y+1 \quad z \xrightarrow{C(\sin(\omega t)+1)V} z+1$$

$$x \xrightarrow{\beta x} x-1 \quad y \xrightarrow{\beta y} y-1 \quad z \xrightarrow{\beta_z z} z-1$$

**Appendix 1—figure 29.** Process 6 — Shared time-varying upstream signal. Here we consider a process where the X, Y, and Z production rates are driven by a common upstream signal. This signal is a deterministic sine function. The rates scale with the cell volume V so that the rate of production of the concentrations, given by the abundance production rates divided by the cell volume, are themselves sine functions. This could model, for example, three genes that are co-regulated by an oscillating signal.

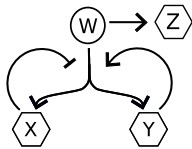

$$x \xrightarrow{\alpha \frac{([y]+\delta)}{[y]+\delta+\left(\frac{[x]}{\delta+[w]}\right)^2}} x+1 \quad y \xrightarrow{\frac{([y]+\delta)}{[y]+\delta+\left(\frac{[x]}{\delta+[w]}\right)^2}} y+1 \quad z \xrightarrow{\lambda_z[w]} z+1 \quad w \xrightarrow{\lambda_w V} w+1$$

$$x \xrightarrow{\beta x} x-1 \quad y \xrightarrow{\beta y} y-1 \quad z \xrightarrow{\beta_z z^2} z-1 \quad w \xrightarrow{\beta_w w} w-1$$

**Appendix 1—figure 30.** Process 7 — Asymmetric feedback with confounding variable and concentration dependencies 1. This process is similar to process 2 in that the production rate of X and Y depends on both components in different ways. In particular, the shared production rate of X and Y is a hill function that increases with Y concentration but is suppressed by X concentrations. There is also a confounding variable W that regulates the dual reporters by acting as the hill coefficient in the shared production rate of X and Y. The Z production is also regulated by the W component, through a linear dependency on the W concentration.

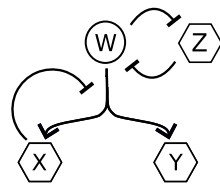

$$x \xrightarrow{\frac{\lambda}{1+\left(\frac{[x]}{\delta+w}\right)^3}} x+1 \quad y \xrightarrow{\frac{\lambda}{1+\left(\frac{[x]}{\delta+w}\right)^3}} y+1 \quad z \xrightarrow{\frac{\lambda_z}{1+\left(\frac{[w]}{k}\right)^3}} z+1 \quad w \xrightarrow{\frac{\lambda_z}{1+\left(\frac{[z]}{k}\right)^3}} w+1$$

$$x \xrightarrow{\beta x} x-1 \quad y \xrightarrow{\beta y} y-1 \quad z \xrightarrow{\beta_z z} z-1 \quad w \xrightarrow{\beta_w w} w-1$$

**Appendix 1—figure 31.** Process 8: Interaction mediated by a confounding variable. Here we have a process where the X and Y production rate is a hill function that depends non-linearly on the abundance of component W. In addition, the production rate of the W component is repressed by the Z concentration, which itself is repressed by the W concentration. Therefore, X and Y are correlated with Z because they are affected by Z (through W), but also because W affects X, Y, and Z.

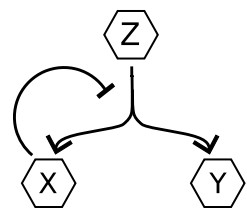

$$x \xrightarrow{\alpha \frac{\lambda z}{K+x}} x+1 \quad y \xrightarrow{\frac{\lambda z}{K+x}} y+1 \quad z \xrightarrow{\lambda_z} z+1$$

$$x \xrightarrow{xz} x-1 \quad y \xrightarrow{yz} y-1 \quad z \xrightarrow{\beta_z z} z-1$$

**Appendix 1—figure 32.** System 9 — Fluctuating degradation rates. In the previous process, the X and Y components underwent first-order degradation with constant degradation time. The invariant of Eq. (LABEL:EQ:_causality_constraint_(manuscript)) holds when the degradation constants fluctuate as long as they are not affected by X or Y (see Section 4.1). To demonstrate that the invariant holds with fluctuations in the degradation constants, we simulated the above process where the degradation constant is given by the Z abundance, where the production of Z follows a Poisson process. This could model, for example, stochastic fluctuations in the abundances of an enzyme that degrades the X and Y components.

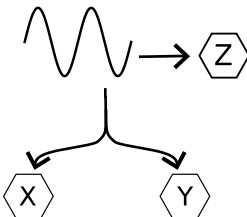

$$x \xrightarrow{A(\sin(\omega t)+1)V} x+1 \quad y \xrightarrow{B(\sin(\omega t)+1)V} y+1 \quad z \xrightarrow{C(\sin(\omega t)+1)V} z+1$$

$$x \xrightarrow{\beta xz} x-1 \quad y \xrightarrow{\beta yz} y-1 \quad z \xrightarrow{\beta_z z} z-1$$

**Appendix 1—figure 33.** Process 10 — Oscillating degradation rates. This process is similar to the previous process, but here all the components are driven by a shared upstream oscillating signal. This could model, for example, a system where a degradation enzyme is regulated by an oscillating signal.

## 12. Derivation of *Equation 2* in growing and dividing cells

### A. Abundances

The class of systems of *Equation 2* does not consider the effect on the X and Y abundances caused by random partitioning of molecules at cell division. Here we show that *Equation 2* still holds when cell division is added to the class of systems. In particular, the X and Y production and degradation reactions are governed by the same *Equations 4.3; 4.4*, with the addition that the shared unspecified production rate R can now depend on the cell volume V along with the other cellular components: $R(\mathbf{z}_{\mathrm{aff}}, \mathbf{z}_{\mathrm{aff}}^c, V)$.

Additionally, the cell volume divides at unspecified times $\{\tau_k\}$ which can vary over the cell ensemble. At these moments, the volume is reduced by multiplicative factors that can lie between 0 and 1. That is, $V \to a_k V$ at $\tau_k$, as we follow one of the daughter cells, where $\{a_k\}$ is a set of factors, with each $a_k$ satisfying $0 < a_k < 1$. As an example, in the case where cells divide perfectly symmetrically at every division, the factors would all be given by $a_k = 1/2$. In general, we allow for stochastic divisions in cell volume, that is we let the $a_k$ factors be randomly distributed along with the division times through unspecified ensemble distributions.

At division times, the molecular content of the cell is divided between the two daughter cells. To allow for random partitioning of molecules at cell division, we let the X and Y abundances of a single

lineage be reduced by unspecified distributions $\mathcal{A}_k(x)$ and $\mathcal{B}_k(y)$ that can vary over the divisions. We only specify that the averages satisfy $\langle \mathcal{A}_k(x) \rangle = a_k x$ and $\langle \mathcal{B}_k(y) \rangle = a_k y$. That is, on average, the abundances are reduced in proportion to the cell division factors. As an example, the $X$ and $Y$ abundances could be divided according to a binomial distribution with probability $a_k$ to remain in the tracked daughter cell.

In total, this amounts to adding the following 'reaction' to the class of systems

$$\left( V, \ x, \ y \right) \xrightarrow{\text{At time } \tau_k} \left( a_k V, \ \mathcal{A}_k(x), \ \mathcal{B}_k(y) \right) \quad \text{for } \tau_1, \ \tau_2, \ \tau_3 \dots$$

To derive *Equation 2*, we define the two disjoint sets $\mathbf{z}_{\text{aff}}$ and $\mathbf{z}_{\text{aff}}^c$ similarly to Section 4.1. In addition, we include the volume variable $V$ into the set of components not affected by X or Y. Similarly to Section. 5, we consider the average stochastic dual reporter dynamics conditioned on the history of their upstream influences: $\bar{x}(t) = \mathrm{E}\left[ x_t | \mathbf{z}_{\text{aff}}^c[-\infty, t], V[-\infty, t] \right]$ and $\bar{y}(t) = \mathrm{E}\left[ y_t | \mathbf{z}_{\text{aff}}^c[-\infty, t], V[-\infty, t] \right]$. All cell lineages in this conditional probability space have the same volume history, so they all undergo divisions at the same times $\tau_1, \tau_2, \dots$, with the same division factors $a_1, a_2, \dots$ (i.e. these are no longer random variables as they are conditioned on from the conditioning of the volume history). Between any two adjacent division times, $\tau_i$ and $\tau_{i+1}$, the time evolution is specified completely by the reactions in *Equations 4.3; 4.4*. That is, for $\tau_i < t < \tau_{i+1}$, the time evolution of the conditional averages is given by

$$\frac{d\bar{x}}{dt} = \bar{R}(t) - \bar{x}\beta(t) \quad \& \quad \frac{d\bar{y}}{dt} = \alpha\bar{R}(t) - \bar{y}\beta(t) \quad \text{when} \quad \tau_i < t < \tau_{i+1}, \tag{12.1}$$

where $\bar{R}(t) = E[R(\mathbf{z}_{\text{aff}}, \mathbf{z}_{\text{aff}}^c, V) | \mathbf{z}_{\text{aff}}^c[-\infty, t], V[-\infty, t]]$. Division occurs at $t = \tau_{i+1}$, at which point we have $V(\tau_{i+1}) \to a_{i+1} V(\tau_{i+1})$ and

$$\bar{x}(\tau_{i+1}) = \mathrm{E}\left[ x_{\tau_{i+1}} | \mathbf{z}_{\text{aff}}^c[-\infty, \tau_{i+1}], V[-\infty, \tau_{i+1}] \right] \to \mathrm{E}\left[ \mathcal{A}\left( x_{\tau_{i+1}} | \mathbf{z}_{\text{aff}}^c[-\infty, \tau_{i+1}], V[-\infty, \tau_{i+1}] \right) \right]$$

$$= a_{i+1}\bar{x}(\tau_{i+1}), \tag{12.2}$$

and similarly $\bar{y}(\tau_{i+1}) \to a_{i+1}\bar{y}(\tau_{i+1})$. This gives us the boundary conditions for the above differential equation. Dividing the right equation in *Equation 12.1* by $\alpha$, we find that the two differential equations are identical in $\bar{x}$ and $\bar{y}/\alpha$, and they have the same boundary conditions. Once the transience of the system has vanished, we are left with $\bar{x}(t) = \bar{y}(t)/\alpha$. It then follows analogously to the proof of Theorem 1 that $\eta_{xz_k} = \eta_{yz_k}$ when $Z_k$ is any component not affected by $X$ or $Y$.

## B. Concentrations

We now show that *Equation 2* holds in terms of the molecular concentrations of $X$ and $Y$. In particular, we let $x_c := x/V$ and $y := y/V$, where $x$ and $y$ denote the abundances of $X$ and $Y$, respectively. We condition on the history of the upstream influences and the volume:

$$\bar{x}_c(t) = \mathrm{E}\left[ \frac{x_t}{V_t} \middle| \mathbf{z}_{\text{aff}}^c[-\infty, t], V[-\infty, t] \right] = \frac{1}{V(t)} \mathrm{E}\left[ x_t | \mathbf{z}_{\text{aff}}^c[-\infty, t], V[-\infty, t] \right] = \frac{\bar{x}}{V(t)},$$

where $V(t)$ can be pulled out from the expectation brackets since it is specified by the conditioning and becomes equivalent to a constant at time $t$. When $\tau_i < t < \tau_{i+1}$, *Equation 12.1* governs the dynamics of $\bar{x}$ and $\bar{y}$. We use the product rule to find

$$\frac{d\bar{x}_c}{dt} = \frac{d}{dt}\left( \frac{\bar{x}}{V(t)} \right) = \bar{R}_c(t) - \bar{x}_c\beta(t) - \bar{x}_c\frac{V'(t)}{V(t)}, \tag{12.3}$$

where $R_c := R/V$ is interpreted as the production rate of the concentration $X_c$. At division time $\tau_{i+1}$, the cell volume is reduced $V \to a_{i+1}V$ and so is $\bar{x}$ (see *Equation 12.2*). We thus have $\bar{x}_c \to \frac{a_{i+1}\bar{x}}{a_{i+1}V} = \bar{x}_c$, i.e., $\bar{x}_c$ is unchanged at the division times. As a result, $\bar{x}_c(t)$ is continuous, meaning *Equation 12.3* holds for all $t$. Note that *Equation 12.3* holds for any volume dynamics between the division times, with the only requirement that it is a differential function between the division times.

We can then take the expectation of $\bar{x}_c$ over all possible histories of $\mathbf{z}_{\text{aff}}^c$ and all possible histories of the volume dynamics $V(t)$

$$llE[\bar{x}_c(t)]_{\text{histories}} = E\Big[E\big[x_t/V(t)|\mathbf{z}^c_{\text{aff}}[-\infty,t], V[-\infty,t]\big]\Big]_{\text{histories}}$$

$$= E\Big[E\big[x_t/V_t|\mathbf{z}^c_{\text{aff}}[-\infty,t], V[-\infty,t]\big]\Big]_{\text{histories}} = \langle x_t/V\rangle = \langle x_c\rangle, \qquad (12.4)$$

which follows from the law of total expectation. We now consider any component $Z_k$ not affected by $X$ or $Y$. It can be a molecular abundance, concentration, or another stochastic cellular variable like the growth rate of the cell. We then take the following expectation over all possible histories

$$llE[\bar{x}_c(t)z_k(t)]_{\text{histories}} = E\Big[E\big[x_{c,t}|\mathbf{z}^c_{\text{aff}}[-\infty,t], V[-\infty,t]\big]\cdot E\big[z_{k,t}|\mathbf{z}^c_{\text{aff}}[-\infty,t], V[-\infty,t]\big]\Big]_{\text{histories}}$$

$$= E\Big[E\big[x_{c,t}z_{k,t}|\mathbf{z}^c_{\text{aff}}[-\infty,t], V[-\infty,t]\big]\Big]_{\text{histories}} = \langle x_{c,t}z_{k,t}\rangle, \qquad (12.5)$$

where the second step comes from the fact that conditioning on the history of $\mathbf{z}^c_{\text{aff}}$ effectively conditions on the history of $z_k$ (so $x_c$ and $z_k$ are independent when conditioning on the $\mathbf{z}^c_{\text{aff}}$ history), the last step follows from the law of total expectation, $x_{c,t}$ is the concentration of $X$ at time $t$, $z_{k,t}$ is the measured quantity of $Z_k$ at time $t$, and $z_k(t) = E\big[z_{k,t}|\mathbf{z}^c_{\text{aff}}[-\infty,t], V[-\infty,t]\big]$ which is fully determined through the conditioning of the history of $\mathbf{z}^c_{\text{aff}}$ of which $Z_k$ is an element. From *Equations 12.4; 12.5*, it follows that

$$\text{Cov}\left(x_{c,t}, z_{k,t}\right) = \text{Cov}\left(\bar{x}_c(t), z_k(t)\right). \qquad (12.6)$$

Similarly, we can derive the equivalent differential equation for the Y concentration

$$\frac{d\bar{y}_c}{dt} = \alpha\bar{R}_c(t) - \bar{y}_c\beta(t) - \bar{y}_c\frac{V'(t)}{V(t)}, \qquad (12.7)$$

which holds for all $t$. Dividing *Equation 12.7* by $\alpha$, we find that the differential equations governing $\bar{x}_c(t)$ and $\bar{y}_c(t)/\alpha$ are identical. As a result, after the initial transience has decayed, we have $\bar{x}_c(t) = \bar{y}_c(t)/\alpha$. We can thus derive the analogues of *Equations 12.3; 12.4* for $\bar{y}/\alpha$. It then follows that

$$\langle x_c\rangle = \langle y_c\rangle/\alpha \quad \& \quad \text{Cov}(x_c, z_k) = \text{Cov}(y_c, z_k)/\alpha. \qquad (12.8)$$

Dividing $\text{Cov}(x_c, z_k)$ by $\langle x_c\rangle\langle z_k\rangle$, and $\text{Cov}(y_c, z_k)/\alpha$ by $\langle y_c\rangle\langle z_k\rangle/\alpha$, we find

$$\eta_{x_c z_k} = \eta_{y_c z_k}.$$

If we then re-label the variables and let $x$ and $y$ denote concentrations of $X$ and $Z$ respectively, we obtain *Equation 2*.

## 13. Simulation algorithm for growing and dividing cells

Many of the processes in *Appendix 1—table 1* have time-dependent rates through the time-varying cell volume $V(t)$. As a result, we employ the Gillespie algorithm *Gillespie, 1977* in combination with a 'trick' that allows us to simulate exact time trajectories of the abundances and concentrations with time-varying rates (*Voliotis et al., 2016*).

First, we simulate the division times and division factors for a single growing and dividing cell. There are three volume dynamics that we consider; see Materials and methods of the main text. In the first two, division times and division factors are constant. We set the constant division times to 1 and division factors to 1/2. The third is simulated with a simple Python script which produces an array of division times $\{\tau_i\}$ and division factors $\{a_i\}$ as described in the Materials and Methods of the main text.

The algorithm is then as follows:

1. The state of the system at time $t_0$ is $\mathbf{s} = (x, y, z, w)$. The waiting time $t$ for the next reaction event to occur is chosen by picking a random number from an exponential distribution with cumulative distribution function $F(t) = 1 - e^{-\int_t^{t_0+t} r_T(\mathbf{s},t)dt}$, where $r_T(\mathbf{s}, t)$ is the total rate of reaction events at time $t$:

$$r_T|(\mathbf{s}, t) = r_x^+(\mathbf{s}, t) + r_x^-(\mathbf{s}, t) + r_y^+(\mathbf{s}, t) + r_y^-(\mathbf{s}, t) + r_z^+(\mathbf{s}, t) + r_z^-(\mathbf{s}, t), r_w^+(\mathbf{s}, t) + r_w^-(\mathbf{s}, t)$$

with the rates given in S *Appendix 1—table 1* and the $t$ dependence comes through the time dependent functions $V(t)$ and $\sin(\omega t)$ in the rates. Determining the waiting time from this distribution requires numerically integrating $r_T(\mathbf{s}, t)$ which is computationally time-consuming while introducing numerical error. Instead, we use the following trick. We introduce a virtual component $\phi$ that follows the following null rate

$$\phi \xrightarrow{r_\phi(\mathbf{s},t)} \phi, \tag{13.1}$$

where $r_\phi(\mathbf{s}, t)$ is a function that satisfies $r_\phi(\mathbf{s}, t) + r_T(\mathbf{s}, t) = f(\mathbf{s}) > 0$. That is, adding $r_\phi(\mathbf{s}, t)$ to $r_T(\mathbf{s}, t)$ eliminates the explicit time dependence and leads to a positive function $f$ that only depends on the state $\mathbf{s}$. For example, in process 6 from *Appendix 1—table 1*, we can take $r_\phi(\mathbf{s}, t) = \max\{r_T(\mathbf{s}, t)\}_t - r_T(\mathbf{s}, t)$, where

$$\max\{r_T(\mathbf{s}, t)\}_t = 2AV_{max} + 2BV_{max} + 2CV_{max} + x\beta + y\beta + z\beta_z,$$

where $V_{max}$ is the max volume $V(t)$ takes throughout the simulation. For instance, if $V(t)$ exponentially grows from $V_0$ to $2V_0$ and divides back to $V_0$ with constant division times, then $V_{max} = 2V_0$ (the first volume dynamics we consider). In another example, process 3, we could have

$$r_\phi(\mathbf{s}, t) = (1 + \alpha)\lambda z / V_{min} + \lambda_z + x\beta + y\beta + z\beta_z - r_T(\mathbf{s}, t).$$

By introducing the null variable $\phi$ into the system, the total reaction rate becomes $\tilde{r}_T(\mathbf{s}, t) = r_\phi(\mathbf{s}, t) + r_T(\mathbf{s}, t) = f(\mathbf{s})$, a function that does not explicitly depend on time. As a result, since the abundances $\mathbf{s}$ do not change by definition of the waiting time in $(t_0, t_0 + t)$, the waiting time is picked from the following cumulative distribution function

$$F(t) = 1 - e^{-\int_{t_0}^{t_0+t} \tilde{r}_T(\mathbf{s},t)dt} = 1 - e^{-f(\mathbf{s})t}. \tag{13.2}$$

The waiting time $t$ for the next event to occur is thus generated through the cumulative distribution function given by *Equation 13.2*, which does not require numerical integration.

2. Check if a cell division $\{\tau_i\}$ occurred between time $t_0$ and $t_0 + t$. If so, the system time is updated to when the division occurs: $t_0 \to \tau_k$ where $\tau_k$ the particular division time that occurred. We then update the cell volume that divides with the division factor $a_k$: $V(t_0) \to a_k V(t_0)$. The abundances are then reduced due to cell division and random partitioning $(x, y, z, w) \longrightarrow (\text{Bin}(x, a_k), \text{Bin}(y, a_k), \text{Bin}(z, a_k), \text{Bin}(w, a_k))$, where $\text{Bin}(x, a_k)$ is the binomial distribution on $x$ with probability $a_k$. We then go back to step 1.

3. Else, if no division time occurred between $t_0$ and $t_0 + t$, then update the cell volume $V \to V(t_0 + t)$. Then determine which of the reactions occurs at the time $t_0 + t$, where the i-th reaction occurs with probability $\frac{r_i}{\tilde{r}_T}$. For instance, an $X$ molecule produced with probability $r_x^+(\mathbf{s}, t_0 + t)/\tilde{r}_T(\mathbf{s})$.

4. Update the system according to the reaction that was determined in the previous step. For instance, if the event turns out to be that an $X$ molecule is produced, the system is updated as

$$(x, \ y, \ z, \ w) \longrightarrow (x + 1, \ y, \ z, \ w).$$

Note that if the null reaction in *Equation 13.1* is picked, nothing occurs.

5. Update the time: $t_0 \to t_0 + t$. Go back to 1 and re-iterate.

## 14. Simulation of a 10-component system with a regulatory cascade

To demonstrate the invariant of *Equation 2* in a system with many components, we simulated the following birth-death process. The following reactions describe the dynamics of the components not affected by $X$:

$$z_1 \xrightarrow{1/(1+z_2^2)} z_1 + 1, \quad z_2 \xrightarrow{1/(1+z_3^2)} z_2 + 1, \quad z_3 \xrightarrow{1/(1+z_1^2)} z_3 + 1, \quad z_4 \xrightarrow{1/(1+z_1^2)} z_4 + 1,$$
$$z_5 \xrightarrow{z_3} z_5 + 1, \quad z_6 \xrightarrow{z_5} z_6 + 1, \quad z_7 \xrightarrow{z_1/(1+z_1)} z_7 + 1, \quad z_7 \xrightarrow{z_7/2} z_7 - 1. \tag{14.1}$$

The reactions that govern the $X$ dynamics, as well as the passive reporter $Y$, are given by

$$x \xrightarrow{2z_1/(1+x)} x+1, \quad y \xrightarrow{z_1/(1+x)} y+1,$$
$$x \xrightarrow{x} x-1, \quad y \xrightarrow{y} y-1. \tag{14.2}$$

Here, the production rate of $X$ and $Y$ depends linearly on $Z_1$, but is repressed through a negative feedback loop by $X$. The components that are affected by $X$ form a regulatory cascade with non-linear hill function rates. The reactions that describe the dynamics of the components affected by $X$ are:

$$z_8 \xrightarrow{x^3/(1+(x/2)^3)} z_8+1, \quad z_9 \xrightarrow{z_8^3/(1+(z_8/2)^3)} z_9+1, \quad z_{10} \xrightarrow{z_9^3/(1+(z_9/2)^3)} z_{10}+1,$$
$$z_8 \xrightarrow{z_8} z_8-1, \quad z_9 \xrightarrow{z_9} z_9-1, \quad z_{10} \xrightarrow{z_{10}} z_{10}-1. \tag{14.3}$$

In addition, we simulate growing and dividing cells, where the cell volume $V(t)$ divides by a factor of 2 at each cell division, with a fixed division time of $t_d = 1$. Between two subsequent divisions that occur at times $\tau_i$ and $\tau_{i+1}$ respectively, the volume grows exponentially: $V(t) = 2^{(t-\tau_i)}$. At each division, the abundances of each molecule from the reaction network above are reduced according to a binomial distribution with probability 0.5.

The reaction network topology for this system is shown in *Appendix 1—figure 34*. The normalized covariances computed from simulations of this network are plotted in *Appendix 1—figure 34B*. All components not affected by $X$ satisfy the invariant of *Equation 2*, whereas the three components that are affected by $X$ display a clear violation of the invariant.

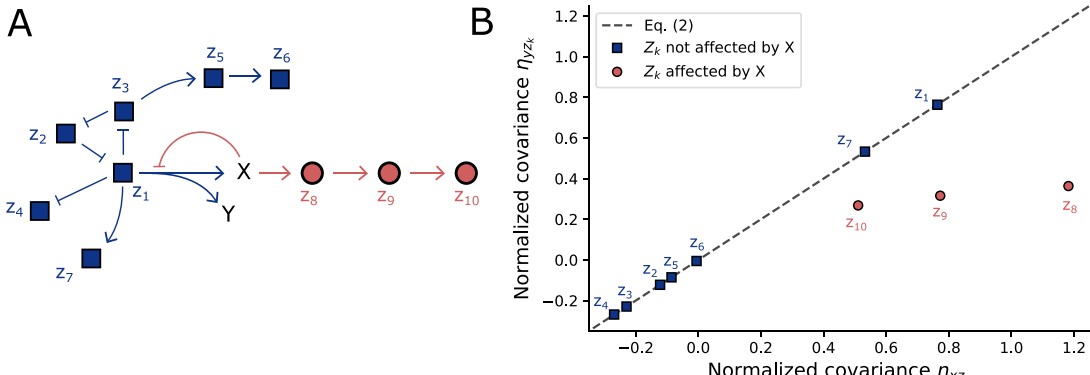

**Appendix 1—figure 34.** *Equation 2* is satisfied in an example network made of 10 components. (**A**) The reaction network topology for the system given by *Equations 14.1; 14.2; 14.3*. Blue squares are components not affected by $X$. Red circles are affected by $X$. (**B**) Plotted are the normalized covariances between the *concentrations* of $X$, $Y$ with those of the other components in the network. Components that are not affected by $X$ (blue squares) satisfy the invariant of *Equation 2*. Components affected by $X$ (red circles) violate the invariant. Note, the further down the cascade regulated by $X$, the closer the normalized covariances move towards the dashed line. This suggests that the distance from the dashed line is inversely proportional to the relative distances of components down regulatory cascades. Simulations were performed according to the algorithm described in Section 13. Normalized covariances were computed by integrating over the trajectories to obtain time averages for first and second moments, waiting for 10,000 cell divisions to occur before integrating to allow the system to reach stationarity. Each computed normalized covariance corresponds to the average of 40 independent simulations that ran for $10^6$ cell divisions. Estimated 95% confidence intervals (which are too minuscule to see) for the normalized covariances correspond to two times the standard error of the mean over these 40 simulations. Using error propagation, in each of the components not affected by X, the ratio $\eta_{xz_i}/\eta_{yz_i}$ produced 95% confidence intervals that encompassed the predicted value of 1. See Materials and methods for additional simulation details.

In *Appendix 1—figure 34B* we find that, of the components that are affected by $X$, the normalized covariances with $Z_8$ lie furthest away from the dashed line given by *Equation 2*, whereas the normalized covariances with $Z_{10}$ lie closest. This suggests that the "distance" down a regulatory cascade can scale inversely with the degree to which the normalized covariances violate *Equation 2*.

Note that the strength of an interaction is not the sole factor that affects the degree to which *Equation 2* is violated. In particular, though the dual reporters $X$ and $Y$ are identically regulated, they will differ due to the probabilistic nature of the reactions governing their dynamics. These

'intrinsic fluctuations' are what we exploit to detect causal interactions: when *X* affects a component of interest *Z*, but *Y* does not, then the intrinsic fluctuations in *X* will propagate to *Z*, but those from *Y* will not, thus creating an asymmetry between the normalized covariances. Therefore, the size of these intrinsic fluctuations affects the difference between the normalized covariances when there is a causal interaction. As a result, the degree to which *Equation 2* is violated cannot equate to an absolute measure for the distance down a regulatory cascade, because both the strength of the interaction and the size of the intrinsic fluctuations in *X* affect the degree of violation.

However, it might be possible to infer the *relative* distance down a cascade. That is, in a given network, the normalized covariances between *X*, *Y*, and two other components of interest $Z_i$, $Z_j$ that are affected by *X* can be compared. If the asymmetry between $\eta_{xz_i}$ and $\eta_{yz_i}$ is larger than the asymmetry between $\eta_{xz_j}$ and $\eta_{yz_j}$, then we might be able to conclude that *X* affects $Z_i$ with a stronger interaction than the interaction from *X* to $Z_j$, because here the intrinsic fluctuations in *X* are the same in both cases.

Note that in the particular example in *Appendix 1—figure 34A*, the definition of "distance" down the regulatory cascade is clear. However, if there exists feedback from any of the components in the cascade back onto *X* and *Y*, then this definition is not well defined.

## 15. Fluctuations in plasmid copy numbers can reduce the degree of violation of *Equation 2*

All 4 of the positive control synthetic circuits used in this study have the target *geneZ* placed on a plasmid. All genes placed on the same plasmid will be impacted by the shared fluctuations that originate from plasmid copy number fluctuations. If the plasmid fluctuations are large enough, they can reduce any differences between $\eta_{xz}$ and $\eta_{yz}$ that result from a causal interaction from *X* to *Z*. To demonstrate this, we simulated the following two birth-death processes. The following reactions describe the dynamics of an open-loop regulatory cascade:

$$w \xrightarrow{p\lambda} w+1, \quad x \xrightarrow{p\lambda/\left(1+\left(\frac{w^2}{K}\right)\right)} x+1, \quad y \xrightarrow{p\lambda/\left(1+\left(\frac{w}{K}\right)^2\right)} y+1, \quad z \xrightarrow{p\lambda/\left(1+\left(\frac{z}{K}\right)^2\right)} z+1, \text{ (15.1)}$$

$$w \xrightarrow{w/\tau} w-1, \quad x \xrightarrow{x/\tau} x-1, \quad y \xrightarrow{y/\tau} y-1, \quad z \xrightarrow{x/\tau} z-1.$$

Here, *W* represses the production *X* and *Y*, while *X* represses the production of *Z*. The production rates are proportional to the plasmid copy number *p* which can also fluctuate. The following reactions describe the dynamics of a closed-loop regulatory cascade:

$$w \xrightarrow{p\lambda/\left(1+\left(\frac{z}{K}\right)^3\right)} w+1, \quad x \xrightarrow{p\lambda/\left(1+\left(\frac{w}{K}\right)^3\right)} x+1, \quad y \xrightarrow{p\lambda/\left(1+\left(\frac{w}{K}\right)^3\right)} y+1, \quad z \xrightarrow{p\lambda/\left(1+\left(\frac{z}{K}\right)^3\right)} z.$$

$$w \xrightarrow{w/\tau} w-1, \quad x \xrightarrow{x/\tau} x-1, \quad y \xrightarrow{y/\tau} y-1, \quad z \xrightarrow{x/\tau} z-1.$$

(15.2)

Here, *W* represses the production *X* and *Y*, *X* represses the production of *Z*, and *Z* represses the production of *W*. The production rates are proportional to the plasmid copy number *p* which can also fluctuate. We simulate the above two systems in two cases. In the first case, there are no plasmid copy number fluctuations, setting *p* = 1 at all times. This corresponds to the case where the genes are all located on the chromosome. In the second case, there are plasmid copy number fluctuations with dynamics modeled by the following birth-death process:

$$p \xrightarrow{\lambda_p} p+1, \quad p \xrightarrow{p/\tau_p} p-1.$$

(15.3)

We randomly sample parameters for these systems and find that in the majority of cases, there is a larger degree of violation of *Equation 2* in the case without plasmid fluctuations, see *Appendix 1—figure 35*.

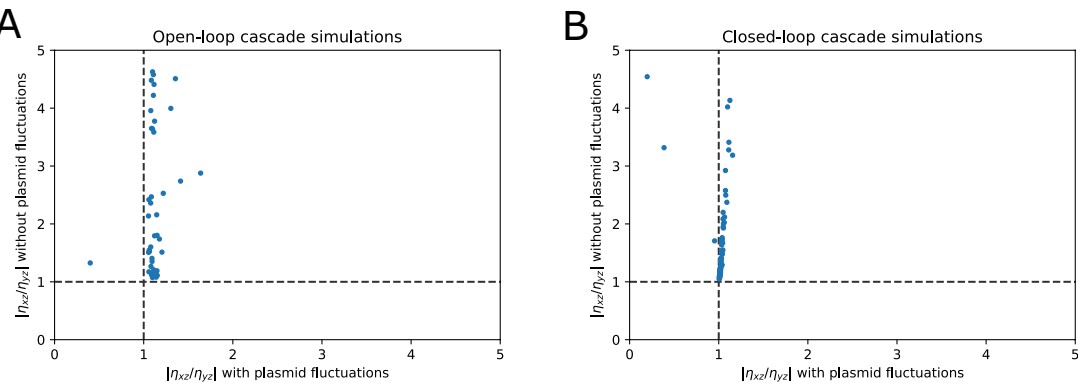

**Appendix 1—figure 35.** Fluctuations in plasmid copy numbers reduce the degree of violation of *Equation 2* in simulated test cases. (**A**) Results of the simulations of the open-loop cascade system given by *Equation 15.1*. For each randomly sampled set of parameters, the system was simulated without plasmid fluctuations ($p = 1$ is constant) and with plasmid fluctuations (with dynamics given by *Equation 15.3*). The degree of violation of the invariant of *Equation 2* was computed as the absolute value of the ratio of the normalized covariances $|\eta_{xz}/\eta_{yz}|$. As $X$ in this system represses the $Z$ production rate, there can be a deviation from $|\eta_{xz}/\eta_{yz}| = 1$. We find that the degree of violation of the invariant tends to be bigger in the case without plasmid fluctuations. (**B**) Similar to A but for simulations of the closed-loop cascade system given by *Equation 5.12*. We find that the degree of violation of the invariant tends to be bigger in the case without plasmid fluctuations. For both the open-loop and closed-loop systems, 100 sets of parameters were chosen randomly, with $\tau = 1$, $\lambda = 50r_1$ and $K = 10r_2$ where $r_i$ are uniformly distributed random numbers in (0,1). Plasmid copy number fluctuations were simulated with $\tau_p = 5r_3$ and $\lambda_p = 5/\tau_p$ which sets the average copy number to 5. Each simulation was performed using the Gillespie algorithm and ran until each reaction occurred at least $10^5$ times.

## 16. The invariant relation holds in the face of measurement noise

Here we show that *Equation 2* holds when different types of measurement noise models are added to the class of dual reporter systems given by *Equations 4.3; 4.4*.

### A. Multiplicative noise

Let $M_x$ and $M_y$ be identically distributed random variables, independent of $x$ and $y$, that model a multiplicative noise factor for $X$ and $Y$ respectively. If $x_m$ and $y_m$ are the measured abundances of $X$ and $Y$ respectively, then we have

$$x_m = M_x x \quad \& \quad y_m = M_y y, \tag{16.1}$$

where $x$ and $y$ are the true abundances of $X$ and $Y$ respectively. Let $Z_k$ be a component not affected by $X$ or $Y$. If there is measurement noise that affects the $Z_k$ measurement, then we let it be arbitrary and include it in the $z_k$ variable. We only assume that any dependence of $M_x$ and $M_y$ on $z_k$ that occurs as a result of measurement noise is the same, that is $E[M_x|z_k] = E[M_y|z_k]$. We use the law of total expectation as follows

$$\langle M_x x z_k \rangle = E\left[E\left[M_x x z_k | \mathbf{z}_{\text{aff}}^c[-\infty, t], M_x = m\right]\right] = E\left[m z_k(t) E\left[x | \mathbf{z}_{\text{aff}}^c[-\infty, t]\right]\right] = E\left[m z_k(t) \bar{x}(t)\right],$$

where $z_k(t)$ is the determined $z_k$ trajectory resulting from conditioning on the history of $\mathbf{z}_k^c$, in the first step we condition on the history of the components not affected by $X$ or $Y$ along with the value of $M_x$, and where the expectation on the right is taken over all the possible histories of $\mathbf{z}_{\text{aff}}^c$ and all the possible values of $M_x$. Similarly, we have $\langle M_y y z_k \rangle = E[m z_k(t) \bar{y}(t)]$. According to the paragraph above *Equation 5.16*, we have $\bar{x}(t) = \bar{y}(t)/\alpha$, therefor since $M_x$ and $M_y$ are identically distributed, we have

$$\langle M_x x z_k \rangle = \langle M_y y z_k \rangle / \alpha.$$

Similarly, we can derive

$$\langle M_x x \rangle = \langle M_y y \rangle / \alpha.$$

It follows from the above two equations and *Equation 16.1* that $\eta_{x_m z_k} = \eta_{y_m z_k}$.

## B. Additive noise

Let $A_x$ and $A_y$ be random variables, with $\langle A_x \rangle = \langle A_y \rangle = 0$, that model an additive noise term for $X$ and $Y$ respectively. If $x_m$ and $y_m$ are the measured abundances of $X$ and $Y$ respectively, then we have

$$x_m = x + A_x \quad \& \quad y_m = y + A_y, \tag{16.2}$$

where $x$ and $y$ are the true abundances of $X$ and $Y$ respectively. It follows that

$$\langle x_m \rangle = \langle x \rangle \quad \& \quad \langle y_m \rangle = \langle y \rangle.$$

Let $Z_k$ be another cellular component. We assume that the additive noise terms in $X$ and $Y$ are statistically independent of $z_k$. We thus have

$$\mathrm{Cov}(x_m, z_k) = \mathrm{Cov}(x, z_k) \quad \& \quad \mathrm{Cov}(y_m, z_k) = \mathrm{Cov}(y, z_k).$$

It follows that $\eta_{x_m z_k} = \eta_{x z_k}$ and $\eta_{y_m z_k} = \eta_{y z_k}$. If $Z_k$ is a component not affected by $X$ or $Y$, then the invariant relation of *Equation 2* must hold, and so $\eta_{x_m z_k} = \eta_{y_m z_k}$.

## C. Binomial readout and undercounting noise

Here we show how stochastic undercounting affects the invariant relation of *Equation 2*. This is motivated by the fact that common experimental methods can lead to a systematic undercounting of the mRNA levels and protein levels. For example, in fluorescence in situ hybridization, fluorescent probes only bind to mRNA molecules with a fixed probability. We would thus like to know how the invariant relation of *Equation 2* changes when the reporter abundances $X$ and $Y$ are detected with fixed probabilities $p_x$ and $p_y$, respectively. We let $x_m$ and $y_m$ correspond to the measured $X$ and $Y$ abundances, respectively. They are defined as

$$x_m = B(x, p_x) \quad \& \quad y_m = B(y, p_y),$$

where $B(n, p)$ is a binomial distribution with $n$ trials with probability $p$. That is, when each molecule is detected with a fixed probability, the resulting measurement is a binomial readout of the true abundance. Taking the expectation, we have

$$\langle x_m \rangle = E[B(x, p_x)] = E\big[E[B(x, p_x|x)]\big] = E[p_x x] = p_x \langle x \rangle.$$

Now calculating the covariance with cellular component $Z_k$, we have

$$\langle x_m z \rangle = E[B(x, p_x) z_k] = E\Big[E\big[B(x, p_x) z_k | x, z_k\big]\Big] = E\Big[z_k E[B(x, p_x)|x]\Big] = E[p_x x z_k] = p_x \langle xz \rangle$$

$$\Rightarrow \mathrm{Cov}(x_m, z_k) = \langle x_m z_k \rangle - \langle x_m \rangle \langle z_k \rangle = p_x \mathrm{Cov}(x, z_k).$$

As a result, we have

$$\eta_{x_m z} = \eta_{xz} \quad \& \quad \eta_{y_m z} = \eta_{yz},$$

where the second equation follows by symmetry. Therefore, the normalized covariances are unaffected by the molecule detection probabilities. The invariant relation of *Equation 2* thus holds in the face of systematic undercounting of molecules, allowing for arbitrary (and different) detection probabilities for $X$ and $Y$.

## D. Poisson-Gaussian noise model

Here we consider the case where measurement noise is introduced in fluorescent microscopy images. Fluorescent imaging noise is often modeled by the Poisson-Gaussian noise model which we consider here (*Yang, 2015*). If $\tilde{x}_m$ and $\tilde{y}_m$ are the measured intensities of the $X$ and $Y$ protein fluorescence, then we have

$$\tilde{x}_m = \frac{1}{\chi}\mathcal{P}(\chi\tilde{x}) + \mathcal{N}(0, b),$$

where $\chi \geq 0$ and $b \geq 0$ are arbitrary constants, $\mathcal{P}$ is the Poisson distribution, $\mathcal{N}$ is the normal (Gaussian) distribution, and $\tilde{x}$ is the original fluorescent signal which is proportional to the abundance $x$. Taking the expectation, we have

$$
\begin{aligned}
ll\langle\tilde{x}_m\rangle &= E\left[\frac{1}{\chi}\mathcal{P}(\chi\tilde{x})\right] + E\left[\mathcal{N}(0, b)\right] \\
&= \frac{1}{\chi}E\left[E[\mathcal{P}(\chi\tilde{x})|\tilde{x}]\right] + 0 \\
&= \frac{1}{\chi}E\left[\chi\tilde{x}\right] = \langle\tilde{x}\rangle = f_x\langle x\rangle,
\end{aligned}
\tag{16.3}
$$

where $f_x$ is an unknown proportionality constant between the $\tilde{x}$ signal and the protein abundance. That is, the expectation of the measured signal is the same as the expectation of the original signal. We now consider cellular component $Z_k$, which can correspond to the signal of another fluorescent protein in the cell and can have arbitrary measurement noise. Assuming that the measurement noise in $Z_k$ is independent of the measurement noise in $X$ and $Y$, which could be the case if three different images are taken in three different imaging channels, we take the covariance

$$
\begin{aligned}
\langle\tilde{x}_m z_k\rangle &= E\left[\frac{1}{\chi}z_k\mathcal{P}(\chi x)\right] + E\left[z\mathcal{N}(0, b)z_k\right] \\
&= \frac{1}{\chi}E\left[E\left[z_k\mathcal{P}(\chi\tilde{x})|\tilde{x}, z_k\right]\right] + E\left[E\left[z_k\mathcal{N}(0, b)|\tilde{x}, z_k\right]\right] \\
&= \frac{1}{\chi}E\left[\chi\tilde{x}z_k\right] + 0 = \langle\tilde{x}z_k\rangle.
\end{aligned}
$$

Therefore

$$\text{Cov}(\tilde{x}_m, z_k) = \text{Cov}(\tilde{x}, z_k) = f_x\text{Cov}(x, z_k),\tag{16.4}$$

where $f_x$ is an unknown proportionality constant between the $\tilde{x}$ signal and the protein abundance. Putting **Equations 16.3; 16.4** together, we have

$$\eta_{\tilde{x}_m z_k} = \eta_{xz_k} \quad \& \quad \eta_{\tilde{y}_m z_k} = \eta_{yz_k},$$

where the second equation follows by symmetry. The normalized covariances are thus not affected by the Poisson-Gaussian noise, and the invariant equation of **Equation 2** can still be exploited.

## E. Segmentation noise

Let $\tilde{x}_m$ be the signal of the fluorescent $X$ proteins that we measure. The image is made up of pixels which we index with $p$, where the $p$-th pixel has intensity $I_p$. Each cell is segmented and has cell area $A$ and segmentation area $S$. The measured signal is thus

$$\tilde{x}_m = \sum_{p \in S p \in A} I_p + \sum_{p \in S p \notin A} I_p.$$

The second term corresponds to additive noise from whenever segmentation fits an area larger than the cell area. This corresponds to fluctuations in the image background. Assuming that the background can be removed and that these fluctuations are negligible compared to the fluorescent protein signal, we can set this term to zero. Moreover, assuming that the fluorescent proteins X are evenly distributed in the cell, then the third term on the right is just the fraction of the total signal that is in the segmentation area

$$\sum_{p \in A p \in S} I_p = f_x x\theta,$$

where $\theta = S/A$ when $S < A$ and $\theta = 1$ when $S \geq A$, and $f_x$ is an unknown proportionality constant between the fluorescence and the abundance. We thus have

$$\tilde{x}_m = f_x \theta x,$$

We now let $\theta$ be a stochastic variable that we include in the cloud of variables not affected by $X$, that is, the segmentation noise is not affected by $x$ through an unspecified Markov chain. In that case, when we condition on the history of upstream variables in $X$, we can still write down *Equation 5.5*, and we still have

$$\bar{x}(t) = \bar{y}(t)/\alpha.$$

However, now we are also conditioning on the histories of $\theta$, which means that

$$llf_x\theta(t)\bar{x}(t) = f_y\theta(t)\bar{y}(t)/(\alpha f_y/f_x)$$

$$\Rightarrow E\left[\tilde{x}_m|\mathbf{z}_{\text{aff}}^c[-\infty, t]\right] = E\left[\tilde{y}_m|\mathbf{z}_{\text{aff}}^c[-\infty, t]\right]/(\alpha f_y/f_x)$$

$$\Rightarrow E\left[E\left[\tilde{x}_m|\mathbf{z}_{\text{aff}}^c[-\infty, t]\right]\right] = E\left[E\left[\tilde{y}_m|\mathbf{z}_{\text{aff}}^c[-\infty, t]\right]\right]/(\alpha f_y/f_x)$$

$$\Rightarrow \langle\tilde{x}_m\rangle = \langle\tilde{y}_m\rangle/(\alpha f_y/f_x).$$

Moreover, we have

$$\langle\tilde{x}_m z_k\rangle = E\left[E\left[f_x\theta x z_k|\mathbf{z}_{\text{aff}}^c[-\infty, t]\right]\right] = f_x E\left[\theta(t)\bar{x}(t)z_k(t)\right] = \langle\tilde{y}_m z_k\rangle/(\alpha f_y/f_x).$$

We thus have $\eta_{\tilde{x}_m z_k} = \eta_{\tilde{y}_m z_k}$, and *Equation 2* holds in the face of the segmentation noise modeled here.

## 17. Estimating confidence intervals

The corrections from the preceding sections rely on the distribution of measurements. As a result, sampling error affects the corrections and their accuracy, along with the final estimators of the normalized covariances. Here, we show two methods we employed to estimate the confidence intervals for the normalized covariances, taking into account sampling error and its effect on the corrections discussed in the preceding sections. The computed normalized covariances and their confidence intervals for both methods are shown in *Appendix 1—figure 16*. The two methods give similar results (*Figure 3D,E*, *Appendix 1—figure 6*).

### A. Sampling with replacement (bootstrapping)

The goal is to estimate the error bars for a normalized covariance measurement of a strain of cells in a mother machine experiment. Here we use bootstrapping, where the data corrections and the normalized covariances are computed over many samples of the data, allowing for replacement in sampling. If the experiment produces N fluorescence time-traces of cells with plasmid and M fluorescence time-traces of cells that lost the plasmid, the following pipeline computes the normalized covariance between the CFP and RFP proteins (the YFP and RFP case follows anagolously), see *Appendix 1—figure 16A*.

1. Take a random sample of size N of the cells with plasmid, allowing for replacement. Take another random sample of size M of the cells that lost the plasmid, allowing for replacement.

2. Use the sample of size N to compute the temporal drift curves of the CFP and RFP channels, and drift correct the data from both samples according to the procedure outlined in the temporal drift section above.

3. Compute the uneven illumination correction curves of the CFP and YFP channels, using the sample of size N, as outlined in the uneven illumination section above. Correct the data from both samples with the uneven illumination correction curves.

4. From the sample of size M (lost plasmid), pool data into a single distribution and compute the average CFP fluorescence. This corresponds to the autofluorescence and media fluorescence of the CFP channel.

5. Remove the autofluorescence and media fluorescence (obtained in the previous step) from all the CFP measurements from the sample of size N.

6. From the sample of size N, pool data into a single distribution and compute the normalized covariance between CFP and RFP of the sample $\eta^{sample}$.

7. Go back to step 1 and repeat 100 times, saving each $\eta_{sample}$

8. When 100 normalized covariances $\eta_{sample}$ have been computed, take the average as the final $\eta$ estimate, with confidence intervals given by 2 times the standard deviation of the 100 normalized covariances $\eta_{sample}$.

## B. Sampling without replacement (splitting data)

The goal is to estimate the error bars for a normalized covariance measurement of a strain of cells in a mother machine experiment. Here we divide the data from an experiment into $N_{samples}$ disjoint sets. The data corrections and the normalized covariances are computed in each set. If the experiment produces N fluorescence time-traces of cells with plasmid and M fluorescence time-traces of cells that lost the plasmid, the following pipeline computes the normalized covariance between the CFP and RFP proteins (the YFP and RFP case follows anagolously), see *Appendix 1—figure 16B*.

1. Randomly divide the N cell traces into $N_{samples} = \min(M, 10)$ disjoint samples (each cell is in one and only one sample). Also randomly split the M cell traces with lost plasmid into $N_{samples}$ disjoint sets. This ensures that each disjoint set has at least 1 cell that lost the plasmid to estimate the autofluorescence, but we cap the number of sets to 10 to allow for enough cells in each set. Most strains had more than 10 cells that lost the plasmid in a single mother machine experiment (the lowest was 7).

2. Take a sample of cells with plasmid and a sample of cells that lost plasmid. Compute the temporal drift curves of the CFP and RFP channels using the former sample, and drift correct the data from both samples according to the procedure outlined in the temporal drift section above.

3. Compute the uneven illumination correction curves of the CFP and YFP channels, using the sample of cells with plasmid, as outlined in the uneven illumination section above. Correct the data from both samples with the uneven illumination correction curves.

4. From the sample of size M (lost plasmid), pool data into a single distribution and compute the average CFP fluorescence. This corresponds to the autofluorescence and media fluorescence of the CFP channel.

5. Remove the autofluorescence and media fluorescence (obtained in the previous step) from all the CFP measurements from the sample of cells with plasmid.

6. From the sample of cells with plasmid, pool data into a single distribution and compute the normalized covariance between CFP and RFP of the sample $\eta^{sample}$.

7. Go back to step 2 and repeat until $\eta^{sample}$ has been computed for each of the $\eta^{sample}$ samples.

8. When the $N_{samples}$ normalized covariances $\eta_{sample}$ have been computed, take the average as the final $\eta$ estimate, with confidence intervals given by 2 times the standard error of mean. Note that the denominator of the standard error is given by $\sqrt{N_{samples} - 1}$ because a degree of freedom is used to compute the average.

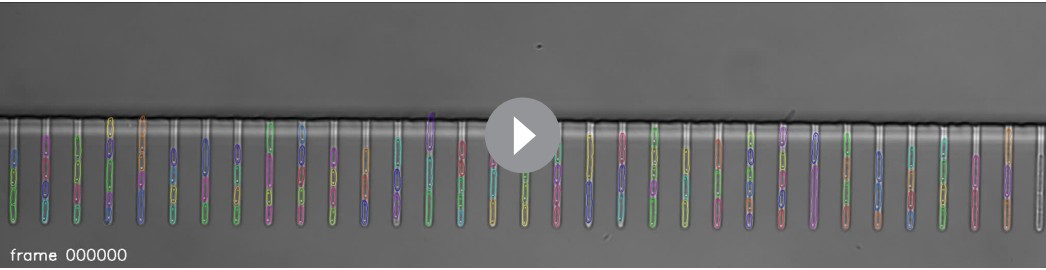

**Appendix 1—video 1.** Bright field images of a single imaging position of a mother machine experiment with colored cell boundaries produced from the DelTA segmentation pipeline. Here, segmentation was done on bright field images.

https://elifesciences.org/articles/92497/figures#video1

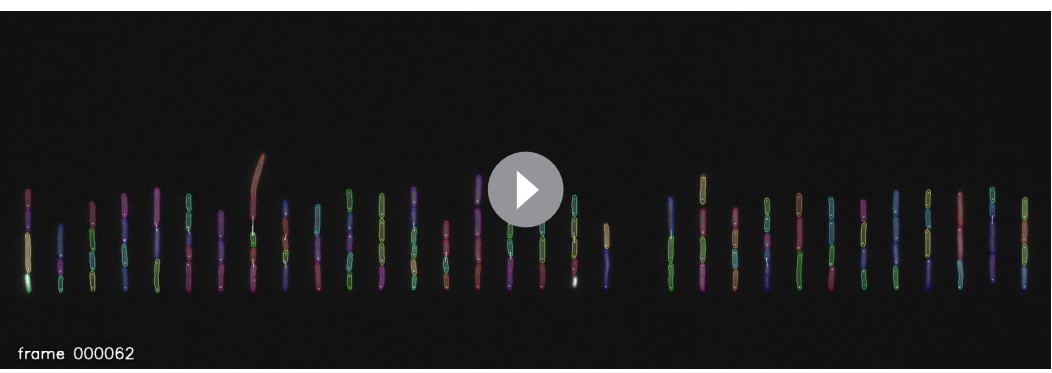

frame 000062

**Appendix 1—video 2.** Fluorescent images of a single imaging position of a mother machine experiment with colored cell boundaries produced from the DelTA segmentation pipeline. Here, segmentation was done on the RFP channel, which produced similar results to those in *Figure 3*.

https://elifesciences.org/articles/92497/figures#video2

