## [Editor Report · eLife Assessment]

By taking advantage of noise in gene expression, this **important** study introduces a new approach for detecting directed causal interactions between two genes without perturbing either. The main theoretical result is supported by a proof. Preliminary simulations and experiments on small circuits are **solid**, but further investigations are needed to demonstrate the broad applicability and scalability of the method.

---

## [Referee Report · Reviewer #2 (Public Review)]

Summary:

This paper describes a new approach to detecting directed causal interactions between two genes without directly perturbing either gene. To check whether gene X influences gene Z, a reporter gene (Y) is engineered into the cell in such a way that (1) Y is under the same transcriptional control as X, and (2) Y does not influence Z. Then, under the null hypothesis that X does not affect Z, the authors derive an equation that describes the relationship between the covariance of X and Z and the covariance of Y and Z. Violation of this relationship can then be used to detect causality.

The authors benchmark their approach experimentally in several synthetic circuits. In 4 positive control circuits, X is a TetR-YFP fusion protein that represses Z, which is an RFP reporter. The proposed approach detected the repression interaction in 2 of the 4 positive control circuits. The authors constructed 16 negative control circuit designs in which X was again TetR-YFP, but where Z was either a constitutively expressed reporter, or simply the cellular growth rate. The proposed method detected a causal effect in two of the 16 negative controls, which the authors argue is perhaps not a false positive, but due to an unexpected causal effect. Overall, the data support the potential value of the proposed approach.

Strengths:

The idea of a "no-causality control" in the context of detected directed gene interactions is a valuable conceptual advance that could potentially see play in a variety of settings where perturbation-based causality detection experiments are made difficult by practical considerations.

By proving their mathematical result in the context of a continuous-time Markov chain, the authors use a more realistic model of the cell than, for instance, a set of deterministic ordinary differential equations.

The authors have improved the clarity and completeness of their proof compared to a previous version of the manuscript.

Limitations:

The authors themselves clearly outline the primary limitations of the study: The experimental benchmark is a proof of principle, and limited to synthetic circuits involving a handful of genes expressed on plasmids in *E. coli*. As acknowledged in the Discussion, negative controls were chosen based on the absence of known interactions, rather than perturbation experiments. Further work is needed to establish that this technique applies to other organisms and to biological networks involving a wider variety of genes and cellular functions. It seems to me that this paper's objective is not to delineate the technique's practical domain of validity, but rather to motivate this future work, and I think it succeeds in that.

Might your new "Proposed additional tests" subsection be better housed under Discussion rather than Results?

I may have missed this, but it doesn't look like you ran simulation benchmarks of your bootstrap-based test for checking whether the normalized covariances are equal. It would be useful to see in simulations how the true and false positive rates of that test vary with the usual suspects like sample size and noise strengths.

It looks like you estimated the uncertainty for eta_xz and eta_yz separately. Can you get the joint distribution? If you can do that, my intuition is you might be able to improve the power of the test (and maybe detect positive control #3?). For instance, if you can get your bootstraps for eta_xz and eta_yz together, could you just use a paired t-test to check for equality of means?

The proof is a lot better, and it's great that you nailed down the requirement on the decay of beta, but the proof is still confusing in some places:

On pg 29, it says "That is, dividing the right equation in Eq. 5.8 with alpha, we write the ..." but the next equation doesn't obviously have anything to do with Eq. 5.8, and instead (I think) it comes from Eq 5.5. This could be clarified.

Later on page 29, you write "We now evoke the requirement that the averages xt and yt are stationary", but then you just repeat Eq. 5.11 and set it to zero. Clearly you needed the limit condition to set Eq. 5.11 to zero, but it's not clear what you're using stationarity for. I mean, if you needed stationarity for 5.11 presumably you would have referenced it at that step.

It could be helpful for readers if you could spell out the practical implications of the theorem's assumptions (other than the no-causality requirement) by discussing examples of setups where it would or wouldn't hold.

---

## [Author Response]

The following is the authors’ response to the previous reviews

We have made the following small adjustments and resubmit the manuscript to be published as a Version of Record with eLife.

Changes in main text of the manuscript:

We have moved the “Proposed additional tests” subsection to the Discussion section as suggested by the referee.

We have added a link to a Github repository and a link to a Zenodo data repository at the beginning of the Materials and Methods section in the “Data and materials availability” subsection. The Github repository contains simulation code and data, and single-cell data analysis code. The Zenodo link contains our experimental data (we await your confirmation before we publish it officially on Zenodo).

Changes in the supplemental information files

We have fixed the typo on page 29 of the SI in which Eq. (8) was referred to in a derivation. It should be Eq. (5) instead. We thank the referee for catching this mistake which has now been corrected.

We have fixed a typo on page 29 of SI, in which the word “evoke” is now “invoke”.

We have clarified the derivation on page 29 of the SI. The referee is correct that the limit condition was used to set the right-hand side of Eq. (5.11) to zero.